# Dogs were widely distributed across western Eurasia during the Palaeolithic

William A. Marsh[1,36 ✉], Lachie Scarsbrook[2,3,36 ✉], Eren Yüncü[4], Lizzie Hodgson[5], Audrey T. Lin[6,7], Maria De Iorio[8,9], Olaf Thalmann[2], Mark G. Thomas[10], Mahaut Goor[2], Anders Bergström[11], Angela Noseda[12,13], Sarieh Amiri[14,15], Fereidoun Biglari[16], Dušan Borić[17], Katia Bougiouri[18], Alberto Carmagnini[2], Maddalena Gianni[19], Tom Higham[19,20], Ophelie Lebrasseur[3,21], Anna Linderholm[22,23], Marcello A. Mannino[24], Caroline Middleton[25], Gökhan Mustafaoğlu[26], Angela Perri[27], Joris Peters[28,29], Mike Richards[30], Özlem Sarıtaş[31], Pontus Skoglund[32], Rhiannon E. Stevens[33], Chris Stringer[1], Kristina Tabbada[3], Helen M. Talbot[5], Laura G. Van der Sluis[19,20], Silvia M. Bello[1], Vesna Dimitrijevic[34], Louise Martin[33], Marjan Mashkour[12,14], Simon A. Parfitt[1,33], Sonja Vukovic[34], Selina Brace[1], Oliver E. Craig[5], Douglas Baird[25], Sophy Charlton[5], Greger Larson[3,37 ✉], Ian Barnes[1,37 ✉] & Laurent A. F. Frantz[2,35,37 ✉]

Archaeological evidence suggests that dogs diverged from wolves during the Palaeolithic, more than 15,000 years ago[1–7]. The earliest unequivocal genetic evidence, however, is associated with dog remains from Mesolithic archaeological contexts approximately 10,900 years ago[8,9]. Here we generate both nuclear and mitochondrial genomes from canid remains at Pınarbaşı in Türkiye (15,800 years ago)[10] and Gough's Cave in the UK (14,300 years ago)[11], as well as from dogs excavated from two Mesolithic sites in Serbia (Padina between 11,500–7,900 years ago and Vlasac 8,900 years ago)[12,13]. Our analyses indicate that a genetically homogeneous dog population was already widely distributed across Europe and Anatolia during the Late Upper Palaeolithic (by at least 14,300 years ago). This finding suggests that dogs were exchanged among genetically and culturally distinct western Eurasian Late Palaeolithic human populations, namely the Magdalenian, Epigravettian and Anatolian hunter-gatherers[10,14–16]. Last, we identify a major influx of eastern Eurasian dog ancestry during the Mesolithic, concomitant with the movement of eastern hunter-gatherer populations into Europe[14], which led to the establishment of the primary ancestry characteristics that define European dog populations today.

Despite the application of both molecular and morphological approaches, the temporal and geographic origins of dog domestication remain unknown. Previous estimates have varied from 135,000 to 15,000 years ago[7], which reflects both the large uncertainties associated with genetic-based dating methods and the difficulties in distinguishing skeletal remains of dogs and wolves on the basis of morphology. This is especially true in the earliest stages of domestication, during which there may have been an absence of detectable lineage-defining characteristics[17–19].

Morphological analyses of archaeological canid remains have suggested that dogs were present across Eurasia during the Upper Palaeolithic[1–6] (35,000–15,000 years ago). Stable dietary isotopes ($\delta^{13}C$ and $\delta^{15}N$) have also been used to tentatively distinguish dogs from wolves in Palaeolithic contexts[20]. Without corroborating nuclear genomic data, however, definitive identification of these remains as dogs has been challenging. For instance, although Upper Palaeolithic remains from Europe (for example, approximately 34,000 years calibrated before present (cal BP) from Goyet in Belgium[1]; approximately 28,500 years cal BP from Předmostí in Czechia[2]) and Russia

(for example, approximately 33,000 years cal BP from Razboinichya Cave in the Altai Mountains[3]; approximately 17,600 years cal BP from Eliseevichi in western Russia[5]) were initially identified as early dogs using morphological approaches, nuclear genomes generated from these individuals have shown they belonged to now-extinct wolf populations[21,22]. The earliest definitive dogs, on the basis of nuclear genetic data, have been identified at the Mesolithic site of Veretye (Karelia, Russia), dated to approximately 10,900 years cal BP (ref. 8).

Although Palaeolithic dogs are yet to be unequivocally identified, several candidates have been suggested. An approximately 14,300 years cal BP canid mandible from the Late Upper Palaeolithic site of Bonn-Oberkassel (Germany), for example, shows dog-like morphology, was co-interred alongside a dual human burial and shows pathologies that would have been lethal if not for prolonged human care[23]. Claims for Palaeolithic dogs, based on either morphological or biomolecular analyses, or both, have also been made for material from sites across continental Europe including Kesserloch (Switzerland[24,25]), Erralla (Spain[26]), Grotta Paglicci (Italy[27]) and Le Morin (France[28]). Other candidates for early dogs include potential burials of juvenile canids

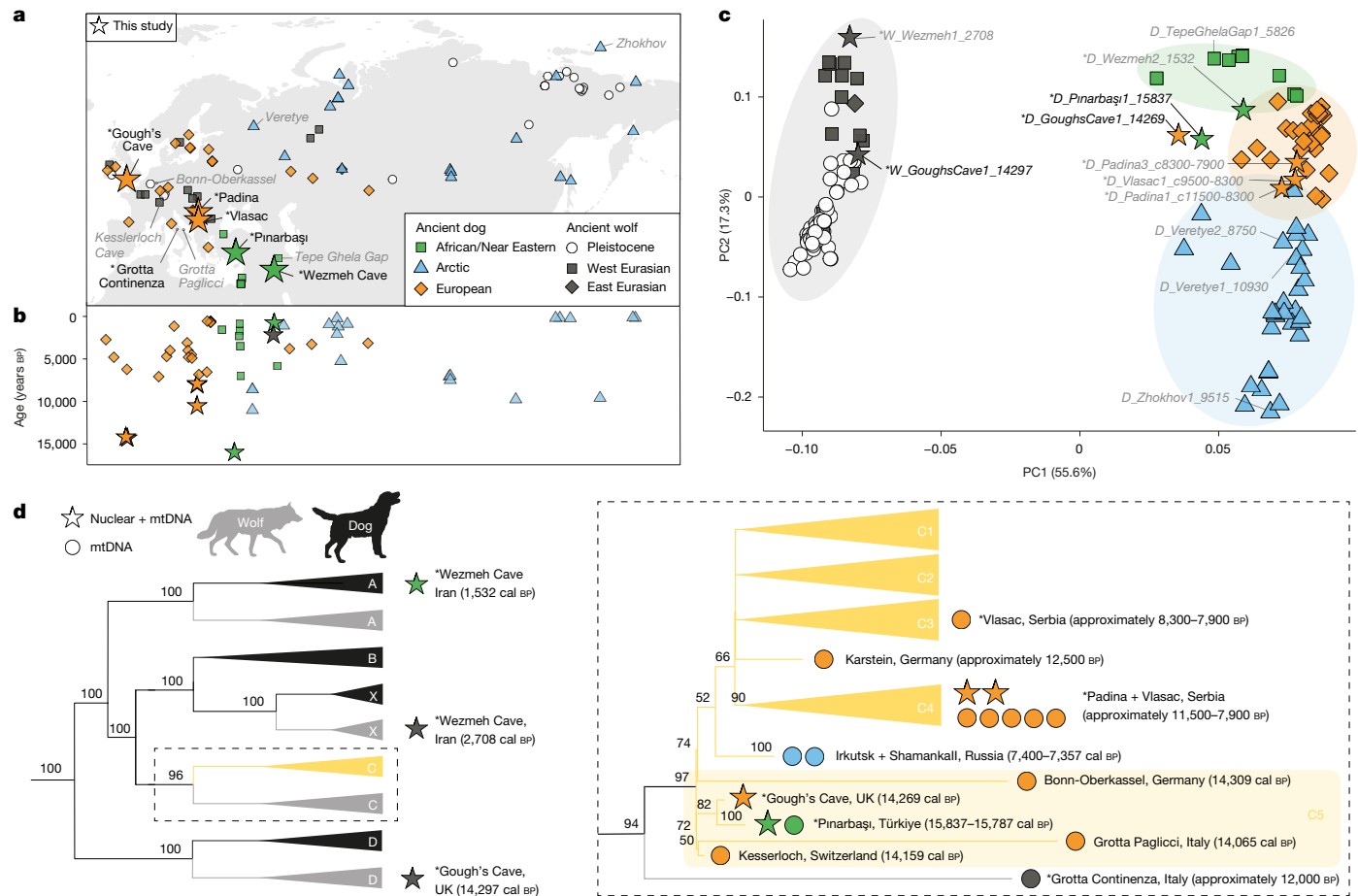

**Fig. 1 | Dogs were widespread in Palaeolithic Europe and Anatolia. a**, Location of the published ancient African and/or Near Eastern (green), European (orange) and Arctic (blue) dog, as well as Pleistocene (white) and Holocene (dark grey) Eurasian wolf genome data analysed in this study ($n = 139$), with unpublished canid specimens shown as stars ($n = 8$). Sites from which new genomic data (mitochondrial or nuclear) were generated are labelled in black text. Frequently mentioned sites from which data were already available are labelled in grey text. **b**, Temporal distribution of ancient Eurasian dog and unpublished wolf nuclear genomes in **a**, based on either direct radiocarbon dates, or securely dated contexts (Supplementary Table 4). **c**, Principal component (PC) analysis

of ancient dogs ($n = 73$) and wolves ($n = 66$). Frequently mentioned samples are labelled in either black (Palaeolithic) or grey (Mesolithic-Byzantine). **d**, Simplified phylogeny based on mitogenomes (see Supplementary Fig. 6 for the complete tree) of 202 dogs (black) and 17 wolves (grey). The C haplogroup is shown in more detail (right panel) to highlight the position of C5 dogs (which included Gough's Cave and Pınarbaşı) as a sister clade to all other C dogs, as well as the position of the Grotta Continenza wolf as a sister lineage to the whole C (including C5). Polytomies represent nodes for which bootstrap support values were less than 50. Across all figures, an asterisk (*) indicates data generated in this study.

at the Epipalaeolithic site of Pınarbaşı (16,100–12,900 years cal BP) on the Central Anatolian Plateau in Türkiye[10] (Supplementary Fig. 1) and canid remains from a Late Upper Palaeolithic Magdalenian horizon (15,100–14,200 years cal BP) at Gough's Cave in the UK[29,30] (Supplementary Fig. 2). Remains at both of these sites show postmortem treatment that mirrors the human remains[15,29].

Here, to test the hypothesis that dogs were present in western Eurasia during the Palaeolithic, we directly dated and generated nuclear genome data (median depth 1.5×, 0.1–3.0×) from canid remains from Gough's Cave (UK, $n = 2$), Pınarbaşı (Türkiye, $n = 1$) and a suspected Palaeolithic deposit from Wezmeh Cave (Iran, $n = 2$). We also generated low-coverage genome data (median depth 0.11×; 0.07–0.17×) from morphometrically identified dogs from Mesolithic contexts in Serbia (Padina; $n = 2$; Vlasac; $n = 1$, refs. 12,13; Fig. 1a,b, Supplementary Tables 1 and 2 and Supplementary Fig. 3). These data were analysed alongside previously published ancient dog ($n = 68$) and wolf ($n = 71$) genomes spanning the past 100,000 years (Supplementary Table 1), as well as 276 modern dog, wolf and outgroup canid genomes (which are a representative subset of the 1,700 genomes included in the NHGRI Dog Genome Project Database; Supplementary Table 3, ref. 31).

In addition, we generated low-coverage sequencing data (less than 0.001×) and used in-solution capture to obtain high-coverage mitochondrial genomes (median depth 15.7×, 5.0–72.3×) from canid remains from Pınarbaşı ($n = 3$), Padina ($n = 2$), Vlasac ($n = 1$) and a Late Upper Palaeolithic canid from Grotta Continenza (Italy, $n = 1$; Supplementary Tables 4 and 5). Finally, to assess the degree to which the dietary isotopes derived from canid remains overlapped with those from Palaeolithic humans (for example, ref. 32), we performed carbon ($\delta^{13}C$) and nitrogen ($\delta^{15}N$) compound-specific stable isotope analysis of individual collagen amino acids of human and canid remains at Gough's Cave ($n = 6$) and Pınarbaşı ($n = 6$; Supplementary Table 6).

## Dogs were widespread during the Palaeolithic

To determine whether these newly sequenced canids were genetically closer to modern and ancient dogs or wolves, we first performed principal component analysis (Fig. 1c and Supplementary Fig. 4) and unsupervised ADMIXTURE ($K = 2$, which differentiates dogs and wolves; Supplementary Fig. 5) analyses based on nuclear genomic data. Of the eight newly sequenced individuals, six clustered with modern and ancient dogs, including an approximately 15,800 year cal BP

individual from Pınarbaşı (1.3×; 15,915–15,669 years cal BP), an approximately 14,300 year cal BP individual from Gough's Cave (1.7×; 14,793–14,090 years cal BP), all three Mesolithic Serbian individuals (Padina (0.1× and 0.2×) and Vlasac (0.1×); 11,500–7,900 years BP), and an approximately 1,532 year cal BP individual from Wezmeh Cave in Iran (0.1×; 1,574–1,419 years cal BP).

We also identified an approximately 14,300 year cal BP wolf from the same context at Gough's Cave (0.2×; 14,808–14,091 years cal BP) and an approximately 2,700 year cal BP wolf from Wezmeh Cave in Iran (3.0×; 2,745–2,542 years cal BP). The discrepancy between contextual and direct age estimates for canids from Wezmeh Cave reflects recent anthropogenic disturbance of the Palaeolithic horizon[33]. All taxonomic classifications were consistent with assignments generated using a pipeline that distinguishes dogs and wolves given ultra-low-coverage sequencing (screening) data[34]. Using the screening data (a few million reads per library), this pipeline identified a further 13 dogs and two wolves at these sites, as well as a wolf from the Palaeolithic site of Grotta Continenza (Extended Data Fig. 1 and Supplementary Table 4).

The definitive nuclear genome-based identification of dogs from Gough's Cave and Pınarbaşı enabled us to evaluate newly generated and publicly available mitochondrial DNA (mtDNA) data to assess the status of other Palaeolithic canids. Dog mtDNA can be divided into five monophyletic haplogroups (A–D and X), each of which are sister to extinct and extant wolf lineages (Fig. 1d and Extended Data Fig. 2). Most of the suspected European Palaeolithic dogs for which mtDNA has previously been generated[20,24,27] branch off just outside the diversity of C haplogroup dogs in previous analyses[35]. This phylogenetic placement, outside the diversity of modern and ancient dogs, yet closer to dogs than wolves, has made it difficult to assess their domestication status.

We constructed a maximum-likelihood phylogeny using 220 mitogenomes (including 202 dogs and 17 wolves) representative of all known dog haplogroups, and a coyote as an outgroup (Fig. 1d and Supplementary Fig. 6). The dogs from Gough's Cave and Pınarbaşı cluster together with other suspected European Palaeolithic dogs, sister to C haplogroup dogs, a haplogroup that we termed C5 (Fig. 1d). This clade also includes canids from Epigravettian sites in Germany (Bonn-Oberkassel, 14,805–14,088 years cal BP, ref. 24), Switzerland (Kesslerloch, 14,318–14,039 years cal BP, ref. 20) and Italy (Grotta Paglicci, 14,310–13,859 years cal BP, ref. 27; Fig. 2a). The Grotta Continenza wolf formed a sister lineage to the diversity of all C haplogroup dogs including C5.

The placement of the Bonn-Oberkassel, Kesslerloch and Grotta Paglicci individuals within a dog-specific mitochondrial clade suggests that they also carry dog nuclear ancestry. This association between dog nuclear ancestry and the C5 mtDNA clade therefore indicates that dogs were widely distributed across western Eurasia during the Late Upper Palaeolithic.

## Genetically similar Palaeolithic dogs

To assess the degree of similarity between the Late Upper Palaeolithic dogs, we first inferred a Bayesian time-calibrated mtDNA phylogeny. The estimated time to the most recent common ancestor of the Gough's Cave and Pınarbaşı individuals was approximately 16,900 years ago (95% highest posterior density, 18,569–15,860 years ago), less than 3,000 years before the death of the Pınarbaşı dog (Extended Data Fig. 2). Furthermore, pairwise distance calculations, outgroup-$f_3$ (Extended Data Figs. 3–5) and kinship ($\theta_{Gough'sCave/Pınarbaşı}$ 0.010, $\theta_{median}$ 0.005; Supplementary Figs. 7 and 8) analysis of autosomal data indicate that dogs from Gough's Cave and Pınarbaşı are more genetically similar to each other than to any other dog. Combined, our mtDNA and nuclear results indicate the Pınarbaşı and Gough's Cave individuals at the eastern and western extremes of the known distribution of

Palaeolithic dogs were genetically very similar, and were members of a population that expanded across western Eurasia between 18,500 years and 14,000 years ago.

Of the five Palaeolithic dogs (Fig. 2a), each is associated with one of three genetically and culturally distinct human hunter-gatherer populations found across Europe and Anatolia in the Late Upper Palaeolithic: the Magdalenian (Gough's Cave[29,30]), the Epigravettian (Bonn-Oberkassel, Kesslerloch and Grotta Paglicci[20,24,27]) and Anatolian hunter-gatherers (Pınarbaşı[10]). The spread of these dogs across the region is likely to have been linked with the migration, dispersal and interaction of humans associated with these Palaeolithic cultures. In fact, although most Palaeolithic dogs in this study are associated with human populations of Epigravettian-associated genetic ancestry, including Anatolian hunter-gatherers at Pınarbaşı[36], the dog from Gough's Cave was recovered from a depositional context alongside humans with Magdalenian-associated ancestry[37].

The presence of genetically similar dogs across sites with genetically differentiated human populations in the Palaeolithic suggests that dogs and humans have had different population histories. To assess this, we first established the degree to which humans and dog ancestries from the same archaeological contexts, across 35 sites that spanned the Late Upper Palaeolithic to the Medieval period, were correlated using outgroup-$f_3$ statistics as a genetic similarity measure (Extended Data Fig. 3c and Supplementary Table 7). After correcting for time and spatial autocorrelation (Methods), we found a strong positive correlation (Mantel $r \approx 0.40$, $P < 0.0001$) between human–human and dog–dog outgroup-$f_3$ values, suggesting shared evolutionary histories between humans and dogs that cannot be explained by shared spatial or temporal structure alone.

In our comparison of the differences between human–human and dog–dog outgroup-$f_3$ at the same site, we observed that the dogs from the Palaeolithic sites of Pınarbaşı and Gough's Cave fell within the tail of the distribution. This indicates that there existed a greater degree of genetic similarity between the dogs than among the associated humans at the same sites (Extended Data Fig. 3d) and provides further evidence that a relatively homogeneous dog population spread between genetically and culturally distinct human populations across Europe and Anatolia in the Late Upper Palaeolithic.

One plausible scenario to explain the discrepancy between human and dog ancestry in the Palaeolithic is that the dispersal of dogs was coupled with the spread of Epigravettian-associated ancestry and material culture roughly 16,000 years ago, which, after a period of interaction, eventually replaced the previously predominant Magdalenian culture and ancestry across northern Europe[14,16,37]. This timeframe aligns well with our mtDNA-based most recent common ancestor estimate for the Gough's Cave and Pınarbaşı dogs (95% highest posterior density, 18,569–15,860 years ago), which provides an estimated upper bound for the divergence of their ancestral populations (which must have taken place after approximately 18,500 years ago), and the youngest radiocarbon date (Gough's Cave dog, 14,808–14,091 years cal BP; Supplementary Table 4), which provide a minimum age by which their ancestral population must have already diverged. Crucially, the timeframe for the spread of dogs in western Eurasia (between 18,500–14,000 years ago) postdates the earlier divergence of Magdalenian and Epigravettian human populations, which most probably occurred during or before the Last Glacial Maximum (approximately 24,000–21,000 years ago[14,38]). The spread of Palaeolithic dogs across western Eurasia, therefore, most probably occurred after the divergence of these two distinct human populations, possibly during the spread of Eppigravettian-associated ancestry and material culture roughly 16,000 years ago.

The Gough's Cave dog, however, dates to the Late Magdalenian (from 15,000 years ago; Fig. 2b), a period for which there is at present no evidence of humans carrying Epigravettian-associated ancestry in the UK[37]. Yet, this period is characterized by transitions in lithic technology (for example, *Federmessergroupen*) that are associated with people

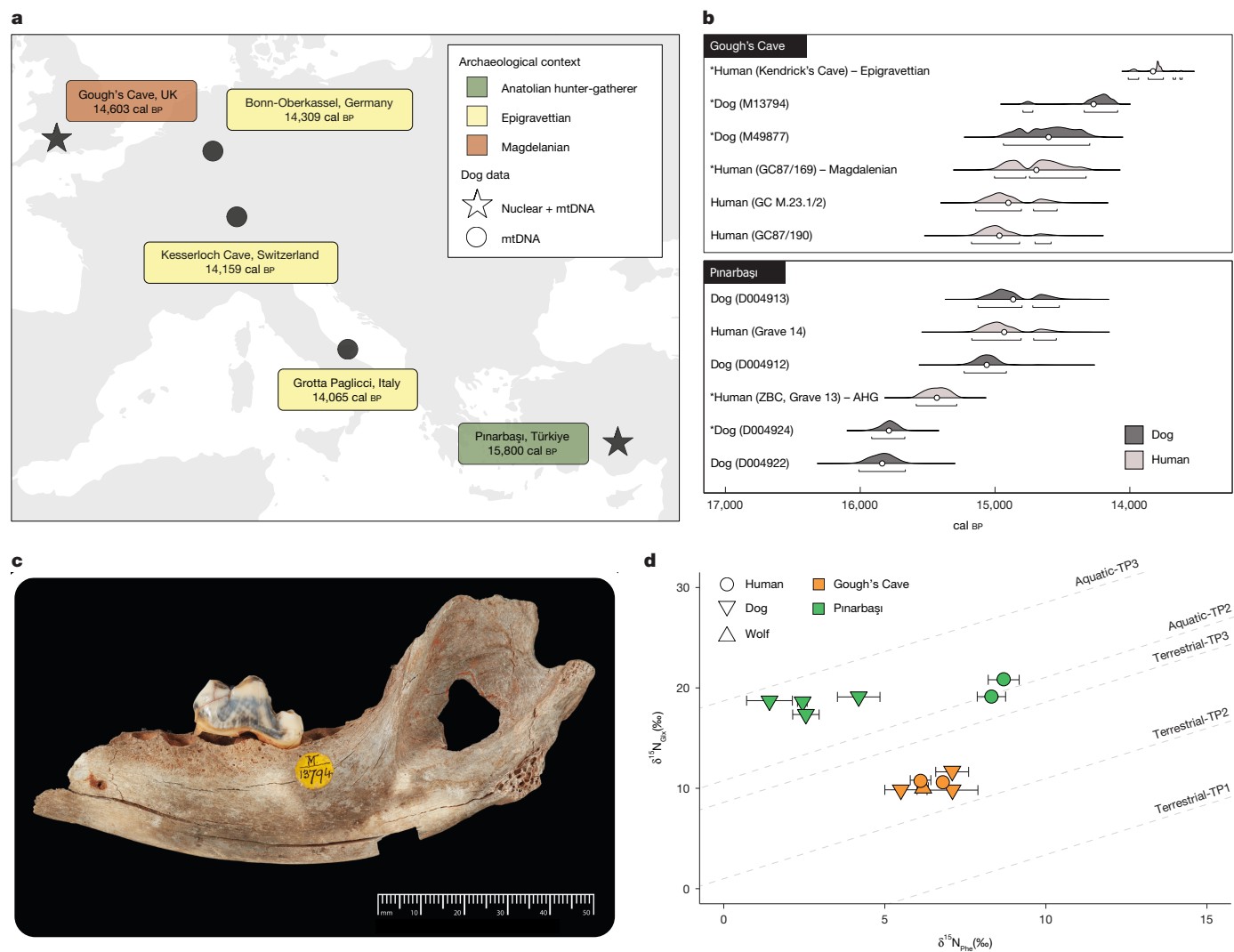

**Fig. 2 | Close human–dog associations during the Palaeolithic. a**, Earliest directly dated dogs from Late Upper Palaeolithic archaeological contexts across western Eurasia based on both mitochondrial (circle) and nuclear (star) DNA. **b**, Calibrated radiocarbon dates of canid and human remains at both Gough's Cave and Pınarbaşı. The genetic ancestry of human remains, Anatolian hunter-gatherer (AHG), Epigravettian-associated and Magdalenian-associated, are indicated for those with publicly available nuclear genomes[36,37]. Dates were calibrated using OxCal v.4.4 and the IntCal20 calibration curve, with mean age (circles) and 95% confidence intervals reported[69,70]. **c**, The Gough's Cave dog (NHMUK PV M13794a) mandible in lateral view. Anthropic piercing of the

masseteric fossa can be seen. **d**, $\delta^{15}N$ values of glutamic acid (Glx) and phenylalanine (Phe) of bone collagen from human and canid bones. To infer trophic positions (TP), measured values are compared against isotopic thresholds (dashed lines) derived from trophic discrimination factors (TDF) and beta values, calibrated using terrestrial (vascular) and aquatic (non-vascular) autotrophs[71]. The interpretation of the Pınarbaşı neonatal canids assumes a minimal offset in $_{Phe}$ and $_{Glx}$ $\delta^{15}N$ values between mothers and their foetal offspring[54]. Photograph in **c** reproduced with permission from The Trustees of the Natural History Museum, London.

carrying Epigravettian-related ancestry on the European continent (for example, at Bonn-Oberkassel)[39–42], whereas later individuals from the UK (for example, Kendrick's Cave, 13,780–13,354 years cal BP) carry Epigravettian-associated ancestry[14,37]. Further, the putative presence of dogs (based on short mtDNA fragments) has also been suggested at another Magdalenian context, in Spain (Erralla Cave, 17,410–17,096 years cal BP, ref. 26) and, unlike at Gough's Cave, these remains postdate the earliest evidence of Epigravettian ancestry in the region by more than 1,000 years (El Mirón, approximately 18,700 years cal BP, refs. 43,44).

Combined, our results are consistent with a dispersal of dogs across Late Palaeolithic Europe alongside the expansion of Epigravettian-associated ancestry and material culture approximately 16,000 years ago. Under this scenario, people carrying Magdalenian-associated ancestry in the UK, and perhaps in Spain, acquired dogs through interactions with Epigravettians. These interactions did not leave any

signal of Epigravettian ancestry in the Magdalenian humans at Gough's Cave[37], implying that the exchange of dogs between Palaeolithic human groups was not always accompanied by detectable gene flow within the human populations. However, an alternative scenario that involves dogs being exchanged through networks that are not associated with the Epigravettian expansion cannot at present be ruled out due to the paucity of Palaeolithic dog remains.

## Close Palaeolithic human–dog association

The discovery of dogs within Magdalenian, Epigravetian and Anatolian hunter-gatherer contexts across western Eurasia indicates dogs were integrated with culturally, geographically and genetically distinct human groups. At Gough's Cave, human remains in a Magdalenian context show postmortem anthropic modification including the shaping of skulls into skull-cups and the engraving of human remains[15,45].

This is indicative of funerary cannibalism, a behaviour identified at Magdalenian sites across Europe[15,16]. Similar postmortem anthropic modification is evident on the dog remains at Gough's Cave, most notably a perforation on the masseteric fossa (Fig. 2c and Supplementary Fig. 2), indicating the shared treatment practices of humans and dogs postmortem. Similar shared postmortem treatment of dog and human remains is also evident at Pınarbaşı, where neonatal and juvenile dogs were buried in the same area of the site as contemporaneous human burials[10] (Fig. 2b). These shared patterns of postmortem treatment extend the symbolic treatment of dogs identified during the Mesolithic to earlier Palaeolithic hunter-gatherers[46,47].

To investigate whether similar close associations were also evident during the lifetime of the dogs, we evaluated the degree of dietary similarity between canids and humans at both Gough's Cave and Pınarbaşı through the measurement of bulk and amino acid stable isotopes ($\delta^{13}C$ and $\delta^{15}N$) in bone collagen[48] (Fig. 2d, Supplementary Fig. 9 and Supplementary Table 6).

At Gough's Cave, the dogs and humans have comparable bulk isotope and amino acid $\delta^{15}N$ values, and the $\delta^{15}N_{Glx-Phe}$ proxy for trophic position (Fig. 2d) indicates a similar degree of omnivory[49,50]. The Gough's Cave wolf occupied a similar trophic position to both these species (Fig. 2d). Niche partitioning between dogs and wolves (for example, ref. 51) may have, therefore, been less pronounced in Late Upper Palaeolithic Europe compared with later periods, notably after the emergence of starch-rich diets in the Neolithic[52] (Supplementary Fig. 10). We note, however, that the isotopic approach lacks the resolution to definitively discriminate between wild and domestic canids, as different diets may result in identical isotopic signatures[53]. Alternatively, the dietary overlap between wolves and humans at Gough's Cave could also indicate close association, although this remains tentative given the lack of comparable isotopic data from wolves outside anthropogenic contexts in the Late Upper Palaeolithic.

At Pınarbaşı, although perinatal dogs (and by proxy their mothers) had different isotopic signatures to the humans (Fig. 2d), the $\delta^{15}N_{Glx-Phe}$ values of both species were elevated relative to Gough's Cave. Notwithstanding uncertainties in the mother–offspring isotopic offset[54], these data indicate an aquatic dietary component (Fig. 2d and Supplementary Table 6). The remains of small freshwater fish, probably net-caught, are common in the human-occupied layers at Pınarbaşı[10], suggesting that dogs were either being directly or indirectly provisioned by humans.

## Genetic ancestry of Palaeolithic dogs

Dogs can be broadly separated into two lineages: an eastern lineage, found in Arctic, East Asian and precontact American dogs (which mostly disappeared after CE 1492, ref. 55), and a western lineage that encompasses dogs from Europe and the Near East[56]. The presence of both eastern and western ancestries in Karelian Mesolithic dogs approximately 10,900 years ago[8] implies that the two ancestries diverged during the Palaeolithic. To establish the ancestry of Palaeolithic and Mesolithic European dogs, we calculated shared drift with representatives of eastern (an approximately 9,500 years cal BP Siberian hunter-gatherer dog from Zhokhov Island[57]) and western (an approximately 5,800 years cal BP dog from the Neolithic Iranian site of Tepe Ghela Gap[58]) dog lineages using outgroup-$f_3$ statistics of the form $f_3$(Coyote, $D\_TepeGhelaGap1\_5826/D\_Zhokhov1\_9515$, $X$).

Palaeolithic dogs from Europe and Anatolia share more drift with the western Eurasian lineage (Extended Data Figs. 4 and 5), which is also apparent from principal component analysis (Fig. 1c and Supplementary Fig. 4), and $D$ statistics of the form $D$(Coyote, $X$, $D\_Zhokhov1\_9515$, $D\_TepeGhelaGap1\_5826$) (Supplementary Fig. 11). These results indicate that Palaeolithic dogs form part of the western Eurasian dog lineage, which pushes the divergence of eastern–western dog populations to at least 15,800 years ago.

## Wolf–dog admixture in the ancient Near East

A recent genomic study[22] indicated that both modern and ancient western Eurasian dogs possess genetic ancestry similar to present-day wolves in Syria. To assess whether newly sequenced western Eurasian Palaeolithic and Mesolithic dogs also possess Near Eastern wolf ancestry, we performed $D$ statistics of the form $D$(Coyote, Near Eastern Wolf, Early Western Eurasian Dogs, $D\_Zhokhov1\_9515$). We used a newly sequenced genome (3×) from an approximately 2,700-year-old wolf from Iran (Wezmeh Cave; Fig. 1a) as a proxy for Near Eastern wolf ancestry, because some present-day Near Eastern wolf populations (for example, Israel and Saudi Arabia) probably acquired dog ancestry through hybridization more recently (Extended Data Fig. 6a).

We identified excess allele sharing between ancient Near Eastern dogs (including the Palaeolithic dog from Pınarbaşı) and the Wezmeh wolf, consistent with gene flow between dogs and the local Near Eastern wolf population as far back as 15,800 years ago (Extended Data Fig. 6a and Supplementary Fig. 12). We found no evidence of excess allele sharing with Mesolithic dogs from Veretye (Karelia, north-west Russia; approximately 10,900 years cal BP) or Padina (Serbia; 11,500–7,900 years BP). However, tests were significant for a Mesolithic dog from Vlasac (Serbia; 9,500–8,300 years BP). These results indicate that Near Eastern wolf ancestry was variable in dogs across western Eurasia during the Palaeolithic and Mesolithic.

In the Near East, dogs from the Neolithic and onwards show excess allele sharing with Near Eastern wolves when compared with contemporary European dogs (Extended Data Fig. 6a). $D$ statistics of the form $D$(Coyote, Near Eastern Wolf, $X$, $D\_Pinarbasi1\_15787$) were significantly negative in most cases, suggesting that these Near East dogs possess more Near Eastern wolf ancestry than the Pınarbaşı dog (Supplementary Fig. 13). To quantify the extent of excess Near Eastern wolf ancestry in these Neolithic and post-Neolithic populations, we computed $F_4$-ratios (Extended Data Fig. 6b) and performed supervised ADMIXTURE analysis (Supplementary Fig. 14). These analyses identified the highest levels of wolf ancestry ($F_4$ ratio 19.0%, ADMIXTURE 13.5%) in a 7,000-year-old dog from Tel Hreiz (Israel; Supplementary Fig. 14). This component declined over the subsequent millennia to less than 5% ($F_4$ ratio 2.9–4.8%, ADMIXTURE 3.2–4.8%) in 2,300-year-old dogs from Ashkelon (Israel).

By contrast, high levels of wolf ancestry are maintained in modern Basenjis ($F_4$ ratio 13.9–17.4%; Extended Data Fig. 6b), possibly due to the isolation of this dog population in sub-Saharan Africa until the colonial period. A previous study suggested that dogs in Africa experienced gene flow with endemic African canids[59]. It is possible, therefore, that Near Eastern wolf ancestry identified in Basenjis may derive from gene flow with these endemic canids after dogs dispersed into Africa.

To visualize both the direction and number of these admixture events, we constructed admixture graphs using TreeMix[60] (Extended Data Fig. 7 and Supplementary Figs. 14 and 15) and AdmixtureBayes[61] (Fig. 3a and Supplementary Fig. 16). These results further support the conclusion that ancient Near Eastern (excluding Pınarbaşı) and African dogs possess mixed ancestry. Specifically, the two primary ancestry components are 96% western Eurasian dogs (represented by dogs from Pınarbaşı and Gough's Cave) and 4% Near Eastern wolves (represented by the Wezmeh wolf).

Overall, our results indicate that Neolithic (and more recent) Near Eastern and African dogs possess more Near Eastern wolf ancestry than Palaeolithic and Mesolithic western Eurasian dogs. Although it remains possible that this wolf-related ancestry was the result of an independent domestication process[22,56], it is more likely to result from geographically and temporally restricted gene flow between Neolithic and post-Neolithic western Eurasian dogs and Near Eastern wolves.

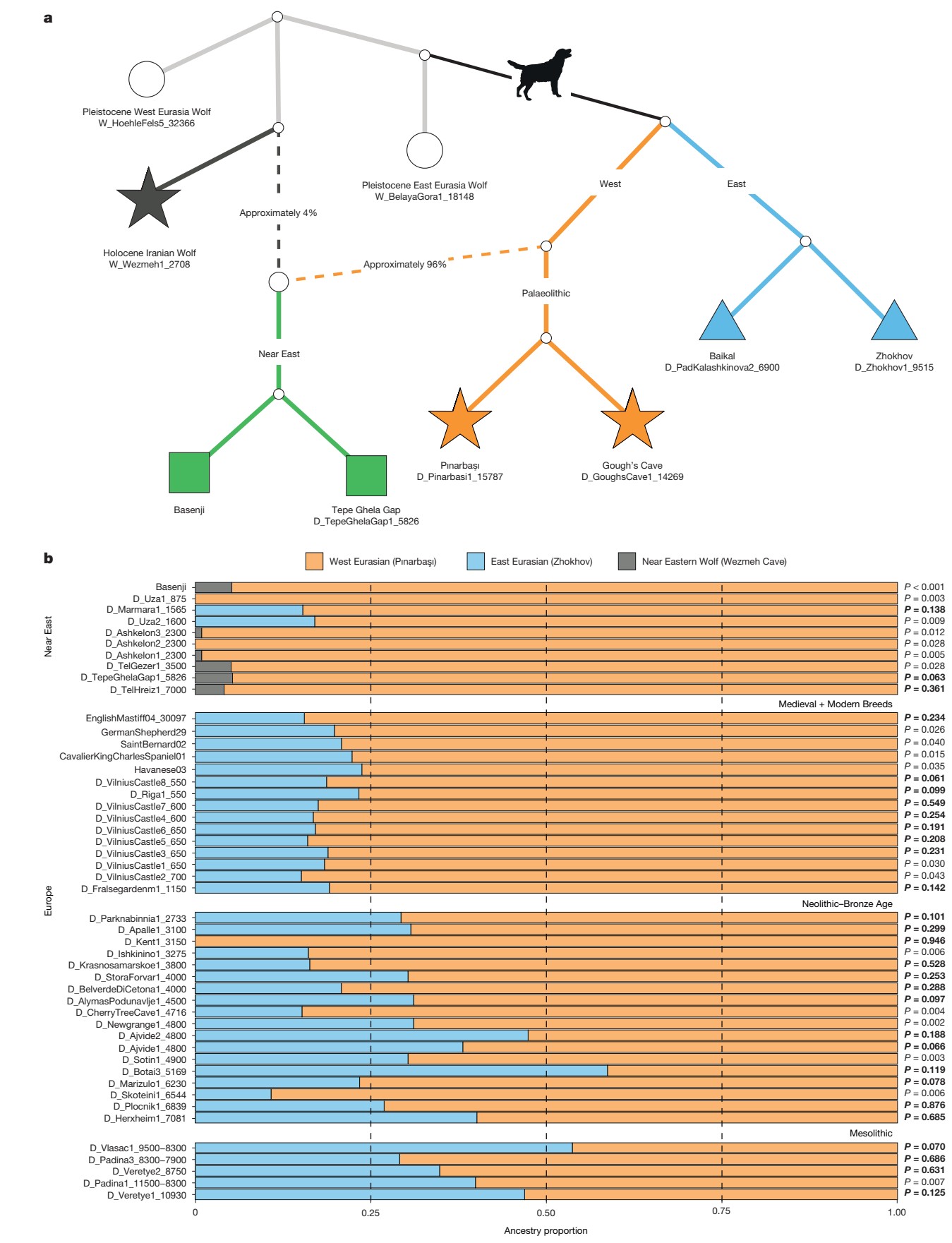

**Fig. 3** | See next page for caption.

**Fig. 3 | Ancestry of European dogs from Mesolithic to today. a**, Admixture graph generated using AdmixtureBayes, allowing for a single admixture event ($k = 1$), which showed the highest posterior support (0.77) of all models (see Supplementary Fig. 16 for further admixture events). Mean ancestry proportions were estimated in AdmixtureBayes using coyotes as an outgroup. **b**, Proportion of Palaeolithic western Eurasian (orange; *D_Pinarbasi1_15787*), eastern Eurasian (blue; *D_Zhokhov1_9515*) and Near Eastern wolf (grey; *W_Wezmeh1_2708*) ancestries in modern and ancient Near Eastern (top) and European (bottom) dogs, estimated using qpAdm. Models were not rejected if their *P* value exceeded a threshold of 0.01 (those exceeding 0.05 are shown in bold). European dogs are separated into key time periods (Supplementary Fig. 21). The Gough's Cave dog (*D_GoughsCave1_14269*), which was modelled as a mixture of Palaeolithic western Eurasian (82.4%) and Near Eastern wolf (17.6%) ancestries ($P < 0.01$), was excluded as the source of basal ancestry could not be identified (Supplementary Information).

## Genomic legacy of Palaeolithic dogs

Our analyses indicate that both the Gough's Cave and Pınarbaşı dogs derive most of their ancestry from the western Eurasian dog lineage, similar to Neolithic (and later Near Eastern) dogs. Previous work has shown that from the Late Neolithic onwards, European dogs possess mixed ancestry from both western and eastern Eurasian ancestry[22,56,58,62]. This shift in ancestry proportions has been attributed to an influx of dog ancestry from the East during migration of Steppe pastoralists associated with the Corded Ware and Yamnaya cultures of the Late Neolithic and Early Bronze Age[62].

Nevertheless, ancient human genomes have indicated earlier migrations (before 9,000 years ago) of people with eastern hunter-gatherer genetic ancestry[14] into eastern, central and southern Europe. Analyses of dogs from eastern hunter-gatherer contexts from north-eastern Europe (Veretye, Karelia) and the central Eurasian Steppe (Botai, Kazakhstan) have shown these individuals carried eastern Eurasian dog ancestry[63]. These results indicate that eastern Eurasian dog ancestry may have reached other parts of Europe earlier than previously thought, possibly during the Mesolithic.

To assess the degree of eastern Eurasian dog ancestry in western Eurasian dogs, we implemented a tri-source model of dog ancestry in qpAdm (Fig. 3b and Supplementary Fig. 17), using three ancestry sources based on the results of *D* statistics and admixture graphs: Palaeolithic western Eurasian (*D_Pinarbasi1_15787*) and eastern Eurasian (*D_Zhokhov1_9515*) dogs and Near Eastern wolves (*W_Wezmeh1_2708*). As expected, most ancient Near Eastern dogs possessed both western Eurasian dog and Near Eastern wolf ancestry. The eastern Eurasian dog ancestry component is absent until the Early Byzantine period (1,565 years cal BP at Marmara, Türkiye). By contrast, Balkan Mesolithic dogs from the Iron Gates in Serbia (Padina and Vlasac) and North-West Russia (Veretye) were best modelled as a combination of western Eurasian (mean 56.2%) and eastern Eurasian (mean 43.8%) dog ancestry.

Genomic analyses of Mesolithic hunter-gatherers from the same contexts as several of these dogs (Iron Gates and Veretye) have shown that they also possessed eastern hunter-gatherer genetic ancestry[14,64]. Coupled with the positive correlation between human–dog ancestries (Extended Data Fig. 3c), our results indicate that eastern Eurasian dog ancestry may have been introduced into Europe during the spread of eastern hunter-gatherers during the Mesolithic[14], rather than during the later Steppe pastoralist migrations, as previously proposed[62].

Our qpAdm analysis also indicates that the eastern Eurasian dog ancestry persisted in European dog populations from the Mesolithic to the present. Roughly 30% of this eastern Eurasian ancestry component is present in dogs during the Neolithic (for example, Pločnik, Serbia), 33% during the Bronze Age (for example, Belverde Di Cetona, Italy), 22% during the Iron Age (for example, Parknabinnia, Ireland), 18% during the Medieval period (for example, Vilnius Castle, Lithuania) and roughly 20% in modern breed dogs (Fig. 3b and Supplementary Fig. 17). Although this does not preclude further influxes of eastern Eurasian dog ancestry into Europe post-Mesolithic[58,62], these findings show that the fundamental ancestry components (both eastern and western Eurasian ancestry) in European dogs were established by at least 10,900 years ago and persisted into modern breeds (Supplementary Fig. 17).

## Conclusions

Our results provide genomic evidence for the presence of genetically similar dogs in the UK, Germany, Italy, Switzerland and Türkiye during the Late Upper Palaeolithic (between 15,800–14,200 years cal BP; Figs. 1 and 2). The ancestry of this Palaeolithic population was retained in dogs throughout the Holocene (Fig. 3) and into modern breeds. We propose that this early dog population first spread across Late Upper Palaeolithic Europe alongside the expansion of Epigravettian-associated ancestry and material culture roughly 16,000 years ago. The presence of dogs in close association with humans carrying unadmixed Magdalenian-associated genetic ancestry (for example, at Gough's Cave), however, indicates exchange of dogs between culturally distinct human populations during the Late Upper Palaeolithic in the absence of widespread gene flow or population turnover.

The Late Upper Palaeolithic dog population seems to have already been largely reproductively isolated from wolves. In fact, despite more than 15,000 years of the co-occurrence of dogs and wolves in western Eurasia, dogs have acquired little to no wolf ancestry aside from isolated instance(s) of admixture with wolves in the Near East that took place before 7,000 years ago. This contrasts with patterns of introgression seen in other domestic species, such as pigs and cattle, which interbred extensively with local populations of wild boar and aurochs following their introduction to Europe[65–67]. The near absence of wolf ancestry in ancient and modern dog genomes[68] indicates that a substantial barrier to gene flow between wolves and dogs had been established in Europe and Anatolia by the Late Upper Palaeolithic. Combining many lines of evidence (for example, DNA, isotopes, material culture) that span the past 20,000 years across Eurasia will help to establish the precise phenotypic and cultural mechanisms responsible for the emergence of dogs, and their subsequent reproductive isolation from wolves.

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

¹Centre for Human Evolution Research, Natural History Museum, London, UK. ²Chair of Animal Systems Genomics, Faculty of Veterinary Medicine, Ludwig-Maximilians-Universität, Munich, Germany. ³Palaeogenomics and Bio-Archaeology Research Network, School of Archaeology, University of Oxford, Oxford, UK. ⁴Department of Biological Sciences, Middle East Technical University, Ankara, Turkey. ⁵BioArCh, Department of Archaeology, University of York, York, UK. ⁶Richard Gilder Graduate School, American Museum of Natural History, New York, NY, USA. ⁷Department of Anthropology, National Museum of Natural History, Smithsonian Institution, Washington, DC, USA. ⁸Yong Loo Lin School of Medicine, National University of Singapore, Singapore, Singapore. ⁹Institute for Human Development and Potential, A*STAR, Singapore, Singapore. ¹⁰Department of Genetics, Evolution and Environment, University College London, London, UK. ¹¹School of Biological Sciences, University of East Anglia, Norwich, UK. ¹²BioArchéologie, Interactions Sociétés Environnements, Muséum National d'Histoire

Naturelle, Paris, France. [13]Histoire Naturelle des Humanités Préhistoriques, Muséum National d'Histoire Naturelle, Paris, France. [14]Bioarchaeology Laboratory, Central Laboratory, University of Tehran, Tehran, Iran. [15]University of Georgia, Athens, GA, USA. [16]Paleolithic Department, National Museum of Iran, Tehran, Iran. [17]Dipartimento di Biologia Ambientale, SAPIENZA Università di Roma, Rome, Italy. [18]Section for Molecular Ecology and Evolution, Globe Institute, University of Copenhagen, Copenhagen, Denmark. [19]Department of Evolutionary Anthropology, Faculty of Life Sciences, University of Vienna, Vienna, Austria. [20]Human Evolution and Archaeological Sciences (HEAS) Network, University of Vienna, Vienna, Austria. [21]Centre de Recherche sur la Biodiversité et l'Environnement, Université de Toulouse, Toulouse INP, CNRS, IRD, CRBE, Toulouse, France. [22]Centre for Palaeogenetics, Stockholm University, Stockholm, Sweden. [23]Department of Geological Sciences, Stockholm University, Stockholm, Sweden. [24]Department of Archaeology and Heritage Studies, Aarhus University, Aarhus, Denmark. [25]Department of Archaeology, Classics and Egyptology, University of Liverpool, Liverpool, UK. [26]Haci Bayram Veli University, Ankara, Turkey. [27]Chronicle Heritage, Phoenix, AZ, USA. [28]Institute of Palaeoanatomy, Domestication Research and the History of Veterinary Medicine, Ludwig-Maximilan-University Munich, Munich, Germany. [29]SNSB-State Collection of Palaeoanatomy Munich, Munich, Germany. [30]Simon Fraser University, Burnaby, British Columbia, Canada. [31]Department of Prehistory, Division of Archaeology, Faculty of Arts and Sciences, Hitit University, Çorum, Turkey. [32]Ancient Genomics Laboratory, The Francis Crick Institute, London, UK. [33]Institute of Archaeology, University College London, London, UK. [34]Laboratory for Bioarchaeology, Department of Archaeology, Faculty of Philosophy, University of Belgrade, Belgrade, Serbia. [35]School of Biological and Behavioural Sciences, Queen Mary University of London, London, UK. [36]These authors contributed equally: William A. Marsh, Lachie Scarsbrook. [37]These authors jointly supervised this work: Greger Larson, Ian Barnes, Laurent A. F. Frantz. [✉]e-mail: w.marsh@nhm.ac.uk; lachiescarsbrook@gmail.com; greger.larson@arch.ox.ac.uk; I.Barnes@nhm.ac.uk; laurent.frantz@lmu.de

# Reporting Summary

## Statistics

For all statistical analyses, confirm that the following items are present in the figure legend, table legend, main text, or Methods section.

| n/a | Confirmed | |
|---|---|---|
| ☐ | ☒ | The exact sample size (*n*) for each experimental group/condition, given as a discrete number and unit of measurement |
| ☐ | ☒ | A statement on whether measurements were taken from distinct samples or whether the same sample was measured repeatedly |
| ☐ | ☒ | The statistical test(s) used AND whether they are one- or two-sided *Only common tests should be described solely by name; describe more complex techniques in the Methods section.* |
| ☐ | ☒ | A description of all covariates tested |
| ☐ | ☒ | A description of any assumptions or corrections, such as tests of normality and adjustment for multiple comparisons |
| ☐ | ☒ | A full description of the statistical parameters including central tendency (e.g. means) or other basic estimates (e.g. regression coefficient) AND variation (e.g. standard deviation) or associated estimates of uncertainty (e.g. confidence intervals) |
| ☐ | ☒ | For null hypothesis testing, the test statistic (e.g. *F*, *t*, *r*) with confidence intervals, effect sizes, degrees of freedom and *P* value noted *Give P values as exact values whenever suitable.* |
| ☐ | ☒ | For Bayesian analysis, information on the choice of priors and Markov chain Monte Carlo settings |
| ☒ | ☐ | For hierarchical and complex designs, identification of the appropriate level for tests and full reporting of outcomes |
| ☐ | ☒ | Estimates of effect sizes (e.g. Cohen's *d*, Pearson's *r*), indicating how they were calculated |

*Our web collection on statistics for biologists contains articles on many of the points above.*

## Software and code

Policy information about availability of computer code

| Data collection | *Provide a description of all commercial, open source and custom code used to collect the data in this study, specifying the version used OR state that no software was used.* |
|---|---|
| Data analysis | Radiocarbon Analyis: Oxcal v.4.4<br>Isotopic Analysis: R Studio<br>Ancient DNA Analysis: nf-core/eager v.2.4.6, CanID v1.0, bcftools, ANSGD, samtools v.1.16.1, MAFFT v7.505, IQ-TREE v.2.1.4, BEAST2 v.2.6.7, Tracer v.2.6.7, LogCombiner v.2.6.7, EIGENSOFT v.8.0.0, EMU v.0.9, struct-f4, ADMIXTURE v.1.3.0, READv2, GLIMPSE v1.1.1, PLINK v.1.90b6.21,  AdmixTools v.7.0.2, TreeMix v.1.13, AdmixtureBayes, R Studio |

For manuscripts utilizing custom algorithms or software that are central to the research but not yet described in published literature, software must be made available to editors and reviewers. We strongly encourage code deposition in a community repository (e.g. GitHub). See the Nature Portfolio guidelines for submitting code & software for further information.

## Data

Policy information about <u>availability of data</u>

All manuscripts must include a <u>data availability statement</u>. This statement should provide the following information, where applicable:
- Accession codes, unique identifiers, or web links for publicly available datasets
- A description of any restrictions on data availability
- For clinical datasets or third party data, please ensure that the statement adheres to our <u>policy</u>

All newly generated ancient sequences (SAMEA120632851–SAMEA120632875) are available in the ENA under BioProject PRJEB104454 (https://www.ebi.ac.uk/ena/browser/view/PRJEB104454). Raw radiocarbon age estimates are reported in Supplementary Dataset 2. Raw isotopic measurements are reported in Supplementary Dataset 6. All other data are included in the manuscript and/or supporting information.

## Research involving human participants, their data, or biological material

Policy information about studies with <u>human participants or human data</u>. See also policy information about <u>sex, gender (identity/presentation), and sexual orientation</u> and <u>race, ethnicity and racism</u>.

| | |
|---|---|
| Reporting on sex and gender | *Use the terms sex (biological attribute) and gender (shaped by social and cultural circumstances) carefully in order to avoid confusing both terms. Indicate if findings apply to only one sex or gender; describe whether sex and gender were considered in study design; whether sex and/or gender was determined based on self-reporting or assigned and methods used. Provide in the source data disaggregated sex and gender data, where this information has been collected, and if consent has been obtained for sharing of individual-level data; provide overall numbers in this Reporting Summary. Please state if this information has not been collected. Report sex- and gender-based analyses where performed, justify reasons for lack of sex- and gender-based analysis.* |
| Reporting on race, ethnicity, or other socially relevant groupings | *Please specify the socially constructed or socially relevant categorization variable(s) used in your manuscript and explain why they were used. Please note that such variables should not be used as proxies for other socially constructed/relevant variables (for example, race or ethnicity should not be used as a proxy for socioeconomic status). Provide clear definitions of the relevant terms used, how they were provided (by the participants/respondents, the researchers, or third parties), and the method(s) used to classify people into the different categories (e.g. self-report, census or administrative data, social media data, etc.) Please provide details about how you controlled for confounding variables in your analyses.* |
| Population characteristics | *Describe the covariate-relevant population characteristics of the human research participants (e.g. age, genotypic information, past and current diagnosis and treatment categories). If you filled out the behavioural & social sciences study design questions and have nothing to add here, write "See above."* |
| Recruitment | *Describe how participants were recruited. Outline any potential self-selection bias or other biases that may be present and how these are likely to impact results.* |
| Ethics oversight | *Identify the organization(s) that approved the study protocol.* |

Note that full information on the approval of the study protocol must also be provided in the manuscript.

# Field-specific reporting

Please select the one below that is the best fit for your research. If you are not sure, read the appropriate sections before making your selection.

☐ Life sciences ☐ Behavioural & social sciences ☒ Ecological, evolutionary & environmental sciences

For a reference copy of the document with all sections, see <u>nature.com/documents/nr-reporting-summary-flat.pdf</u>

# Ecological, evolutionary & environmental sciences study design

All studies must disclose on these points even when the disclosure is negative.

| | |
|---|---|
| Study description | Analysis of suspected Palaeolithic and Mesolithic dog genomes from across Western Eurasia. |
| Research sample | Ancient dog (Canis lupus familiaris; n = 22) and wolf (Canis lupus lupus; n = 5) remains were sampled from across Western Eurasia including the UK, Serbia, Italy, Türkiye and Iran. Detailed information including site and specimen descriptions can be found in the Supplementary Methods. |
| Sampling strategy | All individuals which possessed enough bone/tooth material were sampled for this study (see Supplementary Methods). Material was obtained through collaboration with archaeologists working in the respective regions (see author list). |
| Data collection | DNA was extracted, and double-stranded dual-indexed libraries were prepared from samples at either: Ancient DNA Laboratory, Natural History Museum, London; Department of Veterinary Medicine, Ludwig Maximilian University, Munich; or Research Lab for |

| | Archaeology and the History of Art, University of Oxford, Oxford. |
|---|---|
| Timing and spatial scale | Samples ranged in age from 1532 to 15837 years calBP. |
| Data exclusions | No data was excluded from the analysis. |
| Reproducibility | Detailed description of all experimental work is available in the supplementary information. All institutional/archaeological codes from which samples were taken are also detailed. No-template ("blank") controls were processed alongside all samples. These samples failed to amplify efficiently and no reads sequenced from no-template libraries could be unambiguously mapped, indicating that background and cross-contamination were minimal. |
| Randomization | No experiment requiring randomization was conducted in this study. |
| Blinding | No experiment requiring blinding was conducted in this study. |

Did the study involve field work?  ☐ Yes  ☒ No

# Reporting for specific materials, systems and methods

We require information from authors about some types of materials, experimental systems and methods used in many studies. Here, indicate whether each material, system or method listed is relevant to your study. If you are not sure if a list item applies to your research, read the appropriate section before selecting a response.

## Materials & experimental systems

| n/a | Involved in the study |
|---|---|
| ☒ | Antibodies |
| ☒ | Eukaryotic cell lines |
| ☐ ☒ | Palaeontology and archaeology |
| ☒ | Animals and other organisms |
| ☒ | Clinical data |
| ☒ | Dual use research of concern |
| ☒ | Plants |

## Methods

| n/a | Involved in the study |
|---|---|
| ☒ | ChIP-seq |
| ☒ | Flow cytometry |
| ☒ | MRI-based neuroimaging |

## Palaeontology and Archaeology

| Specimen provenance | M13794, Goughs Cave, UK (NHM)<br>M49877, Goughs Cave, UK (NHM)<br>M13795, Goughs Cave, UK (NHM)<br>M50014, Goughs Cave, UK (NHM)<br>M50015, Goughs Cave, UK (NHM)<br>M13796, Goughs Cave, UK (NHM)<br>M50013, Goughs Cave, UK (NHM)<br>AL2934, Grotta Continenza, Italy (Oxford)<br>OL4117, Padina, Serbia (Oxford)<br>OL4090 / SC1178, Padina, Serbia (Oxford)<br>OL4109 / SC1182, Padina, Serbia (Oxford)<br>OL4118 / LS0288, Padina, Serbia (Oxford)<br>OL4098 / SC1180, Padina, Serbia (Oxford)<br>D004924, Pinarbasi, Türkiye (LMU)<br>D004924, Pinarbasi, Türkiye (LMU)<br>D004924, Pinarbasi, Türkiye (LMU)<br>AL2921/ D004922, Pinarbasi, Türkiye (LMU)<br>AL2918 / D004912, Pinarbasi, Türkiye (LMU)<br>AL2919 / D004913, Pinarbasi, Türkiye (LMU)<br>AL2884, Vlasac, Serbia (Oxford)<br>OL4071, Vlasac, Serbia (Oxford)<br>OL4073 / SC1188, Vlasac, Serbia (Oxford)<br>WZ77, Wezmeh, Iran (LMU)<br>WZ189, Wezmeh, Iran (LMU)<br>WZ194, Wezmeh, Iran (LMU)<br>WZ190, Wezmeh, Iran (LMU) |
|---|---|
| Specimen deposition | Specimens from: Gough's Cave are stored at the Natural History Museum of London (Simon A Parfitt, Silvia M Bello); Wezmeh Cave are stored at Muséum National d'Histoire Naturelle (Marjan Mashkour), Paris; Pinarbasi are stored at University of Liverpool, UK (Douglas Baird, Louise Martin); and Serbia are stored at University of Belgrade - Faculty of Philosophy, Serbia (Vesna Dimitrijevic, |

| | Sonja Vukovic). |

**Dating methods** Collagen extraction and AMS dating was conducted at either the Oxford Radiocarbon Accelerator Unit (Oxford, UK), or the Radiocarbon laboratory of the University of Vienna (Vienna, Austria). Mean age and 95% confidence-intervals were estimated from raw dates (see Supplementary Methods) using the IntCal20 calibration correction in Oxcal v.4.4.

☒ Tick this box to confirm that the raw and calibrated dates are available in the paper or in Supplementary Information.

**Ethics oversight** No ethical approval or guidance was required in this study.

Note that full information on the approval of the study protocol must also be provided in the manuscript.

## Plants

**Seed stocks** *Report on the source of all seed stocks or other plant material used. If applicable, state the seed stock centre and catalogue number. If plant specimens were collected from the field, describe the collection location, date and sampling procedures.*

**Novel plant genotypes** *Describe the methods by which all novel plant genotypes were produced. This includes those generated by transgenic approaches, gene editing, chemical/radiation-based mutagenesis and hybridization. For transgenic lines, describe the transformation method, the number of independent lines analyzed and the generation upon which experiments were performed. For gene-edited lines, describe the editor used, the endogenous sequence targeted for editing, the targeting guide RNA sequence (if applicable) and how the editor was applied.*

**Authentication** *Describe any authentication procedures for each seed stock used or novel genotype generated. Describe any experiments used to assess the effect of a mutation and, where applicable, how potential secondary effects (e.g. second site T-DNA insertions, mosiacism, off-target gene editing) were examined.*

