## [Peer Review File · Nature]

Dogs were widely distributed across Western Eurasia during the Palaeolithic

Corresponding Author: Professor Laurent Frantz

Version 1:

Reviewer comments:

Referee #1

(Remarks to the Author)

The origin of the domestic dog stands as an indefatigably mystery, which many resourceful studies and researchers has been chipping away at for decades. In this regard, Marsh et al. is the largest breakthrough in a decade and likely the biggest milestone yet on the road towards finding the origin of dogs.

The authors present an ambitious and impressive study that extends our understanding of early dog domestication by analysing a remarkable set of newly sequenced Palaeolithic dog genomes from Western Eurasia. Well accompanied by a larger mitochondrial dataset, together building upon previous efforts that have focused on wolves and Holocene dogs. Marsh et al.'s new data pushes further back in time, allowing the authors to trace dog ancestry patterns and reconstruct the demographic and admixture history of early dog populations.

The study's key findings include compelling evidence that Palaeolithic dogs were already widely distributed across Western Eurasia by ~15,000 years ago, showing clear signals of divergence from wolves and genetically, being on the genetic trajectory that continues in dogs today. Overall, this work expands the temporal and geographic knowledge on dog domestication and reshape our understanding of the timing and complexity of the origin of dogs.

The study is of clear relevance to a wide range of academics across both social and natural sciences, as well to the greater public, with the hundreds of millions of dog enthusiasts around the world. I have a few comments and suggestions for the authors, after which I would be happy to see this work published in Nature.

Suggested improvements:

I am in no way any expert in the Epigravettian, but I would advise the authors to double check how they describe potential influence of the Epigravettian culture or ancestry in Palaeolithic UK.

In line 231-233 you write "In the UK however, the earliest evidence of Epigravettian-associated ancestry (between 13,800–13,240 years calBP) postdates the presence of dogs by ~500 years (Figure 2B; 33)."

In line 412-415 you write "In the UK, dogs were associated with humans that carry ancestry associated with the Magdalenian archaeological complex, and predate the arrival of the Epigravettian material culture by almost 500 years 33."

Please clarify the Epigravettian connections to UK, is there influence of Epigravettian material culture, ancestry or both?

Is it known if the culture arrived in the UK?

I also think it could be useful with more elaboration on how Epigravettian ancestry was defined to identify these associations since the culture is so highlighted.

Please add some kind of explanation to why the hole in the masseteric fossa the Gough's Cave dog mandible was anthropic piercing? Is there potentially some literature to cite.

The study often uses a group of coyote genomes as outgroups in various tests, several studies have shown that many coyotes carry dog admixture, see for example;
Monzón, Javier, Roland Kays, and D. E. Dykhuizen. "Assessment of coyote–wolf–dog admixture using ancestry-informative diagnostic SNP s." *Molecular ecology* 23.1 (2014): 182-197.
Have you tested if any of the coyotes used in this study carry some dog diversity?
Have you considered how any canid admixture into the coyotes could affect your tests?

The use of the CanID pipeline is interesting, should the CanID pipeline be considered published as part of this study? If so perhaps present it as part of the study.
Generally, a section on the CanID pipeline in the supplementary would be useful.

Minor comments:

Line 138-139
">1,700 modern dog and wolf genomes (Supplementary Data 3)"
The list seems to include more than dog and wolf genomes?

Line 141
We generated low coverage sequencing data
Is it relevant to be specific about the range of coverage or amount of data somehow?

Line 169-170
"as well as a wolf from the Palaeolithic site of Grotta Continenza in Italy (Extended Figure 1; Supplementary Data 3)."
Should that be Supplementary Data 4?

Line 172-174
"The definitive nuclear genomic identification of dogs from Palaeolithic contexts at Gough's Cave and Pınarbaşı enabled us to re-evaluate existing mitochondrial DNA (mtDNA) data to identify other Palaeolithic canids."
It seems like something is missing or imprecise, I think you mean something like you are able to flag other Palaeolithic canids as likely dogs?

Line 209
"which exceed all other comparisons ($|Z| = 5.4$)"
For readers less familiar with " $|Z|$ " could you potentially elaborate on this result and this noteworthy exceeding?

Line 212
"To test whether this similarity (kinship) is an artefact of inbreeding"
What is the rationale to suspect an artefact of inbreeding? Kinship coefficients (θ) using READv2 is based on pseudo haploid sampling, how can inbreeding change the probability of IBS between individuals in this case?

Line 212
"we imputed (e.g., 28)"
Is 28 the reference for the imputation method or just an example of imputation, I don't fully understand the point of this "(e.g., 28)"?

Line 278
"pre-contact American dogs"
Consider defining what that is?

Line 300
"the dual domestication hypothesis"
An explanation of this hypothesis would be useful, is there a paper to cite potentially?
I am confused if this is the same as the "dual origin" in ref 31. Frantz, L. A. F. et al. Genomic and archaeological evidence suggest a dual origin of domestic dogs. *Science* 352, 1228–1231 (2016).
?

Line 320
"Ancient Near Eastern dogs"
Except for Pınarbaşı? – perhaps elaborate?

Line 337-338
"Alternatively, the ancestral dog population that gave rise to the Basenji may have experienced additional gene flow from African canids 48."
This seems like a puzzling statement, is this irrelevant to the study given frequent use of Basenji in the work. Consider explaining how this is not a problem?

Line 340-344

“When allowing for additional admixture edges ($K = 4$; Extended Figure 7E), we identified contributions from Near Eastern wolves into both the ancestor of African/Near Eastern dogs, and all West Eurasian dogs. With the exception of the Gough’s Cave dog (possibly due to the fact it possess additional basal ancestry not represented in our panel)”
As the Gough’s Cave dog is in a clade with the other western dogs receiving the migration, how can it be an exception?
What is “basal ancestry”, basal to what?
Is it noteworthy there are migrations from the coyote branch?

Line 360-362

“Previous work on ancient East Eurasian dogs has identified shared population histories between dogs and humans, linked to the expansion of different human groups (e.g., Central Steppe Herders; 14,49,50).”
Consider elaborating with more examples and potential exceptions?

Line 377-379

“This indicates that East Eurasian ancestry was present in European dog populations prior to the arrival of the Neolithic from the Near East.”

There is something in this ascertainment that don’t adds up, when the next paragraph explains that the Eastern European Mesolithic dogs from Veretye have already been found to have East Eurasian ancestry.

Consider including the investigation of the Veretye dogs that was published in:

14. Feuerborn, T. R. et al. Modern Siberian dog ancestry was shaped by several thousand years of Eurasian-wide trade and human dispersal. Proc. Natl. Acad. Sci. U. S. A. 118, (2021).

Line 441

Should labeled have been “labelled”?

Line 715

“The small, right population was modified to include a dog from the Americas dating to 4kya and a Dingo”
Could it be elaborated what “the small, right population” is?

Figure 1A

The study is based on six archaeological sites, why is Grotta Continenza not in Figure 1A?

Figure 1A and C

Given the highlighting in the text, consider naming or pointing to the 7,000-year-old dog from Tel Hreiz in Figure 1A and C as you do for other important samples?

Figure 1C

Any particular reason why D_Kartstein1_c12500 is not in this tree?

Figure 1

There are inconsistencies with the dates

In C “D_GoughsCave_14269”

In D “Gough’s Cave (c. 14.3 kya)” and “Bonn-Oberkassel, Germany (14.5 kya)”

In E “Gough’s Cave 14,603 calBP” and “Bonn-Oberkassel 14,309 calBP”

Also consider if formats could be standardised, potentially also for sample names.

Figure 3

Veretye 1 and 2 are strikingly different in Figure 3, in light of the overall thorough discussion of results in the study, consider briefly discussion this observation also.

Further, consider commenting on the wolf-like ancestry in Vlasac1 and Veretye2, some readers may think that is directly Iranian wolf ancestry.

Extended Figure 1

Why is Wezmeh1 unknown when it generally is wolf throughout the manuscript and fall with wolves in the PCA?

All the unknowns seem to have a CanID Status in the supplementary overview, which may be confusing?

Extended Figure 2

There seems to be some inconsistencies with the IDs and dates for some samples compared with the general IDs and dates in the manuscript

TRF.04.09_Veretye_10900 (Extended Figure 2)

OL4061_Veretye_RUSSIA_A_10930 (Extended Figure 2)

Vs.

D_Veretye2_8750 (General in figures)

D_Veretye1_10930 (General in figures)

Extended Figure 2

Is W_GoughsCave3_14297 not in this tree?

Extended Figure 7

Consider specifying the sample(s?) in branch Pleistocene_Wolf

Supplementary text

There seems to be an issue with line spacing in the supplementary text?

Supplementary Figure 11.

It could be relevant with a version of the figure of AMY2B gene copy number, just within the timeframe of dogs.

Supplementary Figure 16, 17, 20 and 21.

For some readers it would be confusing looking for a black dot trying to identify significant tests. Consider modifying the figures or explaining in the figure text.

Supplementary Figure 21.

What is D_Veretye4_10930?, I cant seem to find it in the overviews.

In the supplementary reference list

“Corr, L.T., Berstan, R., Evershed, R.P., 2007. Optimisation of derivatisation procedures for the determination of $\delta^{13}\text{C}$ values of amino acids by gas chromatography/combustion/isotope ratio mass spectrometry. *Raid Commun. Mass Spectrom.* 21, 3759-3771.”

Don't seem to be a numbered reference?

517293_1_extended_data_4784591_sx9ms9 (Supplementary Data 1)

The overview seems incomplete, like the mitochondrial haplotypes are missing for;

D_Padina1_11500-8300

D_Padina3_8300-7900

But I believe they are C based on the manuscript?

Also, there is inconsistencies in fond used.

517293_1_extended_data_4784592_sxmmsm (Supplementary Data 2)

Doublecheck everything is fine in this overview, should cell P16 be empty?

Is the full information given in cell H2-H14?

517293_1_extended_data_4784593_sxwmsw (Supplementary Data 3)

There are only 276 samples in this overview while the manuscript says >1,700.

Many of the 276 samples don't have a "BioProject" or "Accession".

517293_1_extended_data_4784594_sxtmst (Supplementary Data 4)

There are inconsistencies in fond and fond size.

In 4, which tree do NJ refer to?

Consider detailing which analyses columns A, B and C represents?

What is the difference between Sequence ID and Analysis ID?

517293_1_extended_data_4784595_sxfmsf (Supplementary Data 5)

About sample: AL3194_PortauChoix_Canada_A2b_4020 / AL3194

From: Port au Choix, Newfoundland, Canada

The studies "Ramos-Madrigal et al. 2020 / Lin et al. 2023" are cited for reference, should Ramos-Madrigal et al. 2020 be Ní Leathlobhair et al. 2018?

517293_1_extended_data_4784596_sxpmmp (Supplementary Data 6)

There are inconsistencies in fond and fond size.

In one case "et al." is in italics, in the other case it is not.

Also, there is a comment saying "Value's don't match Mikes! - Oliver Craig"

If the comment refers to a still existing issue, please consider that.

Consider using a consistent format for literature references given in the Supplementary Data tables.

Overall, very impressive work,

Best,

Mikkel-Holger Strander Sinding

Referee #2

(Remarks to the Author)

The manuscript "Dogs were widely distributed in Western Eurasia during the Palaeolithic" by William et al. presents genomic data from Pınarbaşı (Türkiye), Gough's Cave (UK), as well as two Mesolithic sites in Serbia. The study combines nuclear and mtDNA with archaeological context and stable isotope analyses on both human and dogs to assess the

geographic scope of early dog domestication. The study contributes significantly to the ongoing debate on the origins and dispersal of early dogs by providing genetic evidence of dog presence in West Eurasia during the paleolithic. The results are novel and well supported by the data. Early dogs from Gough's Cave and Pınarbaşı site demonstrate their phylogenetic placement within domestic dog lineages, and spread alongside the expansion of human ancestry associated with the Epigravettian culture. The data also reveal long-term genetic continuity of dog ancestry in Europe and shows that Palaeolithic and Mesolithic hunter-gatherer migrations and introgression of local wolf DNA in the Near East, offering insight into regional dynamics of domestication and population interaction.

I have a few questions as well as suggestions for improvement, which don't impact any of the major claims of the paper:

1. The assertion that the dog individuals in this study represent domestic dogs rather than wolves is central to this study. However, this claim would be strengthened by a more explicit discussion of the morphological characteristics of these animals, or whether they align more closely with dogs or wolves. In addition to the genetic evidence, a comparative analysis of morphological metrics or a discussion of relevant archaeological interpretations would strengthen this point.
2. Please consider how the current findings contribute to the ongoing debate over single versus multiple domestication centers for dogs. If this finding rule out or supports the dural-origin hypothesis previously proposed?
3. The phrase "widespread across Western Eurasia" might be reconsidered, as the current sample size and distributions are limited to specific regions. Including this limitation would improve the precision of this study.
4. It would be helpful to report imputation metrics and genotype uncertainty, especially for ancient genomes with variable coverage. A brief table in the supplementary Data (e.g., imputation r^2) would improve transparency.
5. The authors claim genetic continuity in European dog ancestry from the Palaeolithic onward. This is an important observation in the context of previous studies reporting significant turnover during later prehistory. A short discussion on how these findings intersect with the known genetic turnovers in European dog populations during the Bronze and Iron Ages would strengthen the broader context of the study (e.g., Botigué et al., 2017; Bergström et al., 2020).
6. The authors report strong genetic similarity between Palaeolithic dogs from Gough's Cave and Pınarbaşı based on outgroup-f3 statistics and phylogenetic analysis. While compelling, this conclusion could be enhanced by f4-statistics or qpGraph modeling to assess whether the observed similarity reflects shared ancestry or gene flow.

Minor suggestions:

1. There is inconsistency in the formatting of isotope notation (e.g., $\delta^{15}\text{N}$ vs. $\delta^{15}\text{N}$). Please standardize throughout the main and supplementary texts.
2. The manuscript inconsistently uses both "archeological" and "archaeological" across sections, I would recommend standardizing to "archaeology" throughout the text.
3. Please clarify "Gene-flow from wolves into Near Eastern dogs", if this is widespread or just localized.
4. Some figure legends could benefit from clearer labeling of new versus previously published samples, and the use of consistent symbols or color coding for sample categories would enhance clarity.

Referee #3

(Remarks to the Author)

In this study Marsh and colleagues attempt to better understand the dynamics of early dog domestication by sequencing ancient Paleolithic and Mesolithic genomes from Eurasia. Specifically, they raise the issue to motivate their study that understanding the evolution from wolves to dogs is made particularly difficult because it can be difficult to distinguish them in the fossil record, particularly for the oldest and potentially most informative samples which are highly fragmented/incomplete/partial. There are numerous specimens that exist from the Paleolithic and Mesolithic across a wide geographic area where it is unclear whether they represent wolves or early dogs. If we could obtain DNA from these fossils this ambiguity could potentially be resolved, leading to a more complete understanding of the process of dog domestication.

Marsh et al are able to sequence eight specimens from such contexts (though two specimens from Wezmeh Cave were found to be younger than previously thought), finding six that were clearly dogs and two which were wolves. Despite the paper being interesting, it is important to note that the study does not resolve the ultimate question about the process of dog domestication (the when, the where and the how). The findings in my opinion are incremental rather than transformative, largely pushing back slightly the time frame of two dog lineages back a few thousand years from that previously established in Bergstrom et al.(2020). Beyond that there are no major new insights into the process of dog domestication. I think the samples are valuable, as they do at least provide some constraints on the timing of dog domestication (at least 16kya), which will be interesting to those interested in the process of dog domestication. But there are serious issues that need to be addressed in terms of some of the analysis and interpretation. I would perhaps urge the authors to simplify what they are trying to do a little bit, and in particular when presenting hypotheses, think about analytical frameworks that provide an actual mechanism to test them.

The ancient DNA work (sequencing of the ancient genomes) is exemplary, reliable and thorough, and the conclusions drawn in the first section of the paper ("Dogs were widely dispersed across West Eurasia by 14,000 years ago") are well supported by the data. The 6 sequenced dogs include two late Paleolithic samples from Turkey and the UK. They authors also establish the canid mtDNA phylogeny context of these samples, allowing them to integrate other ancient specimens that were previously sequenced for their mtDNA (but not autosomes) such as Oberkassel, and impute that these specimens must also have been dogs. Thus this work demonstrates that dogs were domesticated and widespread across Eurasia at least 16kya. This is a simple but powerful conclusion that rests on the excellence of the sampling and is important for those studying the process of dog domestication. This early conclusion in the manuscript is on solid ground and vital.

However, while the paper uses many state-of-the art population genetic methods appropriate for paleogenomic data, the paper then starts to get somewhat speculative in subsequent sections, and more work is required to support the later conclusions, which are often framed initially as testing hypotheses but are never able to provide a clear analytical framework to do so. Also, many of the methods applied are not described very clearly in the main text, and often the reader must infer what was done, either from a figure or the supplementary text.

I list my issues with this remaining part of the manuscript sequentially below.

High degree of similarity between Palaeolithic dogs across West Eurasia

This section draws attention to the fact that the two Paleolithic samples from Gough Cave and Pınarbaşı appear to be genetically very similar despite being quite distant geographically. This certainly is interesting and notable, but the analysis to examine this is somewhat confusing and convoluted. Some clarification is needed about the analysis and why it was conducted and how it was. More importantly, the authors advance a hypothesis that is not supported by the data regarding an Epigravettian-associated expansion.

Line 207: "Further, kinship and pairwise outgroup-f3 analyses (Figure 2A; Extended Figures 3 & 4; Supplementary Figure 7-8) indicate close genetic similarity between the dogs at Gough's Cave and Pınarbaşı"

So what was the kinship estimate from READ (I cannot see it in the supplementary either)? The purpose of the outgroup-f3 analysis in Figure 2A to assess kinship while also correlating with geographic distance is not very clear and left to the reader to figure out. It seems a very imprecise and convoluted way to test for kinship with so many other methods available such as KIN. And then no statement of how related the samples are in terms of degree is ever given. My assumption is that the purpose of the outgroup-f3 is to show that the resulting value is much higher than expected for the two Paleolithic samples given their relative geographic distance but this is never actually stated in the text.

Outgroup-f3 statistics are also measures of similarity and in some parts of the paper are treated as such but in Figure 2a, Extended Figure 3, and their respective descriptions, the outgroup-f3 statistics are labeled as genetic distances. Furthermore, from my understanding the vertical axis in Figure 2a labeled as "Genetic Distance (outgroup-f3)" shows outgroup-f3 values with the higher value denoting higher similarity (as would be expected). In this plot, similarity or shared drift as measured by the outgroup-f3 statistic decreases with geographic distance. The description for the figure describes a positive linear relationship between genetic distance and geographic distance, but the plot in Figure 2a shows a negative linear relationship between similarity/shared drift and geographic distance. This makes sense but as described and plotted is not as clear as it could be. The purpose and conclusion of this analysis should be stated in the text, and the result is interesting enough that it warrants attention. Especially as it is relevant to analyses later in the paper.

Line 212: Estimating ROH for low coverage ancient samples imputed from modern samples seems highly problematic. There are many points of potential error that could occur. The older the sample, the worse the imputation is likely to be. Similarly their lower coverage will compound the issue. It may lead to both false positives and negatives. Some validation is necessary to demonstrate the results are reliable, for example comparing to results for the high coverage NGD sample. I would also strongly recommend the authors look into utilizing HapRoH (Ringbauer et al. Nat Com 2021).

Line 224: "We hypothesize, therefore, that the dispersal of dogs throughout West Eurasia may be linked to the expansion of Epigravettian-associated ancestry and culture seen across Europe after the Last Glacial Maximum between 19,500–13,000 years ago. This timing is also coincident with the most recent common ancestor of dogs in the C haplogroup, estimated to be ~18,900 years ago (95% HPD: 22,989–16,125 years ago)."

There is no reason for these things (the spread of the culture and the mtDNA TMRCA) to be related, unless the implication is that Epigravettian spreading individuals bred one dog strictly to have a single mtDNA lineage, and then brought this lineage of dogs with them everywhere they went, and thus there may be some strong correlation in mtDNA and genome-wide TMRCA. In addition, the following paragraph on the UK dogs being older than Epigravettian culture in the region seems to pretty much suggest that an Epigravettian spread was not involved. Indeed this sample seems to very much falsify the hypothesis proposed in the previous paragraph, so it seems a little convoluted. There are two Paleolithic ancient dog samples. One sample exists with the Epigravettian culture, and one does not. There is not enough evidence here to link to a cultural-associated spread. These two paragraphs (lines 222 and 231) and how they seem to conflict need some rethinking. This is brought up again in the conclusion (lines 409 – 416) and it's suggested that the similarity of these two paleolithic dogs may be associated with the expansion of Epigravettian culture, prior to stating the UK dog predates the arrival of this culture. It is caveated by stating that it may suggest "the existence of interactions between culturally-distinct human populations regardless of gene flow". While what "regardless of gene flow" means is not clear and these ideas require some thought, this may be a conclusion that is better supported by the data.

Close associations between Palaeolithic dogs and humans

The description of post-mortem modifications in the two late Paleolithic dogs is very interesting and valuable. However, though it is important data and justified by the introduction's description of how it has been used to distinguish dogs vs wolf

in the past (line 103), some of the interpretation of the isotope work is questionable.

Line 253: Entire paragraph.

While the comparison of human and dog isotopes might be a good test in principle in terms of looking for dietary overlap, the fact that wolf has the same signature negates the result, and no conclusion can be reached. It is a bit misleading to present dog and human as showing significant overlap first, as if this has some meaning, and then introduce the wolf caveat. Please rephrase. To me a better take away from the isotope work is that caution should be applied when attempting to distinguish dogs and wolves in the paleolithic from dietary isotopes, and that paleogenomic analysis is likely the gold standard to work this out.

Line 265: Entire paragraph.

Again, this paragraph seems to be reaching for meaning when there is not one. It seems a stretch to suggest that humans were feeding dogs based on just being elevated compared to Gough's cave.

Palaeolithic dogs from Gough's Cave and Pınarbaşı belonged to the West Eurasian lineage

While in principle I do not disagree with the final conclusion as stated in the subtitle, there is some uncertainty in how it is being established and how the hypothesis of being an extinct versus existing lineage is actually being tested.

Line 282: "To establish the ancestry of Palaeolithic and Mesolithic European dogs, we calculated shared drift with representatives of Eastern (a ~9,500 years calBP Siberian Hunter-Gatherer dog from Zhokhov Island; 45) and Western (a ~5,800 years calBP dog from the Neolithic Iranian site of Tepe Ghela Gap; 29) dog lineages using outgroup-f3 statistics of the form: $f_3(\text{Coyote}, D_{\text{TepeGhelaGap1_5826}}, D_{\text{Zhokhov1_9515}})$."

It is unclear what is being done here based on the main text, as the outgroup test the authors mention is just one formulation of the test. Do the authors mean conducting two forms of the test like so, $f_3(\text{Coyote}, D_{\text{TepeGhelaGap1_5826}}, X)$ and $f_3(\text{Coyote}, D_{\text{Zhokhov1_9515}}, X)$, rotating through the ancient genomes via X? This seems to be implied by Extended Figure 6, and this same setup is described in Extended Figure 3. If so, please put in the main text, not just let us work it out from the extended figures.

Line 289: "Palaeolithic dogs from Europe and Anatolia share more drift with the West Eurasian lineage (Extended Figures 3 & 6), which is also apparent under PCA (Figure 1C; Supplementary Figure 4). These results indicate that Palaeolithic dogs do not belong to an extinct lineage, but instead form part of the West Eurasian lineage, which indicates that the divergence of East-West dog populations must have occurred by at least 16,000 years ago."

The analytical framework here is not clear. What result would have indicated that the Paleolithic dogs were an extinct lineage in this test? Would they have been along the diagonal of Extended Figure 6? Would they have less shared drift, situated along the diagonal and towards the bottom left? How would the authors distinguish this departure beyond statistical noise? The two Paleolithic samples are also not labeled in the relevant panels in Extended Figure 3 (and in Ext. Fig 3D the Pınarbaşı dog is covered by a label), so it is difficult to assess the results there. The expectations of the test and hypotheses need to be a little bit more defined otherwise a definitive statement as presented in the subtitle cannot be made.

Near Eastern dog ancestry emerged through admixture with local wolves

Again, the authors present competing hypotheses here (post-domestication introgression versus dual domestication, line 298), and while doing many complex analyses, do not make clear how they are distinguishing the two hypotheses. The end result is a conclusion that both hypotheses are still possible, so I am not sure of any real advance here. In addition the results of multiple descriptive analyses are presented that seemingly contradict each other. A clearer analytical framework is needed if the authors are going to contrast these hypotheses.

Line 331: "The highest levels of wolf ancestry (~19%) were identified in a 7,000-year-old dog from Tel Hreiz (Israel)."

In Figure 3B there are higher levels of wolf ancestry estimated for the Veretye 2 sample (Russia) and the Vlasic 1 sample (Serbia), and the Gough's Cave sample seems to be modeled at roughly the same proportions as the Tel Hreiz dog. This 19% seems to be drawn from the f4-ratios shown in Extended Figure 8B, which does not include these two Mesolithic samples or the Paleolithic dog from Gough's Cave. The ADMIXTURE analysis shown in Supplementary Figure 5 includes all of these samples, and shows similar proportions of wolf ancestry in the Tel Hreiz dog and the Gough's Cave dog, but none in the Mesolithic samples mentioned above.

Line 334: "In contrast, high levels of wolf ancestry are maintained in the modern Basenji breed (14–18%)..."

I only see a very small amount of wolf ancestry in Figure 3B for Basenji05, certainly not 15%. There is a discrepancy here with the f4-ratios in Extended Figure 8B, which do show the higher proportions of wolf ancestry described (analyses in Fig. 3B and Ext. Fig. 8B use the same Wezmeh wolf). How is this discrepancy explained? Basenji do not seem to be included in the ADMIXTURE analysis shown in Supplementary Figure 5, so that cannot be compared. Furthermore, there is less of a discrepancy with the wolf proportions estimated in the Tel Hreiz sample, which is included in both of the same analyses (~12% in Fig. 3B and ~19% in Ext. Fig. 8B).

Line 340: "When allowing for additional admixture edges (K = 4; Extended Figure 7E)..."

What was the justification for allowing additional admixture edges? Were there clear parts of the distance matrix with large residuals using 0 or 1 admixture edges? Treemix will put however many edges you ask, but that does not mean it will improve model fit beyond over-fitting. I also see a number of suspicious edges with Coyote in Extended Figure 7 when the migration edges are increased above 1.

Lines 348-351: "While these affinities between Near Eastern wolves and West Eurasian dogs have been identified

previously, the occurrence, timing and extent of gene flow from Near Eastern wolves into local dog populations could not have been identified without the Palaeolithic Pınarbaşı genome.”

I think it is important to make clear how the Pınarbaşı dog has helped identify the occurrence, timing, and extent of gene flow from Near Eastern wolves into local dog populations. This may help focus this section, since it seems some of these tests and conclusions are based around the Pınarbaşı dog having less admixture from Near Eastern wolves than other dogs from the Near East.

Line 355: “Alternatively, it remains possible that part of this ancestry is the remnant of a second, independent domestication of a Western wolf progenitor, followed by gene flow from wolves in the Near East, whose genetic contribution to dog populations in the region decreased over time”.

Where is the evidence for this hypothesis from the results? This possibility seems to come out of nowhere, despite all the tests showing somewhat inconsistent evidence of admixture from Near Eastern wolves (in the sense that there was certainly evidence of Near Eastern wolf admixture into some early dogs, but which dogs show it and their admixture proportions shift when using different types of analyses). Given the premise that was set up at the beginning of the section, how did the authors try to distinguish between the dual domestication and multiple introgression event hypotheses? This has not been clearly laid out and is all rather ad hoc and speculative.

The genomic legacy of West Eurasian Palaeolithic dogs

Figure 3B, Line 375: “In contrast, Southern European Mesolithic dogs from the Iron- Gates in Serbia (Padina and Vlasac) were best modelled as a combination of Western (~58%) and Eastern (~42%) dog ancestry”.

From Figure 3B, the only Serbian Mesolithic dog that seems to have proportions near those described above are Padina1; Padina2 has much less East Eurasian ancestry (~27%), while Vlasac1 is modeled as having no Western Eurasian ancestry and instead ~25% Near Eastern wolf and ~75% Eastern Eurasian ancestry. It is not clear in the text which exact analysis this result is referencing or the corresponding figure but I assume this is based on the qpAdm results.

The previously published Veretye samples are also included, with Veretye1 having near ~50/50 proportions and Veretye2 modeled almost exactly the same as Vlasac1, with ~25% Near Eastern wolf and ~75% Eastern Eurasian ancestry. Both of Veretye samples are from the same context, and previous studies have modeled their ancestry as similar to each other, so this result is confusing and requires substantiation. As mentioned previously, the Veretye2 and Vlasac1 samples also have the highest Near Eastern Wolf proportions of any of the dogs shown in Figure 3B, which is discordant with other analyses presented in this study, and is not addressed (as far as I can tell) in any of the hypotheses or results concerning Near Eastern wolf ancestry. The three dogs from Serbia included in this are low coverage, and substantially lower coverage than the two Veretye dogs (especially Veretye2). A simple difference in coverage may also be difficult to reconcile, since Veretye2 and Vlasac1 have very similar results despite being >5x and 0.1x coverage respectively.

The main text presents results such as these and other qpAdm/qpWave results as very clean and clear cut, but a reading of the figures and supplementary shows a much noisier picture with a lot of inconsistencies. This needs to be addressed in the manuscript and not buried elsewhere.

Line 386: “These results suggest that the introduction of East Eurasian dog ancestry into Europe (which includes mitochondrial haplogroup A) was likely initiated by the spread of Eastern Hunter-Gatherers during the Mesolithic, rather than by later Steppe Pastoralist migrations as previously hypothesised.”

As mentioned in the paragraph above, while the dogs with the highest proportion of Eastern Eurasian ancestry are two Mesolithic dogs from Veretye (Russia) and Vlasac (Serbia), these dogs are also modeled as having ~25% Near Eastern Wolf ancestry. Other dogs from the same areas, associated with the same contexts (and the same human groups that are referenced in the text as having small proportions of Eastern Hunter-Gatherer ancestry), are instead modeled with substantially less East Eurasian ancestry and instead a greater proportion of West Eurasian ancestry (and no Near Eastern Wolf). Looking at Figure 3B, the ancestry proportions of these Mesolithic dogs without Near Eastern Wolf Ancestry (Padina1, Padina3, Veretye1) are not appreciably different from any other European dog that post-dates it. The framing of these results forward in time from the Mesolithic (e.g. referencing the Neolithic and Steppe pastoralists) is strange as the Veretye samples are previously published and the modeling of ancient European dogs (from the Mesolithic onward) with East Eurasian ancestry has been done previously, and is not surprising.

What this study does show is that by sequencing and definitively identifying two Paleolithic dogs for the first time, one of which is modeled without east Eurasian ancestry (which distinguishes it in this analysis from other ancient European dogs that come later), a case could be made that East Eurasian ancestry may have been introduced into Europe sometime between the Mesolithic samples and the Gough's Cave sample. The similarity between the Paleolithic dogs despite their geographic distance could be argued to support this despite the lack of Paleolithic samples from Russia or Serbia, and the geographic distance of the Mesolithic samples from these new Paleolithic samples. As shown earlier in the manuscript, the two Paleolithic dogs are actually more similar to each other than they are to dogs found later, despite these later dogs being found closer geographically. The authors state this themselves on lines 217-220, suggesting it may be due to the presence of Eastern Eurasian dog ancestry in ancient European dogs who come later, but do not mention it in this section. The connection to Eastern Hunter-Gatherer ancestry in human groups is very tenuous and there is no real new evidence presented here to make the claim confidently, nor a clear reason to. This claim is repeated as if it is proven fact on lines 423-424, but the connection is not clear to me.

Lastly, the spread of mitochondrial haplogroup A is mentioned, but from what I can tell, the Mesolithic Serbian samples belong to haplogroup C (shown in Figure 1D). The previously published Mesolithic Veretye samples are haplogroup A. Does this not contradict the claim that is being made? The Serbian samples (Padina1, Padina3, Vlasac1) are also the only samples from this study who have the mitochondrial haplotype column blank in the extended data, so all I can go off of is the Figure 1D. Taken together it makes it hard to see the validity of the association of East Eurasian Dog ancestry with specifically human eastern hunter-gatherer ancestry.

Line 396: "While this does not preclude the possibility of additional influxes of East Eurasian dog ancestry into Europe since the Mesolithic, these findings suggest that the fundamental ancestry composition which characterises many popular modern European breeds such as Border Collie, Cavalier King Charles Spaniel, German Shorthaired Pointer, and Saint Bernard (Supplementary Figure 18) was established by at-least 10,000 years ago."

Looking at Supplementary Figure 18 it is difficult to see how these breeds are characterized by these ancestry components, since they all look indistinguishable. It seems more accurate to say that little of the structure and differentiation present in modern European breeds and breed groups was established at this point in the past, though they all seem to have some common ancestry reaching back at least 10,000 years ago.

Minor Edits:

Figure 1: It may improve Figure 1 to include a label for the C5 clade and help clarify what is meant in the main text (lines 186- 188). As it stands it is difficult to know exactly which dogs are included in the C5 group and to connect the text to the figure.

Figure 1d, Line 445-446: The description for Fig. 1d says to see Supplementary Figure 3 for the complete mitochondrial tree, but Supplementary Figure 3 seems to be a plot of calibrated radiocarbon dates.

Line 245: "Similar post-mortem modification is evident on the dog remains including an anthropic perforation on the masseteric fossa (Figure 2B-C; Supplementary Figure 2)."

I see the picture for 2B but how does this apply to 2C?

Lines 318-320: the D statistic described is not the one shown in Supplementary Figure 13. It seems to be shown in Supplementary Figure 14. This issue also appears on line 348 where Supplementary Figure 13 is referenced again incorrectly.

Line 790: "Neolithic-recent" is confusing.

Supplementary Figure 6: The tree is difficult to read when zoomed in.

Supplementary Figure 7: What is the rationale for this? Specifically, why is the Pinarbasi dog compared only to African/Near-Eastern dogs and the Gough's cave dog to Ancient European dogs?

Referee #4

(Remarks to the Author)

I co-reviewed this manuscript with one of the reviewers who provided the listed reports.

Version 2:

Reviewer comments:

Referee #1

(Remarks to the Author)

Thank you very much for the effort addressing my comments, I only have a few minor points below after which I would be happy to see the work published.

Lines 142-146

"These data were analysed alongside previously published ancient dog (n = 65) and wolf (n = 66) genomes spanning the last 100,000 years (Supplementary Data 1), as well as 276 modern dog, wolf, and outgroup canid genomes (a representative subset of 1,700 genomes in the NHGRI Dog Genome Project Database; Supplementary Data 3; 20)."

I greatly appreciate the work on the supplementary material and accompanying elaborations in main text and supplementary, however I am sorry I am still confused. I cant get the numbers in the text to match the numbers in the supplementary tables. Please just double check the numbers, I am sorry if I am misunderstanding.

In Supplementary Data 1, I count;

68 dogs (excluding you 6 dogs to be released in this study)

71 wolves (excluding you 2 wolves to be released in this study)

1 dhole

So not "previously published ancient dog (n = 65) and wolf (n = 66)"

Also W_GoughsCave1_14297 and W_Wezmeh1_2708 are listed as dog.

In Supplementary Data 3, I count

199 dogs and wolves

So not "276 modern dog, wolf, and outgroup canid genomes"

Also when you state "(a representative subset of 1,700 genomes in the NHGRI Dog Genome Project Database; Supplementary Data 3; 20)."

To me it sounds like there should be 1700 genomes in Supplementary Data 3.
Is it important to state “a representative subset of 1,700 genomes in the NHGRI Dog Genome Project Database”? or maybe put the reference to Supplementary Data 3 after the actual number of genomes included.

Line 315
“signatures, , our”
Comma mistake?

Lines 325-326
“Dogs can be broadly separated into two lineages: an Eastern lineage, primarily found in Arctic, East Asian, and American dogs prior to AD1492,”
This sounds like American dogs went extinct in AD1492, consider modifying a bit, maybe write they went extinct in historical time? potentially cite “Lin, Audrey T., et al. “The history of Coast Salish “woolly dogs” revealed by ancient genomics and Indigenous Knowledge.” Science 382.6676 (2023): 1303-1308.”?

Supplementary line 200
Should there be a comma in 15000?

Supplementary Data 3
Is there no literature reference for “Kendrick's Cave”? Is it published as part of this study?

Finally about the ID of Khatystyr1_8645, should that be Khatystyr1_9682 if the age is the number?

Well done,
Best,
Mikkel Sinding

Referee #2

(Remarks to the Author)
The revision is strong, cohesive, and very close to acceptance. The paper now delivers a convincing synthesis that Late Glacial to early Holocene dogs were present and geographically widespread in Europe and Anatolia. The statistical framework is sound, the archaeological context is clearer, and the figures support the narrative effectively. The manuscript will be valuable to archaeogenetics, zooarchaeology, and Paleolithic studies. With a few targeted clarifications to align causal phrasing and transparency, this is ready for publication.

First, please keep the link between dog dispersal and Epigravettian-related human expansions as one plausible scenario rather than the primary driver. Briefly list alternatives in the main text, for example, postglacial connectivity among multiple western European and Mediterranean refugia, or exchange networks that need not track major ancestry shifts. Add one sentence stating that present sampling does not yet discriminate among these mechanisms.

Second, add a short paragraph explicitly reconciling the early two-source structure with later Neolithic, Bronze Age, and Iron Age changes. A simple forest plot that summarizes qpAdm proportions across periods with medians and uncertainty would separate structural continuity from shifts in mixture fractions.

Third, for f4-ratio or related estimates, give standard errors or confidence intervals, list the exact right-set populations, and justify the Near Eastern proxy choice. If possible, include more than one Near Eastern wolf proxy and summarize concordance.

Fourth, in the supplement, please provide the complete list of coyote outgroups with estimated dog ancestry and uncertainty. Add a compact sensitivity check using an alternative canid or non-canid outgroup for a few sentinel tests to demonstrate that small levels of dog introgression in coyotes do not shift inferences.

Fifth, please consider including the imputation probabilities in the supplementary table. Also, in the kinship paragraph, report a single summary statistic in the main text, for example, median READ theta with an interquartile range, and place full pairwise tables in the supplement.

Last, please typeset the delta (δ) symbol in italics in all isotope notations.

Referee #3

(Remarks to the Author)
In general the authors have done a very robust job in addressing most of my earlier criticisms of the paper. It is now much easier to read and the conclusions are more grounded, and involve less handy-wavy speculation. The new section describes the model of the spread of paleolithic dog ancestry is much easier to understand, and supported by some

interesting new analyses involving dog and human ancestry across space.

There are a few minor points/issues that perhaps need some clarification in a finished draft (but do not necessarily require my re-review).

Basal Ancestry in Gough's Cave

The supplementary material mentions the presence of some form of basal ancestry in the Gough's cave sample (line 1149 Supplementary) that appears to complicate its use in downstream analysis. It is curious that the authors do not mention this in the main manuscript at all, and I wonder why, given it seems an interesting potential finding. There seems to be some similarity to the wolf from Gough Cave (Supp fig 18). It seems to somewhat parallel the idea of near eastern wolf ancestry in near eastern dogs. And this somewhat contradicts the conclusion about a barrier to gene flow between dogs and wolves by the upper paleolithic (line 472-475).

Dog-Human Ancestry Comparison

Lines 236-251 (also lines 712-721, ext. Figure 3, Supp. lines 924-965): "We assessed the extent to which dogs and humans shared their demographic history by comparing the genetic similarity of dogs from Pınarbaşı and Gough's Cave to that of humans at each site."

This analysis is detailed under Dog-Human Ancestry Comparison in the supplemental material. This describes an interesting isolation-by-distance pattern inferred independently within each species (human and dog), which is new to the manuscript and certainly adds some useful and valuable insights. The authors show that genetic distance between ancient humans is spatially autocorrelated, and the genetic distance between ancient dogs is spatially autocorrelated. The following paragraph describes the partial Mantel test, used to compare pairwise outgroup-f3 matrices in humans and in dogs. A p-value is given in the main text on lines 242-243 (" $p < 0.0002$ "); attaching significance to the resulting Mantel r and used to support the conclusion that "human and dog ancestries are closely linked" (line 244). This is stated as reflecting "shared demographic history between humans and dogs" (supp. line 951). Fundamentally I do not think this analysis addressed shared demographic history and that it really doesn't answer a question beyond both species being spatially autocorrelated. For example, if we took other species that don't actually share a demographic history, we may also find a similar pattern simply due to the general common feature of spatial autocorrelation.

Additionally, there are potential statistical issues with this approach. A number of papers in the literature have demonstrated that the Mantel test (and partial Mantel test) suffer from an excess of Type I error in the case where both variables being tested are structured or autocorrelated (Guillot & Rousset, 2013; Quilodrán et al., 2023). Quilodrán et al. 2023 explicitly simulate a similar scenario, demonstrating the resulting Type 1 error rates. In addition, on page 11 they suggest some helpful guidelines for alternatives and conservative interpretation. I would urge caution in interpretation of the results of this test and how it's described in the manuscript.

The genomic legacy of Western Eurasian Palaeolithic dogs

This section focuses on a straw-man hypothesis (stated on lines 404-407) apparently drawn from the literature (initially citing a study of ancient humans). Line 437 cites the dog literature they are referencing. Even being charitable in the interpretation of east Eurasian dog ancestry, this is not addressing a hypothesis that seems to exist in the literature.

Putting its existence aside, the relevance of this hypothesis and how it is addressed by the new data in this study is unclear. Lines 414-416 reference the same hypothesis. Why did Eastern Eurasian ancestry reach parts of Europe earlier than we thought? The text states "other parts of Europe". What other parts of Europe exactly? The dog from Veretye was modeled as deriving ancestry from a similar east Eurasian source in 2020, so this is not exactly new information. The argument on lines 433-437 reference the results of the partial Mantel test (which has issues that were described above). Even ignoring those issues, I do not think the test is showing a correlation between particular ancestries at any time or place. What is the relevance of this new data to this question? It isn't stated.

I would suggest this section be either removed or reduced somewhat. The paper as is does not really need it.

Modern dog model fit

Line 44: "Our qpAdm analysis also indicates that the Eastern Eurasian dog ancestry persisted in European dog populations from the Mesolithic to the present. Approximately 30% of this ancestry component is present in dogs during the Neolithic (e.g., Pločnik, Serbia), 33% during the Bronze Age (e.g., Belverde Di Cetona, Italy), 22% during the Iron Age (e.g., Parknabinnia, Ireland), 18% during the Medieval period (e.g., Vilnius Castle, Lithuania) and ~20% in modern dogs (Figure 3B). Whilst this does not preclude additional influxes of Eastern Eurasian dog ancestry into Europe post- Mesolithic, 52,56 these findings show that the fundamental ancestry components in European dogs (i.e. both Eastern and Western Eurasian ancestry) which characterises many popular modern breeds including the Cavalier King Charles Spaniel, German Shorthaired Pointer, Jack Russell and Labrador (Supplementary Figure 17) was established by at-least 10,900 years ago.." The authors mention the fit to modern samples here, but actually Supplementary Figure 17 seems to show a very poor fit for almost every modern dog tested, with very low qpAdm p-values. So this statement does not seem particularly justified based simply on this analysis.

Dog Exchange

Line 459: "The presence of dogs in close association with humans possessing both Epigravettian and Magdalenian ancestry, however, implies exchange of dogs between culturally distinct human populations during the Late Upper Palaeolithic, sometimes in the absence of widespread gene flow or population turnover, as seen at Gough's Cave." I wonder if the wording here should be softened a bit. While "exchange" can of course mean many different things depending

on the context, as it's used here, it kind of implies dogs were actively being exchanged, almost like they were being traded. This may not be the author's intention but that's how it reads. However, there is no reason to distinguish this kind of active movement between human cultures, and ones where dogs just came with one culture and then ran around like village dogs and became part of the gene pool in the region.

Minor edits:

Line 371: incorrectly states the results of the supervised ADMIXTURE analysis are shown in supplementary figure 12. Consequently the reference to figure supplementary figure 14 on line 386 is incorrect.

Line 336: "Palaeolithic dogs from Europe and Anatolia share more drift with the Western Eurasian lineage (Extended Figures 3–5)"

I do not think Extended Figure 3 is relevant here.

Line 516: incorrectly references supp. fig 17 for additional admixture bayes graphs.

Extended Figure 3: Might be useful somewhere in the main text or figure legend to explain that the geographic distance and time are standardized with mean 0, otherwise the units as presented in the figure are a little confusing.

Referee #4

(Remarks to the Author)

I co-reviewed this manuscript with one of the reviewers who provided the listed reports.

Authors response - 2025-05-12885A

Referee #1 (Remarks to the Author):

The origin of the domestic dog stands as an indefatigable mystery, which many resourceful studies and researchers have been chipping away at for decades. In this regard, Marsh et al. is the largest breakthrough in a decade and likely the biggest milestone yet on the road towards finding the origin of dogs.

The authors present an ambitious and impressive study that extends our understanding of early dog domestication by analysing a remarkable set of newly sequenced Palaeolithic dog genomes from Western Eurasia. Well accompanied by a larger mitochondrial dataset, together building upon previous efforts that have focused on wolves and Holocene dogs. Marsh et al.'s new data pushes further back in time, allowing the authors to trace dog ancestry patterns and reconstruct the demographic and admixture history of early dog populations.

The study's key findings include compelling evidence that Palaeolithic dogs were already widely distributed across Western Eurasia by ~15,000 years ago, showing clear signals of divergence from wolves and genetically, being on the genetic trajectory that continues in dogs today. Overall, this work expands the temporal and geographic knowledge on dog domestication and reshapes our understanding of the timing and complexity of the origin of dogs.

The study is of clear relevance to a wide range of academics across both social and natural sciences, as well to the greater public, with the hundreds of millions of dog enthusiasts around the world. I have a few comments and suggestions for the authors, after which I would be happy to see this work published in Nature.

Suggested improvements:

I am in no way an expert in the Epigravettian, but I would advise the authors to double check how they describe potential influence of the Epigravettian culture or ancestry in Palaeolithic UK.

In line 231-233 you write "In the UK however, the earliest evidence of Epigravettian-associated ancestry (between 13,800–13,240 years calBP) postdates the presence of dogs by ~500 years (Figure 2B; 33)."

In line 412-415 you write "In the UK, dogs were associated with humans that carry ancestry associated with the Magdalenian archaeological complex, and predate the arrival of the Epigravettian material culture by almost 500 years 33."

Please clarify the Epigravettian connections to UK, is there influence of Epigravettian material culture, ancestry or both?

To avoid conflating material culture with genetic ancestry, we have made sure the text specifies whether we are referring to Epigravettian material culture or Epigravettian-associated human ancestry where appropriate.

Is it known if the culture arrived in the UK? I also think it could be useful with more elaboration on how Epigravettian ancestry was defined to identify these associations since the culture is so highlighted.

We agree that additional information about the transition from Magdalenian to Epigravettian ancestry and material culture in the UK would be useful. We have substantially amended this section to address the reviewer's concern (lines 223-283):

Of the five Palaeolithic dogs identified (Figure 2A), each are associated with one of three genetically and culturally distinct human hunter-gatherer populations found across Europe and Anatolia in the Late Upper Palaeolithic: the Magdalenians (Gough's Cave^{1,2}), the Epigravettians (Bonn-Oberkassel, Kesslerloch and Grotta Paglicci^{3,4,5}), and Anatolian Hunter-Gatherers (Pınarbaşı⁶). The spread of these dogs across the region is likely to have been linked with the migration, dispersal and interaction of humans associated with these Palaeolithic cultures. In fact, while most Palaeolithic dogs in this study are associated with human populations of Epigravettian-related ancestry, including Anatolian Hunter-Gatherers at Pınarbaşı⁷, the dog from Gough's Cave was recovered from a depositional context alongside humans with Magdalenian-associated ancestry.

We assessed whether dogs were exchanged between genetically distinct human groups by comparing the genetic similarity of dogs from Pınarbaşı and Gough's Cave to that of humans at each site. We first established the degree to which humans and dog ancestries from the same archaeological contexts, across 35 sites that spanned the Late Upper Palaeolithic to medieval period (e.g., Vilnius Castle, Lithuania), were correlated using outgroup-f₃ statistics (Extended Figure 3C, Extended Data 7). We found a strong positive correlation (Mantel $r \approx 0.40$, $p < 0.0002$) between human-human and dog-dog outgroup-f₃ values, which indicates the human and dog ancestries are closely linked. This analysis shows the dogs from the Palaeolithic sites of Pınarbaşı and Gough's Cave were more genetically similar than expected when compared to humans at the site (Extended Figure 3D). This provides further evidence that a relatively homogeneous dog population spread between genetically and culturally distinct human populations across Europe and Anatolia in the Late Upper Palaeolithic.

One plausible scenario is that the dispersal of these Palaeolithic dogs was coupled with the spread of Epigravettian-associated ancestry and material culture ~16,000 years ago, which, after a period of interaction, eventually replaced the previously predominant Magdalenian culture in Northern Europe⁸⁻¹⁰. This timeframe aligns well with the mtDNA-based TMRCA of the Gough's Cave and Pınarbaşı dogs (95% HPD: 18,569–15,860 years ago), which provides both an upper bound for the divergence of their ancestral populations, and a minimum age associated with the youngest radiocarbon date (i.e.

Gough's Cave dog: 14,808–14,091 calBP; Supplementary Data 4). Crucially, this timeframe postdates with the earlier divergence of Magdalenian and Epigravettian human populations, which most likely occurred during or prior to the Last Glacial Maximum (~24,000–21,000 years ago; ^{8,11}). The spread of Palaeolithic dogs across Western Eurasia, therefore, likely occurred after the divergence of these two distinct human populations.

The Gough's Cave dog dates to the Late Magdalenian (from 15,000 years ago; Figure 2B), a period for which there is as yet no evidence of humans carrying Epigravettian-associated ancestry in the British Isles ¹⁰. This period, however, is characterized by transitions in lithic technology (e.g., *Federmessergroupen*) that are associated with people carrying Epigravettian-related ancestry on the European continent (e.g., at Bonn-Oberkassel) ^{12–15}, whilst later individuals from the British Isles (e.g., Kendrick's Cave: 13,780–13,354 years calBP) do carry Epigravettian-associated ancestry ^{8,10}. Further, the putative presence of dogs (based on short mtDNA fragments) has also been suggested at another Magdalenian context in Spain (Erralla Cave, 17,410–17,096 years cal BP; ¹⁶) and postdates the earliest evidence of Epigravettian ancestry in the region by over 1,000 years (El Mirón, ~18,700 years calBP; ^{17,18}).

Combined, our results suggest that dogs dispersed across Late Palaeolithic Europe alongside the expansion of Epigravettian-associated ancestry and material culture over 16,000 years ago. Under this scenario, Magdalenian humans acquired dogs through interactions with Epigravettians. These interactions did not leave any signal of Epigravettian ancestry in the Magdalenian humans at Gough's Cave ¹⁰, implying that the exchange of dogs between Palaeolithic human groups was not always accompanied by detectable gene flow.

Please add some kind of explanation to why the hole in the masseteric fossa the Gough's Cave dog mandible was anthropic piercing? Is there potentially some literature to cite.

This statement is based on initial investigation of the mandible, which show modifications similar to that seen on the human remains, we have clarified this in the supplementary We have added a further description of the element in the SI to clarify why this specimens was initially deemed to be a domestic dog (lines 96-100):

“The masseteric fossa of both hemi-mandibles are perforated by sub-circular openings. These perforations are ancient in origin and were created by carefully placed percussion blows. This specimen has previously been classified as a domestic dog from morphometric analysis given its reduced size compared with Palaeolithic and modern wolves.”

And in the main text (lines 131-135):

and canid remains from a Late Upper Palaeolithic Magdalenian horizon (15,100–14,200 years calBP) at Gough's Cave in the British Isles (Supplementary Figure 2; ^{1,2}), that show post-mortem treatment mirroring the human remains at the site ^{1,19}.

The study often uses a group of coyote genomes as outgroups in various tests, several studies have shown that many coyotes carry dog admixture, see for example; Monzón, Javier, Roland Kays, and D. E. Dykhuizen. "Assessment of coyote–wolf–dog admixture using ancestry-informative diagnostic SNP s." *Molecular ecology* 23.1 (2014): 182-197.

Have you tested if any of the coyotes used in this study carry some dog diversity?

Have you considered how any canid admixture into the coyotes could affect your tests?

Previous studies have used a coyote as an outgroup for ancestry analyses (i.e. Coyote01 in Figures below). Here, we used multiple coyotes as outgroups for most of our analyses, which were largely free of dog ancestry (compared to others included in the NHGRI Dog Genome Project Database VCF). Outgroup-f3 of the form f3(BlackBackedJackal01, Coyote, Dog/Wolf) showed that Coyote01 shared slightly less drift with dogs and wolves than other coyotes:

Using f4-ratio, we can see however that the excess dog ancestry in other coyotes relative to Coyote01 is marginal (~3 -6%):

Comparing results of simple D-statistics of the form $D(\text{Outgroup}, \text{Near Eastern Wolf}, X, D_Pinarbasi1_15787)$, where “Outgroup” is either Coyote01, or all coyotes without significant dog introgression (i.e. those used throughout the manuscript), shows little difference:

This is likely due to the fact that the very low amount of dog ancestry in the genome of these do not affect polarisation of alleles, at worse they will lead to conflicting genotypes among coyotes, which will lead to missing data in analysis for which we need alleles to be polarised.

The use of the CanID pipeline is interesting, should the CanID pipeline be considered published as part of this study? If so, perhaps present it as part of the study. Generally, a section on the CanID pipeline in the supplementary would be useful.

We have added the following section in the supplementary material (under heading “CanID Taxonomic Identification” line 758), which outlines the methodology that underlines CanID:

To determine the taxonomic status of all canid samples from which we generated low-pass screening data ($n = 24$), we utilised the canid identification (CanID) pipeline (<https://github.com/lachiescarsbrook/CanID>). This workflow generates pseudohaploid genotype calls at sites present in a reference panel of modern dogs and wolves (~2-million diagnostic transversions), and uses both projection-based Principal Component (PC) analysis and Linear Discriminant Analysis (LDA) to classify individuals as either dogs or wolves. With as few as 175 SNPs, this method can accurately (95-100% success rate) distinguish most ancient dog and wolf ancestries (excluding Pleistocene wolves, which require 275 SNPs for the same accuracy). 21 out of 24 samples yielded enough SNPs for accurate classification (between 267–54,013 SNPs), with five wolves and 16 dogs identified (Extended Figure 1). This included three wolves, and four dogs from Gough’s Cave, with one of the dogs directly-dated to within the Magdalenian occupation layer (D_GoughsCave2_14604: 1,363 SNPs); four dogs from Pınarbaşı; and a wolf from Grotta Continenza (W_GrottaContinenza1_c12000: 267 SNPs). Of the three samples with too few SNPs for CanID classification, two were deep sequenced and were identified as dogs (i.e., D_Padina1_c8500 and D_Wezmeh1_1532), and the remaining sample (D_Vlasac1_c8650) shared maternal affinities with dogs at the same site (see below).

The GitHub for CanID (<https://github.com/lachiescarsbrook/CanID>) also has an extensive description of the workflow.

Minor comments:

Line 138-139: “>1,700 modern dog and wolf genomes (Supplementary Data 3)”
The list seems to include more than dog and wolf genomes?

We have changed this sentence to (lines 142-147):

These data were analysed alongside previously published ancient dog ($n = 65$) and wolf ($n = 66$) genomes spanning the last 100,000 years (Supplementary Data 1), as well as 276 modern dog, wolf, and outgroup canid genomes (a representative subset of 1,700 genomes in the NHGRI Dog Genome Project Database; Supplementary Data 3; ²⁰).

Line 141: We generated low coverage sequencing data

Is it relevant to be specific about the range of coverage or amount of data somehow?

The low coverage sequencing data mentioned here referred to the low depth screening performed for all samples prior to deeper sequencing. This did not produce enough data to be included in our analyses, with the exception of CanID, which specifically utilises ultra low coverage (screening) data (i.e. <5M reads). We have now specified the number of reads sequenced for each sample in Supplementary Data 4, and listed the cut-off threshold of <0.001x for low coverage sequencing data in the main text.

We have also modified all mention of coverage (both nuclear and mitochondrial) to include the median depth, as well as the range.

Line 169-170: “as well as a wolf from the Palaeolithic site of Grotta Continenza in Italy (Extended Figure 1; Supplementary Data 3).”

Should that be Supplementary Data 4?

Yes this is correct, we have corrected this typo.

Line 172-174: “The definitive nuclear genomic identification of dogs from Palaeolithic contexts at Gough’s Cave and Pınarbaşı enabled us to re-evaluate existing mitochondrial DNA (mtDNA) data to identify other Palaeolithic canids.”

It seems like something is missing or imprecise, I think you mean something like you are able to flag other Palaeolithic canids as likely dogs?

We agree, the previous sentence was confusing. We have changed to the following (lines 180-182):

“The definitive nuclear genome-based identification of dogs from Gough’s Cave and Pınarbaşı enabled us to re-evaluate existing mitochondrial DNA (mtDNA) data to assess the status of other Palaeolithic canid.”

Line 209: “which exceed all other comparisons ($|Z| = 5.4$)”

For readers less familiar with “ $|Z|$ ” could you potentially elaborate on this result and this noteworthy exceeding?

We have used $|Z|$ to denote the absolute value of Z. We have changed “ $|Z|$ ” to “ $|Z\text{-score}|$ ” throughout the manuscript to make it easier for readers to comprehend. The specific comparison referred to here has since been removed from the manuscript.

Line 212: “To test whether this similarity (kinship) is an artefact of inbreeding”

What is the rationale to suspect an artefact of inbreeding? Kinship coefficients (θ) using READv2 is based on pseudo haploid sampling, how can inbreeding change the probability of IBS between individuals in this case?

We agree with the reviewer that this sentence was confusing, and we have removed it. We have also removed the inbreeding analysis (based on RoH), as it provided limited insight and was potentially confounded by imputation issues.

Line 212

“we imputed (e.g., 28)”

Is 28 the reference for the imputation method or just an example of imputation, I don't fully understand the point of this “(e.g., 28)”?

We have removed all reference to imputation in the main text, see above.

Line 278: “pre-contact American dogs”

Consider defining what that is?

We have changed to: “pre-contact American dogs (i.e. those found in the Americas before AD1492)”.

Line 300: “the dual domestication hypothesis”

An explanation of this hypothesis would be useful, is there a paper to cite potentially?

I am confused if this is the same as the “dual origin” in ref 31. Frantz, L. A. F. et al. Genomic and archaeological evidence suggest a dual origin of domestic dogs. Science 352, 1228–1231 (2016).

For consistency, we have changed the mention of “dual domestication” to “dual origin” throughout the manuscript.

Line 320: “Ancient Near Eastern dogs”

Except for Pınarbaşı? – perhaps elaborate?

We have specified the timeframe of the dogs under consideration as “Ancient Near Eastern dogs from the Neolithic onwards”, thus excluding the Pınarbaşı sample.

Line 337-338: “Alternatively, the ancestral dog population that gave rise to the Basenji may have experienced additional gene flow from African canids 48.”

This seems like a puzzling statement, is this irrelevant to the study given frequent use of Basenji in the work. Consider explaining how this is not a problem?

Basenjis are indeed thought to be admixed with African wolves (Fan et al. 2016). However, all Near Eastern dogs also possess varying levels of wolf ancestry. Basenjis remain the best proxy for these analyses for two reasons: 1) their genomes are sequenced at higher coverage than

early Near Eastern dog genomes, 2) and their isolation in Africa prevented admixture with Eurasian dog lineages, which affected modern Near Eastern dogs.

Line 340-344

“When allowing for additional admixture edges ($K = 4$; Extended Figure 7E), we identified contributions from Near Eastern wolves into both the ancestor of African/Near Eastern dogs, and all West Eurasian dogs. With the exception of the Gough’s Cave dog (possibly due to the fact it possess additional basal ancestry not represented in our panel”

As the Gough’s Cave dog is in a clade with the other western dogs receiving the migration, how can it be an exception?

What is “basal ancestry”, basal to what?

Is it noteworthy there are migrations from the coyote branch?

We have performed additional analysis to further investigate the so-called “basal” ancestry in the Gough’s Cave individual. These are included in a new section of the SI - “Investigating cryptic ancestry in the Gough’s Cave dog - M13794”. We have run f_4 -stats of the form $D\{BBJ, Coy; Pin, GC\} + D\{BBJ, Wolf; Pin, GC\}$ to test for attraction of Gough’s Cave to basal canid ancestry. In all cases, wolves and coyotes share more drift with Pinarbasi over Gough’s Cave, the converse of results from $qpAdm$ and within dog f -statistics. We have also performed additional contamination testing on sample libraries (in *kraken2* and *FASTQSCREEN*; ^{21,22}), with results in all cases ruling out the alignment of non-endogenous (i.e human) reads (when filtering for high endo libs and libs where assigned dog:human ratio is $>150:1$, results are the same). We thus do believe this signal to be genuine, although at present cannot be certain of its source.

We have added in an additional section to the SI - “Basal ancestry in the Gough’s Cave dog” - lines 1152-1181).

Line 360-362: “Previous work on ancient East Eurasian dogs has identified shared population histories between dogs and humans, linked to the expansion of different human groups (e.g., Central Steppe Herders; 14,49,50).”

Consider elaborating with more examples and potential exceptions?

This is beyond the scope of this paper - the details can be found in multiple other publications ^{23,24}

Line 377-379

“This indicates that East Eurasian ancestry was present in European dog populations prior to the arrival of the Neolithic from the Near East.”

There is something in this ascertainment that don’t adds up, when the next paragraph explains that the Eastern European Mesolithic dogs from Veretye have already been found to have East Eurasian ancestry.

Consider including the investigation of the Veretye dogs that was published in:

14. Feuerborn, T. R. et al. Modern Siberian dog ancestry was shaped by several thousand years of Eurasian-wide trade and human dispersal. Proc. Natl. Acad. Sci. U. S. A. 118, (2021).

We have modified language here for clarity (lines 418-437):

To assess the degree of Eastern Eurasian dog ancestry in Western Eurasian dogs, we implemented a tri-source model of dog ancestry in *qpAdm* (Figure 3B; Supplementary Figure 17), using three ancestry sources based on the results of D-statistics and admixture graphs: Palaeolithic Western Eurasian (D_Pinarbasi1_15787) and Eastern Eurasian (D_Zhokhov1_9515) dogs, and Near Eastern wolves (W_Wezmeh1_2708). As expected, the majority of ancient Near Eastern dogs possessed both Western Eurasian dog and Near Eastern wolf ancestry. The Eastern Eurasian dog ancestry component is absent until the Early Byzantine Period (1,565 years calBP at Marmara, Türkiye). In contrast, Balkan Mesolithic dogs from the Iron-Gates in Serbia (Padina and Vlasac) and Northwest Russia (Veretye) were best modelled as a combination of Western Eurasian (mean: 56.2%) and Eastern Eurasian (mean: 43.8%) dog ancestry.

Genomic analyses of Mesolithic Hunter-Gatherers from the same contexts as several of these dogs (Iron-Gates and Veretye) have shown that they also possessed EHG genetic ancestry^{8,25}. Coupled with the positive correlation between human-dog ancestries (Extended Figure 3C), our results suggest that Eastern Eurasian dog ancestry may have been first introduced into Europe during the spread of Eastern Hunter-Gatherers during the Mesolithic⁸, rather than during the later Steppe Pastoralist migrations as previously hypothesised²⁶.

Line 441

Should labeled have been “labelled”?

Corrected.

Line 715

“The small, right population was modified to include a dog from the Americas dating to 4kya and a Dingo”

Could it be elaborated what “the small, right population” is?

The small, right population refers to a set of canid genomes used as “reference” populations in *qpAdm* analysis. This set of individuals incorporates canids that are differentially related to individuals used as source populations. We have modified the text for improved clarity in the methods section.

Figure 1A

The study is based on six archaeological sites, why is Grotta Continenza not in Figure 1A?

We have modified figure 1A to highlight the locations of key mtDNA sites (e.g. Grotta Continenza, Bonn-Oberkassel).

Figure 1A and C

Given the highlighting in the text, consider naming or pointing to the 7,000-year-old dog from Tel Hreiz in Figure 1A and C as you do for other important samples?

We mention a number of individuals in the text, which are not critical to the manuscript (e.g. Ashkelon, Marmara). If we included all sites mentioned in the text, the figure would become too crowded. We only emphasise the location (i.e. spell out the site name) for sites targeted in this study, or those used as key ancestry sources, both dog (e.g. Tepe Ghela Gap, Veretye, Zhokhov) and human (e.g. Bonn-Oberkassel). As per the above recommendation, however, key mtDNA sites are now emphasised.

Figure 1C

Any particular reason why D_Kartstein1_c12500 is not in this tree?

We have now highlighted the position of the Karstein Cave dog within the C1-3 clade in Figure 1C.

Figure 1

There are inconsistencies with the dates

In C "D_GoughsCave_14269"

In D "Gough's Cave (c. 14.3 kya)" and "Bonn-Oberkassel, Germany (14.5 kya)"

In E "Gough's Cave 14,603 calBP" and "Bonn-Oberkassel 14,309 calBP"

Also consider if formats could be standardised, potentially also for sample names.

For Gough's Cave in panel "C" and "D", and Bonn-Oberkassel in panel "D" and "E", differences in the values of age estimates represent rounding errors, and have been corrected. Gough's Cave in panel "E", is not the same individual as in panel "C" and "D" i.e. it is the radiocarbon dated dog at Gough's Cave, identified as dog based on CanID). To make this distinction clearer, Figure 1E has been moved to replace Figure 2A. The new figure caption now reads as:

"(A) Earliest directly-dated dogs from Late Upper Palaeolithic archaeological contexts across Western Eurasia based on both mitochondrial (diamond) and nuclear (square) DNA."

Inconsistencies in formatting of dates have been corrected across all figures, with the mean radiocarbon estimate in years calibrated before present (calBP) used in all instances. Age estimates based on contextual dates have been presented in years before present (BP), prefixed with "c."

Figure 3

Veretye 1 and 2 are strikingly different in Figure 3, in light of the overall thorough discussion of results in the study, consider briefly discussion this observation also.

Further, consider commenting on the wolf-like ancestry in Vlasac1 and Veretye2, some readers may think that is directly Iranian wolf ancestry.

These results were due to incorrect usage of a qpAdm wrapper (https://github.com/pontusssk/qpAdm_wrapper) that only uses chromosomes 1-22 (i.e. the number of human chromosomes) for qpAdm analysis. We have modified this wrapper to analyse all 38 autosomal chromosomes present in dogs, and all analyses re-run. The new results (see Figure 3B) show that Mesolithic dogs can be modelled without the Near Eastern wolf component, and instead carry the dual East-West Eurasian ancestry expected (based on the results of other Mesolithic dogs). This discrepancy was due to the failure of rejecting more complex models in lower coverage data.

Extended Figure 1

Why is Wezmeh1 unknown when it generally is wolf throughout the manuscript and fall with wolves in the PCA? All the unknowns seem to have a CanID Status in the supplementary overview, which may be confusing?

We have changed “Unknown” to “Low-Coverage Sample” in the key in Extended Figure 1. This terminology gets around the fact that once they’re included in the PCA projection they are no longer unknown (i.e. assigned as either dogs or wolves).

Extended Figure 2

There seems to be some inconsistencies with the IDs and dates for some samples compared with the general IDs and dates in the manuscript:

TRF.04.09_Veretye_10900 (Extended Figure 2)

OL4061_Veretye_RUSSIA_A_10930 (Extended Figure 2)

Vs.

D_Veretye2_8750 (General in figures)

D_Veretye1_10930 (General in figures)

Labels for individuals used in both nuclear and mitochondrial genome analyses are now consistent in Extended Figure 2, Supplementary Figure 6, and Supplementary Data 5.

Extended Figure 2

Is W_GoughsCave3_14297 not in this tree?

We have added the following section into the figure caption to explain the exclusion of the Gough’s Cave wolf, as well as other samples that did not meet our criteria for inclusion:

“To ensure our divergence time estimates were robust, we excluded samples with no direct radiocarbon age (e.g., Mesolithic Serbian dogs), and a depth of coverage of <10x (e.g., Palaeolithic Gough’s Cave wolf; see Supplementary Table 4).”

These details have also been added to the methods.

Extended Figure 7

Consider specifying the sample(s?) in branch Pleistocene_Wolf

All labels in Extended Figure 7 are now consistent with the labels used throughout the manuscript.

Supplementary text

There seems to be an issue with line spacing in the supplementary text?

This has been corrected in the new version of the supplementary text.

Supplementary Figure 11.

It could be relevant with a version of the figure of AMY2B gene copy number, just within the timeframe of dogs.

We have added a second panel (B) that contains only ancient dogs.

Supplementary Figure 16, 17, 20 and 21.

For some readers it would be confusing looking for a black dot trying to identify significant tests. Consider modifying the figures or explaining in the figure text.

The key shows a filled and unfilled circle, which indicate significant and non-significant tests, respectively. To make this clearer, we have added the following to all relevant figure captions: "Significantly negative ($|Z| < -3$) or positive ($|Z| > 3$) values, which are shown as filled shapes, indicate an excess of allele sharing between...".

Supplementary Figure 21.

What is D_Veretye4_10930?, I cant seem to find it in the overviews.

This sample was excluded from our study due to its low coverage ($< 0.001x$). We have removed any tests that involve this sample from the Supplementary Figures.

In the supplementary reference list

"Corr, L.T., Berstan, R., Evershed, R.P., 2007. Optimisation of derivatisation procedures for the determination of $\delta^{13}C$ values of amino acids by gas chromatography/combustion/isotope ratio mass spectrometry. *Raid Commun. Mass Spectrom.* 21, 3759-3771. "

Don't seem to be a numbered reference?

Corrected.

517293_1_extended_data_4784591_sx9ms9 (Supplementary Data 1)

The overview seems incomplete, like the mitochondrial haplotype are missing for;

D_Padina1_11500-8300

D_Padina3_8300-7900

But I believe they are C based on the manuscript? Also, there are inconsistencies in font used.

Mitochondrial haplotype information has been added for all novel samples (i.e. C4 for all Mesolithic Serbian dogs). We have also ensured the font is consistent across all Supplementary Tables.

517293_1_extended_data_4784592_sxmmsm (Supplementary Data 2)

Doublecheck everything is fine in this overview, should cell P16 be empty?

Is the full information given in cell H2-H14?

All missing information has been included.

517293_1_extended_data_4784593_sxwmsw (Supplementary Data 3)

There are only 276 samples in this overview while the manuscript says >1,700.

Many of the 276 samples don't have a "BioProject" or "Accession".

As per above, we have changed this sentence to:

"These data were analysed alongside previously published ancient dog (n = 65) and wolf (n = 66) genomes spanning the last 100,000 years (Supplementary Data 1), as well as 276 modern dog, wolf, and outgroup canid genomes (which are a representative subset of 1,700 canid genomes in the NHGRI Dog Genome Project Database; Supplementary Data 3; (Plassais et al. 2019)). All missing project and sample accession codes have been added."

517293_1_extended_data_4784594_sxtmst (Supplementary Data 4)

There are inconsistencies in font and font size.

Modified.

In 4, which tree do NJ refer to?

This was a typo, and should have read "Maximum Likelihood", which corresponds to Supplementary Figure 6. This has been corrected.

Consider detailing which analyses columns A, B and C represents?

What is the difference between Sequence ID and Analysis ID?

Columns A,B, and C are now: "Included in Nuclear DNA Analyses?", "Included in Maximum Likelihood Phylogeny?", and "Included in Bayesian Phylogeny?", respectively, which should make it clearer. The ID columns have also been changed to: "Analysis ID", "Lab ID", "Archaeological ID", to make it clearer why they differ (i.e. being used by different groups – archaeologists/curators, lab technicians, and bioinformaticists).

517293_1_extended_data_4784595_sxfmsf (Supplementary Data 5)

About sample: AL3194_PortauChoix_Canada_A2b_4020 / AL3194

From: Port au Choix, Newfoundland, Canada

The studies "Ramos-Madrigal et al. 2020 / Lin et al. 2023" are cited for reference, should Ramos-Madrigal et al. 2020 be Ní Leathlobhair et al. 2018?

Corrected.

517293_1_extended_data_4784596_sxpmisp (Supplementary Data 6)

There are inconsistencies in font and font size.

In one case "et al." is in italics, in the other case it is not.

Also, there is a comment saying "Value's don't match Mikes! - Oliver Craig"

If the comment refers to a still existing issue, please consider that.

Corrected. These issues were resolved before manuscript submission, so the comment should have been accepted.

Consider using a consistent format for literature references given in the Supplementary Data tables.

We have modified all references to the same format [1st_author,|et al.|pub_year] - e.g. Thalmann, et al. 2013.

Referee #2 (Remarks to the Author):

The manuscript "Dogs were widely distributed in Western Eurasia during the Palaeolithic" by William et al. presents genomic data from Pınarbaşı (Türkiye), Gough's Cave (UK), as well as two Mesolithic sites in Serbia. The study combines nuclear and mtDNA with archaeological context and stable isotope analyses on both human and dogs to assess the geographic scope of early dog domestication. The study contributes significantly to the ongoing debate on the origins and dispersal of early dogs by providing genetic evidence of dog presence in West Eurasia during the paleolithic. The results are novel and well supported by the data. Early dogs from Gough's Cave and Pınarbaşı site demonstrate their phylogenetic placement within domestic dog lineages, and spread alongside the expansion of human ancestry associated with the Epigravettian culture. The data also reveal long-term genetic continuity of dog ancestry in Europe and shows that Palaeolithic and Mesolithic hunter-gatherer migrations and introgression of local wolf DNA in the Near East, offering insight into regional dynamics of domestication and population interaction.

I have a few questions as well as suggestions for improvement, which don't impact any of the major claims of the paper:

1. The assertion that the dog individuals in this study represent domestic dogs rather than wolves is central to this study. However, this claim would be strengthened by a more explicit discussion of the morphological characteristics of these animals, or whether they align more closely with dogs or wolves. In addition to the genetic evidence, a comparative analysis of morphological metrics or a discussion of relevant archaeological interpretations would strengthen this point.

Whilst morphometric analysis has been used previously to differentiate dogs and wolves, in the Palaeolithic a number of putative dogs identified using these methods have been shown to be wolves based on ancient DNA analysis (both mitochondrial and nuclear). We highlight this in the introduction. We agree that a more explicit description as to why Gough Cave and Pinarbasi individuals were thought to be dogs would be useful here. We have therefore added the following text to the SI (lines 96-100):

“The masseteric fossa of both hemi-mandibles are perforated by sub-circular openings. These perforations are ancient in origin and were created by carefully placed percussion blows. This specimen has previously been classified as a domestic dog from morphometric analysis given its reduced size compared with Palaeolithic and modern wolves.”

And in the main text (lines 130-133):

and canid remains from a Late Upper Palaeolithic Magdalenian horizon (15,100–14,200 years calBP) at Gough’s Cave in the British Isles (Supplementary Figure 2; ^{1,2}), that show post-mortem treatment mirroring the human remains at the site ^{1,19}.

Please also note that at Pinarbasi, the material is too fragmentary for morphometric analysis.

2. Please consider how the current findings contribute to the ongoing debate over single versus multiple domestication centers for dogs. If this finding rule out or supports the dural-origin hypothesis previously proposed?

We have edited the section of the manuscript focusing on these hypotheses (see lines 393-398):

Overall, our results strongly indicate that Neolithic (and later) Near Eastern and African dogs possess more Near Eastern wolf ancestry than Palaeolithic and Mesolithic Western Eurasian dogs. Though it remains possible that this wolf-related ancestry was the result of an independent domestication process^{27,28}, it is more likely to result from geographically and temporally restricted gene flow between Neolithic and post-Neolithic Western Eurasian dogs and Near Eastern wolves.

3. The phrase "widespread across Western Eurasia" might be reconsidered, as the current sample size and distributions are limited to specific regions. Including this limitation would improve the precision of this study.

We agree, and have modified the term to "widely distributed" across the manuscript, and change our usage of West Eurasia to "Europe and Anatolia" where appropriate.

4. It would be helpful to report imputation metrics and genotype uncertainty, especially for ancient genomes with variable coverage. A brief table in the supplementary Data (e.g., imputation r^2) would improve transparency.

The validation analysis requested by the reviewer was already conducted in a new study showing that RoH are accurately detected in ancient imputed dog genomes as low as 0.5x²⁹; accepted for publication at *PNAS*). However, our reference panel does not include samples from Palaeolithic dogs, and we have not specifically tested the method's accuracy for this population. Because of this and the limited impact of the RoH analysis in the current paper, we have decided to remove imputation from this current version.

5. The authors claim genetic continuity in European dog ancestry from the Palaeolithic onward. This is an important observation in the context of previous studies reporting significant turnover during later prehistory. A short discussion on how these findings intersect with the known genetic turnovers in European dog populations during the Bronze and Iron Ages would strengthen the broader context of the study (e.g., Botigué et al., 2017; Bergström et al., 2020).

We have expanded this section to clarify how our results relate to previously suggested genetic turnovers in European dog populations (lines 431-450):

Genomic analyses of Mesolithic Hunter-Gatherers from the same contexts as several of these dogs (Iron-Gates and Veretye) have shown that they also possessed EHG genetic ancestry^{8,25}. Coupled with the positive correlation between human-dog ancestries (Extended Figure 3C), our results suggest that Eastern Eurasian dog ancestry may have been first introduced into Europe during the spread of Eastern Hunter-Gatherers during the Mesolithic⁸, rather than during the later Steppe Pastoralist migrations as previously hypothesised²⁶.

Our *qpAdm* analysis also shows that the Eastern Eurasian dog ancestry persisted in European dog populations from the Mesolithic to the present. Approximately 30% of this ancestry component is present in dogs during the Neolithic (e.g., Pločnik, Serbia), 33% during the Bronze Age (e.g., Belverde Di Cetona, Italy), 22% during the Iron Age (e.g., Parknabinnia, Ireland), 18% during the Medieval period (e.g., Vilnius Castle, Lithuania) and ~20% in modern dogs (Figure 3B). Whilst this does not preclude additional influxes of Eastern Eurasian dog ancestry into Europe post-Mesolithic,^{26,30} these findings show that the fundamental ancestry components in European dogs (i.e. both Eastern and Western Eurasian ancestry) which characterises many popular modern breeds including

the Cavalier King Charles Spaniel, German Shorthaired Pointer, Jack Russell and Labrador (Supplementary Figure 17) was established by at-least 10,900 years ago.

6. The authors report strong genetic similarity between Palaeolithic dogs from Gough's Cave and Pınarbaşı based on outgroup-f3 statistics and phylogenetic analysis. While compelling, this conclusion could be enhanced by f4-statistics or qpGraph modeling to assess whether the observed similarity reflects shared ancestry or gene flow.

The outgroup-f3 analyses showing that Gough and Pınarbaşı dogs are genetically most similar to each other was also corroborated with f4-analysis of the form $f4\{\text{Coyote, GC/Pin; Pin/GC, Dogs}\}$ (Extended Figure 3), Treemix (Extended Figure 7) and AdmixtureBayes (Figure 3A) analyses.

Minor suggestions:

1. There is inconsistency in the formatting of isotope notation (e.g., $\delta^{15}\text{N}$ vs. $\delta^{15}\text{N}$). Please standardize throughout the main and supplementary texts.

Corrected. We have used $\delta^{15}\text{N}$ in all cases.

2. The manuscript inconsistently uses both "archeological" and "archaeological" across sections, I would recommend standardizing to "archaeology" throughout the text.

Corrected. All instances have been changed to "archaeological" or "archaeology" for consistency.

3. Please clarify "Gene-flow from wolves into Near Eastern dogs", if this is widespread or just localized.

This is localised in Holocene Near Eastern dogs. We see evidence for local wolves (ie Syrian Wolf, Wezmeh Wolf) introgression with local Near Eastern dogs, resulting in the distinct dog ancestry carried by several Near Eastern dogs in the early-mid Holocene. We have modified the abstract: "we detect gene-flow between dogs and local wolves in the early Holocene", and made changes in section Near Eastern dog ancestry emerged through admixture with local wolves for additional clarity (lines 383-403).

4. Some figure legends could benefit from clearer labeling of new versus previously published samples, and the use of consistent symbols or color coding for sample categories would enhance clarity.

All novel data are now indicated with an asterisk (*). Colours, shapes, and labels have been standardised across all figures.

Referee #3 (Remarks to the Author):

In this study Marsh and colleagues attempt to better understand the dynamics of early dog domestication by sequencing ancient Paleolithic and Mesolithic genomes from Eurasia. Specifically, they raise the issue to motivate their study that understanding the evolution from wolves to dogs is made particularly difficult because it can be difficult to distinguish them in the fossil record, particularly for the oldest and potentially most informative samples which are highly fragmented/incomplete/partial. There are numerous specimens that exist from the Paleolithic and Mesolithic across a wide geographic area where it is unclear whether they represent wolves or early dogs. If we could obtain DNA from these fossils this ambiguity could potentially be resolved, leading to a more complete understanding of the process of dog domestication.

Marsh et al are able to sequence eight specimens from such contexts (though two specimens from Wezmeh Cave were found to be younger than previously thought), finding six that were clearly dogs and two which were wolves. Despite the paper being interesting, it is important to note that the study does not resolve the ultimate question about the process of dog domestication (the when, the where and the how). The findings in my opinion are incremental rather than transformative, largely pushing back slightly the time frame of two dog lineages back a few thousand years from that previously established in Bergstrom et al.(2020). Beyond that there are no major new insights into the process of dog domestication. I think the samples are valuable, as they do at least provide some constraints on the timing of dog domestication (at least 16kya), which will be interesting to those interested in the process of dog domestication.

Our paper extends the timeline for dog domestication by 5,000 years (~50%), providing the first empirical evidence that dogs were part of human life in the Palaeolithic. We also show that dogs and wolves were already mostly reproductively isolated 15,000 years ago, despite co-existing at the same sites (e.g., Gough's Cave), which highlights that substantial biological and behavioural differences had already emerged between them. Finally, we also show that distinct Palaeolithic human groups across Europe, who possessed different ancestry and culture, kept genetically similar dogs—even in cases where little to no other evidence of interaction between these groups exists.

But there are serious issues that need to be addressed in terms of some of the analysis and interpretation. I would perhaps urge the authors to simplify what they are trying to do a little bit, and in particular when presenting hypotheses, think about analytical frameworks that provide an actual mechanism to test them.

We thank the reviewer for their comments and suggestions, all of which have been addressed in our responses below.

The ancient DNA work (sequencing of the ancient genomes) is exemplary, reliable and thorough, and the conclusions drawn in the first section of the paper ("Dogs were widely dispersed across West Eurasia by 14,000 years ago") are well supported by the data. The 6

sequenced dogs include two late Paleolithic samples from Turkey and the UK. They authors also establish the canid mtDNA phylogeny context of these samples, allowing them to integrate other ancient specimens that were previously sequenced for their mtDNA (but not autosomes) such as Oberkassel, and impute that these specimens must also have been dogs. Thus this work demonstrates that dogs were domesticated and *widespread* across Eurasia at least 16kya. This is a simple but powerful conclusion that rests on the excellence of the sampling and is important for those studying the process of dog domestication. This early conclusion in the manuscript is on solid ground and vital.

We agree with the reviewer's assessment. The first section of the manuscript is indeed the most crucial, as it contains the core evidence supporting our title's claim that "Dogs were widely distributed in Western Eurasia during the Palaeolithic."

However, while the paper uses many state-of-the art population genetic methods appropriate for paleogenomic data, the paper then starts to get somewhat speculative in subsequent sections, and more work is required to support the later conclusions, which are often framed initially as testing hypotheses but are never able to provide a clear analytical framework to do so. Also, many of the methods applied are not described very clearly in the main text, and often the reader must infer what was done, either from a figure or the supplementary text.

The subsequent subsections do indeed address more subtle, albeit important, points about the mechanisms by which Palaeolithic dogs spread and their legacy in later populations. We have addressed the reviewer's comments point by point below.

I list my issues with this remaining part of the manuscript sequentially below.

High degree of similarity between Palaeolithic dogs across West Eurasia

This section draws attention to the fact that the two Paleolithic samples from Gough Cave and Pınarbaşı appear to be genetically very similar despite being quite distant geographically. This certainly is interesting and notable, but the analysis to examine this is somewhat confusing and convoluted. Some clarification is needed about the analysis and why it was conducted and how it was.

More importantly, the authors advance a hypothesis that is not supported by the data regarding an Epigravettian-associated expansion.

We have responded extensively to this comment below in section High degree of similarity between Palaeolithic dogs across Western Eurasia of the main text (lines 208-283) and page 2 earlier in this document.

Line 207: "Further, kinship and pairwise outgroup-f3 analyses (Figure 2A; Extended Figures 3 & 4; Supplementary Figure 7-8) indicate close genetic similarity between the dogs at Gough's Cave and Pınarbaşı"

So what was the kinship estimate from READ (I cannot see it in the supplementary either)? The purpose of the outgroup-f3 analysis in Figure 2A to assess kinship while also correlating with geographic distance is not very clear and left to the reader to figure out. It seems a very imprecise and convoluted way to test for kinship with so many other methods available such as KIN. And then no statement of how related the samples are in terms of degree is ever given. My assumption is that the purpose of the outgroup-f3 is to show that the resulting value is much higher than expected for the two Paleolithic samples given their relative geographic distance but this is never actually stated in the text.

We thank the reviewer for their feedback. We have now focused on outgroup-f3 statistics rather than kinship, which is robust to variations in sequencing coverage. We have also revised the text to clarify our primary hypothesis: that the observed genetic similarity between these ancient dogs from distant locations (i.e. Pinarbasi and Gough's Cave) supports a rapid dispersal of dogs across Western Eurasia, without replacement of human ancestry/culture (e.g., Epigravettian, Anatolian Hunter-Gatherer, and Magdalenian).

To test this hypothesis, we expanded our analyses. First, we have expanded our analysis comparing genetic similarity (outgroup-f3) and geographic distance, this time including all pairs of ancient dogs in our dataset. We used Multiple Regression on Distance Matrices to determine the influence of spatial and temporal distance on pairwise outgroup-f3 values. While the Pinarbasi and Gough Cave dogs show high genetic similarity for their geographic separation, this deviation was not statistically significant when including all pairs of dogs, indicating that their similarity does not represent a significant departure from the general trend (Extended Figure 3).

To determine if the genetic similarity between the Pinarbasi and Gough Cave dogs was unusual relative to the ancestry of their associated human populations, we compared outgroup-f3 statistics between human and dog pairs from the same archaeological sites/contexts (Supplementary Data 7). We first showed that outgroup-f3 statistics between all co-located human and dog pairs were strongly positively correlated (Mantel $r \approx 0.40$, $p < 0.0002$). This demonstrates that, in general, human and dog ancestries at a given site are strongly associated.

To directly compare the genetic similarity between human and dog pairs, we standardized the values for each species and then subtracted the standardised dog outgroup-f3 to the human outgroup-f3 (Extended Figure 3). Negative values imply a closer genetic relationship between the dogs than between the humans from a given pair of archaeological sites, while positive values imply the opposite. This analysis reveals that the Gough Cave and Pinarbasi dogs are more related to each other than would be expected based on the ancestry of the humans at their respective sites (Extended Figure 3). These results indicate that dogs during the Palaeolithic were likely exchanged between genetically distinct human communities, such as Magdalenian, Epigravettian, and Anatolian hunter-gatherer populations.

Outgroup-f3 statistics are also measures of similarity and in some parts of the paper are treated as such but in Figure 2a, Extended Figure 3, and their respective descriptions, the outgroup-f3 statistics are labeled as genetic distances.

Furthermore, from my understanding the vertical axis in Figure 2a labeled as “Genetic Distance (outgroup-f3)” shows outgroup-f3 values with the higher value denoting higher similarity (as would be expected). In this plot, similarity or shared drift as measured by the outgroup-f3 statistic decreases with geographic distance. The description for the figure describes a positive linear relationship between genetic distance and geographic distance, but the plot in Figure 2a shows a negative linear relationship between similarity/shared drift and geographic distance. This makes sense but as described and plotted is not as clear as it could be. The purpose and conclusion of this analysis should be stated in the text, and the result is interesting enough that it warrants attention. Especially as it is relevant to analyses later in the paper.

We apologise for this typo, it should indeed read genetic similarity - this figure has now been moved to an extended figure and the axis and figure legend clarified (Extended Figure 3).

Line 212: Estimating ROH for low coverage ancient samples imputed from modern samples seems highly problematic. There are many points of potential error that could occur. The older the sample, the worse the imputation is likely to be. Similarly their lower coverage will compound the issue. It may lead to both false positives and negatives. Some validation is necessary to demonstrate the results are reliable, for example comparing to results for the high coverage NGD sample. I would also strongly recommend the authors look into utilizing HapRoH (Ringbauer et al. Nat Com 2021).

The validation analysis requested by the reviewer was already conducted in a new study by our group showing that RoH are accurately detected in ancient imputed dog genomes as low as 0.5x⁽²⁹⁾; accepted for publication at *PNAS*). However, our reference panel does not include samples from Palaeolithic dogs, and we have not specifically tested the method's accuracy for this population. Because of this and the limited impact of the RoH analysis in the current paper (only one sentence), we have decided to remove it from this current version. The suggestion to use HapRoH is excellent but beyond the scope of this paper.

Line 224: “We hypothesize, therefore, that the dispersal of dogs throughout West Eurasia may be linked to the expansion of Epigravettian-associated ancestry and culture seen across Europe after the Last Glacial Maximum between 19,500–13,000 years ago. This timing is also coincident with the most recent common ancestor of dogs in the C haplogroup, estimated to be ~18,900 years ago (95% HPD: 22,989–16,125 years ago).”

There is no reason for these things (the spread of the culture and the mtDNA TMRCA) to be related, unless the implication is that Epigravettian spreading individuals bred one dog strictly to have a single mtDNA lineage, and then brought this lineage of dogs with them everywhere they went, and thus there may be some strong correlation in mtDNA and genome-wide TMRCA.

While we agree that single loci (e.g. mtDNA) TMRCA do not provide a good estimate of population splits, they provide a good upper bound as they are informative about the time by which an ancestor of two individuals lived. In fact the mtDNA TMRCA can only be older than the population split or contemporary with post divergence gene-flow. In addition, radiocarbon dates provide lower bounds for the population split / gene flow event.

In this case, the mtDNA TMRCA (18,569-15,860 years ago) for the Gough and Pınarbaşı dogs shows that one of their common female ancestors lived during the early Epigravettian period. The fact that this TMRCA is so recent, compared to the dogs' radiocarbon age (14,808–14,091 years calBP), suggests that their female shared ancestors lived only a few thousand years before them. This is valuable information which establishes temporal constraints for the split between their source populations or for a significant gene flow event coinciding with the dispersal of dogs by people. However, we do agree that the previous version of the text was too speculative and did not explain our points clearly and we have substantially re-wrote this section to make this point clearer (lines 218-286):

Combined, our mtDNA and nuclear results suggest the Pınarbaşı and Gough's Cave individuals at the eastern and western extremes of Palaeolithic dog distribution were genetically very similar, and were members of a population that expanded across Western Eurasia between 18,500 and 14,000 years ago.

Of the five Palaeolithic dogs identified (Figure 2A), each are associated with one of three genetically and culturally distinct human hunter-gatherer populations found across Europe and Anatolia in the Late Upper Palaeolithic: the Magdalenians (Gough's Cave^{1,2}), the Epigravettians (Bonn-Oberkassel, Kesslerloch and Grotta Paglicci^{3,4,5}), and Anatolian Hunter-Gatherers (Pınarbaşı⁶). The spread of these dogs across the region is likely to have been linked with the migration, dispersal and interaction of humans associated with these Palaeolithic cultures. In fact, while most Palaeolithic dogs in this study are associated with human populations of Epigravettian-related ancestry, including Anatolian Hunter-Gatherers at Pınarbaşı⁷, the dog from Gough's Cave was recovered from a depositional context alongside humans with Magdalenian-associated ancestry.

We assessed whether dogs were exchanged between genetically distinct human groups by comparing the genetic similarity of dogs from Pınarbaşı and Gough's Cave to that of humans at each site. We first established the degree to which humans and dog ancestries from the same archaeological contexts, across 35 sites that spanned the Late Upper Palaeolithic to medieval period (e.g., Vilnius Castle, Lithuania), were correlated using outgroup-f3 statistics (Extended Figure 3C, Extended Data 7). We found a strong positive correlation (Mantel $r \approx 0.40$, $p < 0.0002$) between human-human and dog-dog outgroup-f3 values, which indicates the human and dog ancestries are closely linked. This analysis shows the dogs from the Palaeolithic sites of Pınarbaşı and Gough's Cave were more genetically similar than expected when compared to humans at the site (Extended Figure 3D). This provides further evidence that a relatively homogeneous dog

population spread between genetically and culturally distinct human populations across Europe and Anatolia in the Late Upper Palaeolithic.

One plausible scenario is that the dispersal of these Palaeolithic dogs was coupled with the spread of Epigravettian-associated ancestry and material culture ~16,000 years ago, which, after a period of interaction, eventually replaced the previously predominant Magdalenian culture in Northern Europe^{8–10}. This timeframe aligns well with the mtDNA-based TMRCA of the Gough's Cave and Pınarbaşı dogs (95% HPD: 18,569–15,860 years ago), which provides both an upper bound for the divergence of their ancestral populations, and a minimum age associated with the youngest radiocarbon date (i.e. Gough's Cave dog: 14,808–14,091 calBP; Supplementary Data 4). Crucially, this timeframe postdates with the earlier divergence of Magdalenian and Epigravettian human populations, which most likely occurred during or prior to the Last Glacial Maximum (~24,000–21,000 years ago;^{8,11}). The spread of Palaeolithic dogs across Western Eurasia, therefore, likely occurred after the divergence of these two distinct human populations.

The Gough's Cave dog dates to the Late Magdalenian (from 15,000 years ago; Figure 2B), a period for which there is as yet no evidence of humans carrying Epigravettian-associated ancestry in the British Isles¹⁰. This period, however, is characterized by transitions in lithic technology (e.g., *Federmessergroupen*) that are associated with people carrying Epigravettian-related ancestry on the European continent (e.g., at Bonn-Oberkassel)^{12–15}, whilst later individuals from the British Isles (e.g., Kendrick's Cave: 13,780–13,354 years calBP) do carry Epigravettian-associated ancestry^{8,10}. Further, the putative presence of dogs (based on short mtDNA fragments) has also been suggested at another Magdalenian context in Spain (Erralla Cave, 17,410–17,096 years cal BP;¹⁶) and postdates the earliest evidence of Epigravettian ancestry in the region by over 1,000 years (El Mirón, ~18,700 years calBP;^{17,18}).

Combined, our results suggest that dogs dispersed across Late Palaeolithic Europe alongside the expansion of Epigravettian-associated ancestry and material culture over 16,000 years ago. Under this scenario, Magdalenian humans acquired dogs through interactions with Epigravettians. These interactions did not leave any signal of Epigravettian ancestry in the Magdalenian humans at Gough's Cave¹⁰, implying that the exchange of dogs between Palaeolithic human groups was not always accompanied by detectable gene flow.

In addition, the following paragraph on the UK dogs being older than Epigravettian culture in the region seems to pretty much suggest that an Epigravettian spread was not involved.

We agree that our previous description of the Magdalenian to Epigravettian transition in the manuscript was not fully developed. While the humans contemporary with the Gough Cave dog possessed exclusively Magdalenian ancestry and the material culture at the cave was clearly of

the Magdalenian industry, there is evidence of transitioning in material cultures in the UK between 14,300–13,00 years BP (e.g. Lynx Cave, Gough's Cave and Cresswell Crags; ^{12–14}). At Bonn-Oberkassel, the transition to the Federmessgruppen industry is associated with Epigravettian-associated human ancestry. As such, the transitional period in lithic industries in the UK may be indicative of Magdalenian populations in-contact with Epigravettian communities, both in the UK or along the northern coast of Europe. We have added more text to more clearly state our arguments (see comment above and summary below).

Indeed this sample seems to very much falsify the hypothesis proposed in the previous paragraph, so it seems a little convoluted. There are two Paleolithic ancient dog samples. One sample exists with the Epigravettian culture, and one does not. There is not enough evidence here to link to a cultural-associated spread. These two paragraphs (lines 222 and 231) and how they seem to conflict need some rethinking. This is brought up again in the conclusion (lines 409 – 416) and it's suggested that the similarity of these two paleolithic dogs may be associated with the expansion of Epigravettian culture, prior to stating the UK dog predates the arrival of this culture. It is caveated by stating that it may suggest "the existence of interactions between culturally-distinct human populations regardless of gene flow". While what "regardless of gene flow" means is not clear and these ideas require some thought, this may be a conclusion that is better supported by the data.

We have revised the text which now better articulates arguments for an Epigravettian-linked spread of domestic dogs throughout Europe (see lines 218-283 and above on pages 2 and 22 of this document). This argument is based on three key points:

- The majority of Palaeolithic dogs are in direct depositional association with humans that carry some degree of Epigravettian culture and/or Epigravettian-associated ancestry (i.e. Pinarbasi, Grotte Paglicci, Bonn-Oberkassel and Kesslerloch). The exceptions to this rule are where dogs are associated with Magdalenian material culture (i.e. Gough's Cave and putatively at Erralla and Le Morin), although at Erralla and Le Morin, no human genetic analysis has been performed.
- The mtDNA TMRCA of Palaeolithic dogs (18,569-15,860 years ago) and their radiocarbon dates (14,808–14,091 years calBP) coincides with the emergence of Epigravettian-associated material culture and ancestry across the continent (at 16,000-13,000 years ago). As we argued above, while a single-locus (i.e. mitogenome) TMRCA is not a precise measure of population splits, it provides a useful upper bound, while the radiocarbon date provides a useful lower bound, both of which are informative about the timeline of their dispersal.
- Although dogs appear in earlier non-Epigravettian contexts/associated with humans that do not carry Epigravettian-associated ancestry (e.g., Magdalenian at Gough Cave), direct radiocarbon dates coincide (14,808–14,091 years calBP) with a period where lithic industries in Northern Europe and the UK were transitioning towards a terminal Upper Palaeolithic form (e.g. the Federmessgruppen, Cresswellian and Hamburgian

industries). This suggests the exchange of dogs may have been part of a broader set of interactions between humans carrying Epigravettian and Magdalenian ancestry that took place during this time period, but that did not leave any genetic trace in the Magdalenian human population at Gough's Cave.

Close associations between Palaeolithic dogs and humans

The description of post-mortem modifications in the two late Paleolithic dogs is very interesting and valuable. However, though it is important data and justified by the introduction's description of how it has been used to distinguish dogs vs wolf in the past (line 103), some of the interpretation of the isotope work is questionable.

This statement is based on initial investigation of the mandible, which showed modifications similar to that seen on the human remains, we have clarified this in the supplementary. We have added a further description of the element in the SI to clarify why this specimens was initially deemed to be a domestic dog (lines 96-100):

“The masseteric fossa of both hemi-mandibles are perforated by sub-circular openings. These perforations are ancient in origin and were created by carefully placed percussion blows. This specimen has previously been classified as a domestic dog from morphometric analysis given its reduced size compared with Palaeolithic and modern wolves.”

And in the main text (lines 130-134):

and canid remains from a Late Upper Palaeolithic Magdalenian horizon (15,100–14,200 years calBP) at Gough's Cave in the British Isles (Supplementary Figure 2; ^{1,2}), that show post-mortem treatment mirroring the human remains at the site ^{1,19}.

Line 253: Entire paragraph.

While the comparison of human and dog isotopes might be a good test in principle in terms of looking for dietary overlap, the fact that wolf has the same signature negates the result, and no conclusion can be reached. It is a bit misleading to present dog and human as showing significant overlap first, as if this has some meaning, and then introduce the wolf caveat. Please rephrase. To me a better take away from the isotope work is that caution should be applied when attempting to distinguish dogs and wolves in the paleolithic from dietary isotopes, and that paleogenomic analysis is likely the gold standard to work this out.

We agree with the reviewer and revised the text to reflect the fact that isotopic data may not be useful to distinguish dog and wolf diets in the Palaeolithic. We have also added more information about the aquatic diet of dogs and people at Pinarbasii, which we suggest that dogs were fed, by people, with 'net-caught' small cyprinids given their large abundance at the site. This section now reads (lines 305-323):

To investigate whether close associations were also evident during the lifetime of the dogs, we tested for shared dietary signatures between canids and humans at both Gough's Cave and Pınarbaşı through the measurement of bulk and amino-acid specific $\delta^{13}\text{C}$ and $\delta^{15}\text{N}$ values from bone collagen (Figure 2D, Supplementary Figure 9; ³¹). At Gough's Cave, the dogs and humans showed significant overlap in $\delta^{15}\text{N}_{\text{Glu}}$ and $\delta^{15}\text{N}_{\text{Phe}}$ values, which are indicative of an omnivorous terrestrial diet (estimated using the $\delta^{15}\text{N}_{\text{Glu-Phe}}$ proxy for trophic position; ^{32,33}). Interestingly, the Gough's Cave wolf showed a similar trophic position, which suggests that the proposed niche partitioning between dogs and wolves (e.g., ³⁴) may not have been as pronounced in Late Upper Palaeolithic Europe as in specific periods and places (e.g., adaptation to starch-rich diets after the Neolithic; Supplementary Figure 10; ³⁵). Because different diets may produce similar isotopic signatures, however, our results may not necessarily discriminate between wild and domestic populations.

At Pınarbaşı, although the trophic level of dogs and humans did not directly overlap, they were both elevated relative to Gough's Cave, and are consistent with an aquatic dietary component (Figure 2D). The remains of small freshwater fish, likely net-caught, are common in the human-occupied layers at Pınarbaşı ⁶, suggesting that dogs were either being directly or in-directly provisioned by humans.

Line 265: Entire paragraph.

Again, this paragraph seems to be reaching for meaning when there is not one. It seems a stretch to suggest that humans were feeding dogs based on just being elevated compared to Gough's cave.

Please see above.

Palaeolithic dogs from Gough's Cave and Pınarbaşı belonged to the West Eurasian lineage. While in principle I do not disagree with the final conclusion as stated in the subtitle, there is some uncertainty in how it is being established and how the hypothesis of being an extinct versus existing lineage is actually being tested.

Below we describe specific improvements we have made to the manuscript to address this comment.

Line 282: "To establish the ancestry of Palaeolithic and Mesolithic European dogs, we calculated shared drift with representatives of Eastern (a ~9,500 years calBP Siberian Hunter-Gatherer dog from Zhokhov Island; 45) and Western (a ~5,800 years calBP dog from the Neolithic Iranian site of Tepe Ghela Gap; 29) dog lineages using outgroup-f3 statistics of the form: $f3(\text{Coyote}, \text{D_TepeGhelaGap1_5826}, \text{D_Zhokhov1_9515})$." It is unclear what is being done here based on the main text, as the outgroup test the authors mention is just one formulation of the test. Do the authors mean conducting two forms of the test like so, $f3(\text{Coyote}, \text{D_TepeGhelaGap1_5826}, \text{X})$ and $f3(\text{Coyote}, \text{D_Zhokhov1_9515}, \text{X})$, rotating through the

ancient genomes via X? This seems to be implied by Extended Figure 6, and this same setup is described in Extended Figure 3. If so, please put in the main text, not just let us work it out from the extended figures.

We apologize for the confusion, the reviewer is correct - these are the outgroup-f3 we computed. We have made this more explicit in the text (lines 330-336):

To establish the ancestry of Palaeolithic and Mesolithic European dogs, we calculated shared drift with representatives of Eastern (a ~9,500 years calBP Siberian hunter-gatherer dog from Zhokhov Island; ³⁶) and Western (a ~5,800 years calBP dog from the Neolithic Iranian site of Tepe Ghela Gap; ³⁰) dog lineages using outgroup-f3 statistics of the form: $f_3(\text{Coyote}, D_TepeGhelaGap1_5826/D_Zhokhov1_9515, X)$.

Line 289: "Palaeolithic dogs from Europe and Anatolia share more drift with the West Eurasian lineage (Extended Figures 3 & 6), which is also apparent under PCA (Figure 1C; Supplementary Figure 4). These results indicate that Palaeolithic dogs do not belong to an extinct lineage, but instead form part of the West Eurasian lineage, which indicates that the divergence of East-West dog populations must have occurred by at least 16,000 years ago."

The analytical framework here is not clear. What result would have indicated that the Paleolithic dogs were an extinct lineage in this test? Would they have been along the diagonal of Extended Figure 6? Would they have less shared drift, situated along the diagonal and towards the bottom left? How would the authors distinguish this departure beyond statistical noise? The two Paleolithic samples are also not labeled in the relevant panels in Extended Figure 3 (and in Ext. Fig 3D the Pinarbasi dog is covered by a label), so it is difficult to assess the results there. The expectations of the test and hypotheses need to be a little bit more defined otherwise a definitive statement as presented in the subtitle cannot be made.

We agree that the term extinct lineage was poorly chosen in this context, we have removed reference to this term. Our analysis of shared genetic drift reveals that the Paleolithic dogs share far greater genetic affinity with the Western dog lineage, beyond statistical noise (Extended Figure 5 [A] and Supplementary Figure 11 [B]).

[A] Extended Figure 5 – The affinities of ancient dogs to Near Eastern and Arctic dog lineages: Levels of shared drift (*outgroup-f3*) between ancient or modern dogs and an ancient Near Eastern dog (Iran_5826; x-axis) and an Arctic dog (Zhokhov_9515; y-axis). Samples are coloured based on both shared ancestry and geographic proximity: African/Near Eastern (green squares), European (orange diamonds), Arctic (blue triangles), and East Asia (yellow triangles).

[B] Supplementary Figure 11. *D*-statistics of the form: $D(\text{Coyote}, X; D_Zhokhov1_9515, D_TepeGhelaGap1_5826)$ and: $D(\text{Coyote}, X; W_BeleyaGora1_18148, \text{Wolf56})$. Fill colour corresponds to test significance ($|Z| < 3$; see figure legend). Pınarbaşı and Gough's Cave show significant allele sharing with both TepeGhelaGap and BeleyaGora, which indicates West Eurasian dog ancestry, but not the secondary gene flow from Near Eastern wolves (exemplified here by Tel Hreiz).

This finding indicates that they lived after the Eastern and Western dog populations diverged, providing a reliable minimum age for this population split. We have clarified this in the main text (lines 333-339):

Palaeolithic dogs from Europe and Anatolia share more drift with the Western Eurasian lineage (Extended Figures 3–5), which is also apparent from PCA (Figure 1C; Supplementary Figure 4), and *D*-statistics of the form $D(\text{Coyote}, X, D_Zhokhov1_9515, D_TepeGhelaGap1_5826)$; Supplementary Figure 11). These results indicate that Palaeolithic dogs form part of the Western Eurasian dog lineage, which pushes the divergence of Eastern-Western dog populations prior to 15,800 years ago.

Near Eastern dog ancestry emerged through admixture with local wolves
Again, the authors present competing hypotheses here (post-domestication introgression versus dual domestication, line 298), and while doing many complex analyses, do not make clear how they are distinguishing the two hypotheses. The end result is a conclusion that both hypotheses are still possible, so I am not sure of any real advance here. In addition the results of multiple descriptive analyses are presented that seemingly contradict each other. A clearer analytical framework is needed if the authors are going to contrast these hypotheses.

This section was intended to engage with a previous paper that posed these two hypotheses^{28,30}. However, the reviewer is correct that our analyses do not have the power to formally address these. For instance, we could not reject the dual origin hypothesis even if the Pınarbaşı and Gough Cave dogs lacked Western wolf ancestry, as they may have lost this ancestry over time through backcrossing with other dogs, or simply never possessed it to begin with.

Similarly, the fact that both Palaeolithic dogs do possess some degree Western wolf ancestry (although less than other Near Eastern wolves) does not necessarily mean that all Western dogs acquired this ancestry through a secondary domestication process of a Western wolf progenitor as we cannot rule out the possibility that this ancestry was acquired through one or more instances of gene flow from wolves.

Although we agree with the reviewer that our data do not provide a way to test these hypotheses directly (and it is debatable whether any data truly can), we think that it is necessary to engage with this previous literature. We have therefore amended this section to better articulate which specific question we can actually test, i.e. do our newly sequenced dogs possess more or less Near Eastern wolf ancestry than later Western Eurasian dogs? We then provide an interpretation of these results with respect to the two hypotheses (lines 393-398):

Overall, our results show that Neolithic (and later) Near Eastern and African dogs possess more Near Eastern wolf ancestry than Palaeolithic and Mesolithic Western Eurasian dogs. Though it remains possible that this wolf-related ancestry was the result of an independent domestication process^{27,28}, it is more likely to result from geographically and temporally restricted gene flow between Neolithic and post-Neolithic Western Eurasian dogs and Near Eastern wolves.

Line 331: “The highest levels of wolf ancestry (~19%) were identified in a 7,000-year-old dog from Tel Hreiz (Israel).”

In Figure 3B there are higher levels of wolf ancestry estimated for the Veretye 2 sample (Russia) and the Vlasic 1 sample (Serbia), and the Gough’s Cave sample seems to be modeled at roughly the same proportions as the Tel Hreiz dog.

This 19% seems to be drawn from the f4-ratios shown in Extended Figure 8B, which does not include these two Mesolithic samples or the Paleolithic dog from Gough’s Cave. The ADMIXTURE analysis shown in Supplementary Figure 5 includes all of these samples, and shows similar proportions of wolf ancestry in the Tel Hreiz dog and the Gough’s Cave dog, but none in the Mesolithic samples mentioned above.

The high level of wolf ancestry in Tel Hreiz was indeed drawn from the F4-ratios. We have now included D-statistics and F4-ratio tests for both the Serbian and Karelian Mesolithic dogs, which have been added to Extended Figure 6B (see below):

Extended Figure 6 - (B) F4-ratio showing proportion of ancestry in Near Eastern dogs derived through gene flow with local wolves (i.e., W_Wezmeh1_2708). Alpha values for all Mesolithic European dogs (i.e. Veretye, Padina, Vlasac) exceeded 1, so the Near Eastern wolf ancestry proportion was set to 0.

Line 334: “In contrast, high levels of wolf ancestry are maintained in the modern Basenji breed (14–18%)...”

I only see a very small amount of wolf ancestry in Figure 3B for Basenji05, certainly not 15%. There is a discrepancy here with the f4-ratios in Extended Figure 8B, which do show the higher proportions of wolf ancestry described (analyses in Fig. 3B and Ext. Fig. 8B use the same Wezmeh wolf). How is this discrepancy explained? Basenji do not seem to be included in the ADMIXTURE analysis shown in Supplementary Figure 5, so that cannot be compared. Furthermore, there is less of a discrepancy with the wolf proportions estimated in the Tel Hreiz sample, which is included in both of the same analyses (~12% in Fig. 3B and ~19% in Ext. Fig. 8B).

We have added additional information of both f4-ratio and ADMIXTURE based ancestry proportion in this section. There are many reasons why the exact values for ancestry proportion might be slightly different across analyses, including the fact that the Zhokhov dog, used as a sister population to the source population (Pinarbasi) in the f4-ratio and as source in qpAdm,

possibility possess some Pleistocene wolf ancestry, which could affect these analyses in different ways.

Line 340: “When allowing for additional admixture edges ($K = 4$; Extended Figure 7E)...”
What was the justification for allowing additional admixture edges? Were there clear parts of the distance matrix with large residuals using 0 or 1 admixture edges? Treemix will put however many edges you ask, but that does not mean it will improve model fit beyond over-fitting. I also see a number of suspicious edges with Coyote in Extended Figure 7 when the migration edges are increased above 1.

We plotted the residual matrix for each iteration (Supplementary Figure 15). As expected the residuals matrix show attraction between population that are more closely related than represented in the tree - for example, without admixture edge, the highest residual is between the ancient Iranian dog (Tepe Gela Gap), the Basenji, and the Wezmeh wolf, consistent with the admixture described in our previous comments. The admixture edges from Coyote to Gough and Basenji are likely due to their basal ancestry from lineages not included in this analysis. However, we agree with the reviewer that TreeMix does not offer a straightforward way to determine when adding admixture edges improves the model's fit versus when they lead to overfitting. This is why we performed an ADMIXTUREBAYES analysis, which allows for a formal comparison between these models. Notably, the first admixture edge inferred by TreeMix is also

strongly supported by the ADMIXTUREBAYES analysis and other tests, including D-statistics.

Supplementary Figure 15. Residuals estimated in TreeMix v.1.13³⁷ after adding: (A) 0, (B) 1, (C) 2, (D) 3, (E) 4, and (F) 5 migration edges.

Lines 348-351: “While these affinities between Near Eastern wolves and West Eurasian dogs have been identified previously, the occurrence, timing and extent of gene flow from Near Eastern wolves into local dog populations could not have been identified without the Palaeolithic Pınarbaşı genome.”

I think it is important to make clear how the Pınarbaşı dog has helped identify the occurrence, timing, and extent of gene flow from Near Eastern wolves into local dog populations. This may help focus this section, since it seems some of these tests and conclusions are based around the Pınarbaşı dog having less admixture from Near Eastern wolves than other dogs from the Near East.

Line 355: “Alternatively, it remains possible that part of this ancestry is the remnant of a second, independent domestication of a Western wolf progenitor, followed by gene flow from wolves in the Near East, whose genetic contribution to dog populations in the region decreased over time”.

Where is the evidence for this hypothesis from the results? This possibility seems to come out of nowhere, despite all the tests showing somewhat inconsistent evidence of admixture from Near Eastern wolves (in the sense that there was certainly evidence of Near Eastern wolf admixture into some early dogs, but which dogs show it and their admixture proportions shift when using different types of analyses). Given the premise that was set up at the beginning of the section, how did the authors try to distinguish between the dual domestication and multiple introgression event

hypotheses? This has not been clearly laid out and is all rather ad hoc and speculative.

These two comments relate to the section on Near Eastern wolf admixture in dogs. We have substantially edited this section (lines 385-398) and made reference to these changes on pages 8 15, and 17 of this document.

The genomic legacy of West Eurasian Palaeolithic dogs

Figure 3B, Line 375: “In contrast, Southern European Mesolithic dogs from the Iron-Gates in Serbia (Padina and Vlasac) were best modelled as a combination of Western (~58%) and Eastern (~42%) dog ancestry”.

From Figure 3B, the only Serbian Mesolithic dog that seems to have proportions near those described above are Padina1; Padina2 has much less East Eurasian ancestry (~27%), while Vlasac1 is modeled as having no Western Eurasian ancestry and instead ~25% Near Eastern wolf and ~75% Eastern Eurasian ancestry. It is not clear in the text which exact analysis this result is referencing or the corresponding figure but I assume this is based on the qpAdm results.

The previously published Veretye samples are also included, with Veretye1 having near ~50/50 proportions and Veretye2 modeled almost exactly the same as Vlasac1, with ~25% Near Eastern wolf and ~75% Eastern Eurasian ancestry. Both of Veretye samples are from the same context, and previous studies have modeled their ancestry as similar to each other, so this result is confusing and requires substantiation. As mentioned previously, the Veretye2 and Vlasac1 samples also have the highest Near Eastern Wolf proportions of any of the dogs shown in Figure 3B, which is discordant with other analyses presented in this study, and is not

addressed (as far as I can tell) in any of the hypotheses or results concerning Near Eastern wolf ancestry. The three dogs from Serbia included in this are low coverage, and substantially lower coverage than the two Veretye dogs (especially Veretye2). A simple difference in coverage may also be difficult to reconcile, since Veretye2 and Vlasac1 have very similar results despite being >5x and 0.1x coverage respectively.

The main text presents results such as these and other qpAdm/qpWave results as very clean and clear cut, but a reading of the figures and supplementary shows a much noisier picture with a lot of inconsistencies. This needs to be addressed in the manuscript and not buried elsewhere.

We have identified an issue in our *qpAdm* analysis which adversely results in non-rejection of models in case of very low coverage samples (see page 9 of this document). After rerunning analysis, results are now similar to that of the previous analyses of these genomes (lines 421-440)

We have also included f4 statistics testing for wolf ancestry in Mesolithic and Neolithic dogs, in the form $D\{\text{Coyote, Wolves; Meso-Neo, Pinarbasi}\}$ (Extended Figure 6A+B). We see no attraction of wolf populations to these Meso-Neo dogs, corroborating updated *qpAdm* analysis.

Line 386: “These results suggest that the introduction of East Eurasian dog ancestry into Europe (which includes mitochondrial haplogroup A) was likely initiated by the spread of Eastern Hunter-Gatherers during the Mesolithic, rather than by later Steppe Pastoralist migrations as previously hypothesised.”

As mentioned in the paragraph above, while the dogs with the highest proportion of Eastern Eurasian ancestry are two Mesolithic dogs from Veretye (Russia) and Vlasac (Serbia), these dogs are also modeled as having ~25% Near Eastern Wolf ancestry. Other dogs from the same areas, associated with the same contexts (and the same human groups that are referenced in the text as having small proportions of Eastern Hunter-Gatherer ancestry), are instead modeled with substantially less East Eurasian ancestry and instead a greater proportion of West Eurasian ancestry (and no Near Eastern Wolf). Looking at Figure 3B, the ancestry proportions of these Mesolithic dogs without Near Eastern Wolf Ancestry (Padina1, Padina3, Veretye1) are not appreciably different from any other European dog that post-dates it. The framing of these results forward in time from the Mesolithic (e.g. referencing the Neolithic and Steppe pastoralists) is strange as the Veretye samples are previously published and the modeling of ancient European dogs (from the Mesolithic onward) with East Eurasian ancestry has been done previously, and is not surprising.

We apologize for the confusion. The *qpAdm* results for the Mesolithic dogs from Serbia have been updated in this version, and we have added background information to this section to ensure the interpretation is clearer, particularly in relation to the proportion of Eastern Eurasian dog ancestry in European dogs from the Neolithic onwards (lines 418-450):

To assess the degree of Eastern Eurasian dog ancestry in Western Eurasian dogs, we implemented a tri-source model of dog ancestry in *qpAdm* (Figure 3B; Supplementary

Figure 17), using three ancestry sources based on the results of D-statistics and admixture graphs: Palaeolithic Western Eurasian (D_Pinarbasi1_15787) and Eastern Eurasian (D_Zhokhov1_9515) dogs, and Near Eastern wolves (W_Wezmeh1_2708). As expected, the majority of ancient Near Eastern dogs possessed both Western Eurasian dog and Near Eastern wolf ancestry. The Eastern Eurasian dog ancestry component is absent until the Early Byzantine Period (1,565 years calBP at Marmara, Türkiye). In contrast, Balkan Mesolithic dogs from the Iron-Gates in Serbia (Padina and Vlasac) and Northwest Russia (Veretye) were best modelled as a combination of Western Eurasian (mean: 56.2%) and Eastern Eurasian (mean: 43.8%) dog ancestry.

Genomic analyses of Mesolithic Hunter-Gatherers from the same contexts as several of these dogs (Iron-Gates and Veretye) have shown that they also possessed EHG genetic ancestry^{8,25}. Coupled with the positive correlation between human-dog ancestries (Extended Figure 3C), our results suggest that Eastern Eurasian dog ancestry may have been first introduced into Europe during the spread of Eastern Hunter-Gatherers during the Mesolithic⁸, rather than during the later Steppe Pastoralist migrations as previously hypothesised²⁶.

Our *qpAdm* analysis also shows that the Eastern Eurasian dog ancestry persisted in European dog populations from the Mesolithic to the present. Approximately 30% of this ancestry component is present in dogs during the Neolithic (e.g., Pločnik, Serbia), 33% during the Bronze Age (e.g., Belverde Di Cetona, Italy), 22% during the Iron Age (e.g., Parknabinnia, Ireland), 18% during the Medieval period (e.g., Vilnius Castle, Lithuania) and ~20% in modern dogs (Figure 3B). Whilst this does not preclude additional influxes of Eastern Eurasian dog ancestry into Europe post-Mesolithic,^{26,30} these findings show that the fundamental ancestry components in European dogs (i.e. both Eastern and Western Eurasian ancestry) which characterises many popular modern breeds including the Cavalier King Charles Spaniel, German Shorthaired Pointer, Jack Russell and Labrador (Supplementary Figure 17) was established by at-least 10,900 years ago.

What this study does show is that by sequencing and definitively identifying two Paleolithic dogs for the first time, one of which is modeled without east Eurasian ancestry (which distinguishes it in this analysis from other ancient European dogs that come later), a case could be made that East Eurasian ancestry may have been introduced into Europe sometime between the Mesolithic samples and the Gough's Cave sample. The similarity between the Paleolithic dogs despite their geographic distance could be argued to support this despite the lack of Paleolithic samples from Russia or Serbia, and the geographic distance of the Mesolithic samples from these new Paleolithic samples. As shown earlier in the manuscript, the two Paleolithic dogs are actually more similar to each other than they are to dogs found later, despite these later dogs being found closer geographically. The authors state this themselves on lines 217-220, suggesting it may be due to the presence of Eastern Eurasian dog ancestry in ancient European dogs who come later, but do not mention it in this section.

The connection to Eastern Hunter-Gatherer ancestry in human groups is very tenuous and there is no real new evidence presented here to make the claim confidently, nor a clear reason to. This claim is repeated as if it is proven fact on lines 423-424, but the connection is not clear to me.

We agree with the reviewer that a strong case can be made for the introduction of Eastern dog ancestry into Europe during the Mesolithic. Recent studies demonstrated that this period was marked by an influx of Eastern hunter-gatherer ancestry into the East European human population^{8,38}. Notably, this ancestry was identified in humans from the same sites as the Veretye dogs in Northeast Europe on the Baltic Sea shore, and from the same context as the Vlasac and Padina dogs (Mesolithic Iron-age hunter-gatherers) in the Northern Balkans. Given that we also identify Eastern dog ancestry at both of these sites, this suggests a strong parallel between human and dog migrations. We argue that this compelling parallel provides a very plausible mechanism for the introduction of Eastern dog ancestry into Europe and is therefore worth mentioning in the text. We have revised this section to make this clearer (lines 431-437).

Lastly, the spread of mitochondrial haplogroup A is mentioned, but from what I can tell, the Mesolithic Serbian samples belong to haplogroup C (shown in Figure 1D). The previously published Mesolithic Veretye samples are haplogroup A. Does this not contradict the claim that is being made? The Serbian samples (Padina1, Padina3, Vlasac1) are also the only samples from this study who have the mitochondrial haplotype column blank in the extended data, so all I can go off of is the Figure 1D. Taken together it makes it hard to see the validity of the association of East Eurasian Dog ancestry with specifically human eastern hunter-gatherer ancestry.

The Veretye dogs possess both Western and Eastern ancestry, it is therefore unsurprising that they possess both mtDNA haplogroup that are more frequent in the West (C) and the East (A). In the case of the Serbian samples we have added their mtDNA haplogroup to Table S1 and to Figure 1D (all possess haplogroup C).

Looking at Supplementary Figure 18 it is difficult to see how these breeds are characterized by these ancestry components, since they all look indistinguishable. It seems more accurate to say that little of the structure and differentiation present in modern European breeds and breed groups was established at this point in the past, though they all seem to have some common ancestry reaching back at least 10,000 years ago.

We have updated the text in response to comments (see above and lines 445 - 450).

Minor Edits:

Figure 1: It may improve Figure 1 to include a label for the C5 clade and help clarify what is meant in the main text (lines 186- 188). As it stands it is difficult to know exactly which dogs are included in the C5 group and to connect the text to the figure.

We have added a label for the C5 haplogroup to Figure 1D. This will make it easier to discern which samples are included in this clade. We have also modified the text which introduces the C5 clade to the following: “The dogs from Gough's Cave and Pınarbaşı cluster together with other suspected European Palaeolithic dogs, sister to C haplogroup dogs, a haplogroup that we termed C5 (see Figure 1D).”

Figure 1d, Line 445-446: The description for Fig. 1d says to see Supplementary Figure 3 for the complete mitochondrial tree, but Supplementary Figure 3 seems to be a plot of calibrated radiocarbon dates.

Corrected to Supplementary Figure 6.

Line 245: “Similar post-mortem modification is evident on the dog remains including an anthropic perforation on the masseteric fossa (Figure 2B–C; Supplementary Figure 2).”
I see the picture for 2B but how does this apply to 2C?

Corrected to exclude mention of 2B, which is a plot of radiocarbon dates.

Lines 318-320: the D statistic described is not the one shown in Supplementary Figure 13. It seems to be shown in Supplementary Figure 14. This issue also appears on line 348 where Supplementary Figure 13 is referenced again incorrectly.

All references to supplementary figures are now correct.

Line 790: “Neolithic-recent” is confusing.

This paragraph has since been changed, with this term removed.

Supplementary Figure 6: The tree is difficult to read when zoomed in.

We have uploaded a higher quality TIFF file in the latest submission.

Supplementary Figure 7: What is the rationale for this? Specifically, why is the Pınarbaşı dog compared only to African/Near-Eastern dogs and the Gough's cave dog to Ancient European dogs?

These analyses have been supplemented with outgroup-f3 comparisons performed on all possible pairwise comparisons of dogs (i.e. Extended Figure 3).

Referee #4 (Remarks to the Author):

I co-reviewed this manuscript with one of the reviewers who provided the listed reports.

References

1. Jacobi, R. & Higham, T. The Later Upper Palaeolithic recolonisation of Britain: new results from AMS radiocarbon dating. in *The ancient human occupation of Britain* (eds. Ashton, N., Lewis, S. G. & Stringer, C.) 223–248 (Elsevier, London, 2011).
2. Carrant, A. P. The Late Glacial mammal fauna of Gough's Cave, Cheddar, Somerset. *Proceedings of the University of Bristol Speleological Society* **17**, 286–304 (1986).
3. Baumann, C. *et al.* A refined proposal for the origin of dogs: the case study of Gnirshöhle, a Magdalenian cave site. *Sci. Rep.* **11**, 5137 (2021).
4. Thalmann, O. *et al.* Complete mitochondrial genomes of ancient canids suggest a European origin of domestic dogs. *Science* **342**, 871–874 (2013).
5. Boschini, F. *et al.* The first evidence for Late Pleistocene dogs in Italy. *Sci. Rep.* **10**, 13313 (2020).
6. Baird, D. *et al.* Juniper smoke, skulls and wolves' tails. The Epipalaeolithic of the Anatolian plateau in its South-west Asian context; insights from Pınarbaşı. *Levantina* **45**, 175–209 (2013).
7. Feldman, M. *et al.* Late Pleistocene human genome suggests a local origin for the first farmers of central Anatolia. *Nat. Commun.* **10**, 1218 (2019).
8. Posth, C. *et al.* Palaeogenomics of upper Palaeolithic to neolithic European hunter-gatherers. *Nature* **615**, 117–126 (2023).
9. Marsh, W. A. & Bello, S. Cannibalism and burial in the late Upper Palaeolithic: Combining archaeological and genetic evidence. *Quat. Sci. Rev.* **319**, 108309 (2023).
10. Charlton, S. *et al.* Dual ancestries and ecologies of the Late Glacial Palaeolithic in Britain. *Nat. Ecol. Evol.* **6**, 1658–1668 (2022).
11. Marginedas, F. *et al.* New insights of cultural cannibalism amongst Magdalenian groups at Maszycka Cave, Poland. *Sci. Rep.* **15**, 2351 (2025).
12. Stevens, R. E., Reade, H., Tripp, J., Sayle, K. L. & Walker, E. A. Changing environment at

- the late upper Palaeolithic site of lynx cave, North Wales. Preprint at <https://doi.org/10.11588/PROPYLAEUM.950.C12581> (2021).
13. Barton, R. N. E., Jacobi, R. M., Stapert, D. & Street, M. J. The Late-glacial reoccupation of the British Isles and the Creswellian. *J. Quat. Sci.* **18**, 631–643 (2003).
 14. Jacobi, R. & Higham, T. The later upper Palaeolithic recolonisation of Britain: New results from AMS radiocarbon dating. in *Developments in Quaternary Sciences* 223–247 (Elsevier, 2011).
 15. Pedersen, J. B., Poulsen, M. E. & Riede, F. Jels 3, a new late Palaeolithic open-air site in Denmark, sheds light on the pioneer colonization of northern Europe. *J. Field Archaeol.* 1–19 (2022).
 16. Hervella, M. *et al.* The domestic dog that lived ~17,000 years ago in the Lower Magdalenian of Erralla site (Basque Country): A radiometric and genetic analysis. *J. Archaeol. Sci. Rep.* **46**, 103706 (2022).
 17. Fu, Q. *et al.* The genetic history of Ice Age Europe. *Nature* **534**, 200–205 (2016).
 18. Bortolini, E. *et al.* Early Alpine occupation backdates westward human migration in Late Glacial Europe. *Curr. Biol.* **31**, 2484–2493.e7 (2021).
 19. Bello, S. M., Saladié, P., Cáceres, I., Rodríguez-Hidalgo, A. & Parfitt, S. A. Upper Palaeolithic ritualistic cannibalism at Gough’s Cave (Somerset, UK): The human remains from head to toe. *J. Hum. Evol.* **82**, 170–189 (2015).
 20. Plassais, J. *et al.* Whole genome sequencing of canids reveals genomic regions under selection and variants influencing morphology. *Nat. Commun.* **10**, 1489 (2019).
 21. Wood, D. E., Lu, J. & Langmead, B. Improved metagenomic analysis with Kraken 2. *Genome Biol.* **20**, 257 (2019).
 22. Wingett, S. W. & Andrews, S. FastQ Screen: A tool for multi-genome mapping and quality control. *F1000Res.* **7**, 1338 (2018).
 23. Feuerborn, T. R. *et al.* Origins and diversity of Greenland’s Qimmit revealed with genomes

- of ancient and modern sled dogs. *Science* **389**, 163–168 (2025).
24. Feuerborn, T. R. *et al.* Modern Siberian dog ancestry was shaped by several thousand years of Eurasian-wide trade and human dispersal. *Proc. Natl. Acad. Sci. U. S. A.* **118**, (2021).
 25. Marchi, N. *et al.* The genomic origins of the world's first farmers. *Cell* **185**, 1842–1859.e18 (2022).
 26. Botigué, L. R. *et al.* Ancient European dog genomes reveal continuity since the Early Neolithic. *Nat. Commun.* **8**, 16082 (2017).
 27. Frantz, L. A. F. *et al.* Genomic and archaeological evidence suggest a dual origin of domestic dogs. *Science* **352**, 1228–1231 (2016).
 28. Bergström, A. *et al.* Grey wolf genomic history reveals a dual ancestry of dogs. *Nature* **607**, 313–320 (2022).
 29. Bougiouri, K. *et al.* Imputation of ancient canid genomes reveals inbreeding history over the past 10,000 years. *bioRxiv* (2024) doi:10.1101/2024.03.15.585179.
 30. Bergström, A. *et al.* Origins and genetic legacy of prehistoric dogs. *Science* **370**, 557–564 (2020).
 31. Larsen, T., Fernandes, R., Wang, Y. V. & Roberts, P. Reconstructing hominin diets with stable isotope analysis of amino acids: New perspectives and future directions. *Bioscience* **72**, 618–637 (2022).
 32. O'Connell, T. C. 'Trophic' and 'source' amino acids in trophic estimation: a likely metabolic explanation. *Oecologia* **184**, 317–326 (2017).
 33. Chikaraishi, Y. *et al.* High-resolution food webs based on nitrogen isotopic composition of amino acids. *Ecol. Evol.* **4**, 2423–2449 (2014).
 34. Baumann, C. *et al.* Dietary niche partitioning among Magdalenian canids in southwestern Germany and Switzerland. *Quat. Sci. Rev.* **227**, 106032 (2020).
 35. Ollivier, M. *et al.* Amy2B copy number variation reveals starch diet adaptations in ancient

- European dogs. *R. Soc. Open Sci.* **3**, 160449 (2016).
36. Sinding, M.-H. S. *et al.* Arctic-adapted dogs emerged at the Pleistocene-Holocene transition. *Science* **368**, 1495–1499 (2020).
 37. Pickrell, J. K. & Pritchard, J. K. Inference of population splits and mixtures from genome-wide allele frequency data. *PLoS Genet.* **8**, e1002967 (2012).
 38. Mathieson, I. *et al.* The genomic history of southeastern Europe. *Nature* **555**, 197–203 (2018).

Round 2

Referee #1 (Remarks to the Author):

Thank you very much for the effort addressing my comments, I only have a few minor points below after which I would be happy to see the work published.

Lines 142-146

“These data were analysed alongside previously published ancient dog (n = 65) and wolf (n = 66) genomes spanning the last 100,000 years (Supplementary Data 1), as well as 276 modern dog, wolf, and outgroup canid genomes (a representative subset of 1,700 genomes in the NHGRI Dog Genome Project Database; Supplementary Data 3; 20).”

I greatly appreciate the work on the supplementary material and accompanying elaborations in main text and supplementary, however I am sorry I am still confused. I cant get the numbers in the text to match the numbers in the supplementary tables. Please just double check the numbers, I am sorry if I am misunderstanding.

In Supplementary Data 1, I count;
68 dogs (excluding you 6 dogs to be released in this study)
71 wolves (excluding you 2 wolves to be released in this study)
1 dhole
So not “previously published ancient dog (n = 65) and wolf (n = 66)”

This is correct, we have fixed this issue and changed the sentence to: “...previously published ancient dog (n = 68) and wolf (n = 71)...”

Also W_GoughsCave1_14297 and W_Wezmeh1_2708 are listed as dog.

Corrected. Both are now listed as wolves.

In Supplementary Data 3, I count
199 dogs and wolves
So not “276 modern dog, wolf, and outgroup canid genomes”

This number (276 modern dog, wolf, and outgroup canid genomes) is correct as the file contains 277 rows (including the header).

Also when you state “(a representative subset of 1,700 genomes in the NHGRI Dog Genome Project Database; Supplementary Data 3; 20).” To me it sounds like there should be 1700 genomes in Supplementary Data 3. Is it important to state “a representative subset of 1,700 genomes in the NHGRI Dog Genome Project Database”? or maybe put the reference to Supplementary Data 3 after the actual number of genomes included.

This parenthetical has been changed to read “(which are a representative subset of the 1,700 genomes included in the NHGRI Dog Genome Project Database; Supplementary Data 3; 27)”, and comes directly after the mention of the 276 modern dogs, wolves, and outgroup canids. We hope this has made it clearer.

Line 315

“signatures, , our”

Comma mistake?

Corrected to “signature, our”

Lines 325-326

“Dogs can be broadly separated into two lineages: an Eastern lineage, primarily found in Arctic, East Asian, and American dogs prior to AD1492,”

This sounds like American dogs went extinct in AD1492, consider modifying a bit, maybe write they went extinct in historical time? potentially cite “Lin, Audrey T., et al. "The history of Coast Salish “woolly dogs” revealed by ancient genomics and Indigenous Knowledge." Science 382.6676 (2023): 1303-1308.”?

This sentence has been changed to read “...an Eastern lineage, found in Arctic, East Asian and pre-contact American dogs (which mostly disappeared post AD1492; 55)...”. We have also added a citation to Lin et al. 2023.

Supplementary line 200

Should there be a comma in 15000?

Corrected to 15,000

Supplementary Data 3

Is there no literature reference for “Kendrick's Cave”? Is it published as part of this study?

Yes (although we assume you mean Supplementary Data 2). The correct citation is (Richards and Hedges 2000), which has been added. We have also added DOI's for all dates not generated in this study.

Finally about the ID of Khatystyr1_8645, should that be Khatystyr1_9682 if the age is the number?

Corrected, as these numbers do reflect the direct or contextual ages of the samples.

Well done,

Best,

Mikkel Sinding

Referee #2 (Remarks to the Author):

The revision is strong, cohesive, and very close to acceptance. The paper now delivers a convincing synthesis that Late Glacial to early Holocene dogs were present and geographically widespread in Europe and Anatolia. The statistical framework is sound, the archaeological context is clearer, and the figures support the narrative effectively. The manuscript will be valuable to archaeogenetics, zooarchaeology, and Paleolithic studies. With a few targeted clarifications to align causal phrasing and transparency, this is ready for publication.

First, please keep the link between dog dispersal and Epigravettian-related human expansions as one plausible scenario rather than the primary driver. Briefly list alternatives in the main text, for example, postglacial connectivity among multiple western European and Mediterranean refugia, or exchange networks that need not track major ancestry shifts. Add one sentence stating that present sampling does not yet discriminate among these mechanisms.

We agree that the dispersal of dogs alongside the Epigravettian expansion is just one possible scenario. To reflect this uncertainty, we have added the following sentence to the end of the paragraph:

“However, an alternative scenario which involves dogs being exchanged through networks that are not associated with the Epigravettian expansion cannot presently be ruled out due to the paucity of Palaeolithic dog remains.”

Second, add a short paragraph explicitly reconciling the early two-source structure with later Neolithic, Bronze Age, and Iron Age changes. A simple forest plot that summarizes qpAdm proportions across periods with medians and uncertainty would separate structural continuity from shifts in mixture fractions.

We have included a new figure in the Supplementary Appendix (Supplementary Figure 21 – see below).

Supplementary Figure 21. Boxplot of the proportion of East Eurasian (D_Zhokhov1_9515) ancestry in Mesolithic to Medieval European dogs, estimated using qpAdm. Individuals where dual-source East and West Eurasian dog ancestry (D_Zhokhov1_9515 and D_Pinarbasi1_15787) was not the best fitting model (i.e. P-value does not exceed a threshold of 0.01) are shown in red. Periods are defined as: Mesolithic (13,000–9,000 years ago), Neolithic (9,000–5,200 years ago), Bronze Age (5,200–2,800 years ago), Iron Age (2,800–1,500 years ago) and Medieval (1,500–500 years ago). Statistical outliers are labelled.

This details the proportion of East Eurasian dog ancestry in dogs for a particular archaeological period. We include additional text in the SI (line 1151), which details interpretation of these results and the erosion of East Eurasian dog ancestry in Europe:

“After initial admixture between East and West Eurasian dog lineages in Mesolithic Europe (mirrored in human genomes), we observe an overall decrease in the proportion of East Eurasian ancestry in European dogs through to the Medieval period. This may represent a reduced influx of East Eurasian dog ancestry after the Mesolithic, an influx of dogs that carry West Eurasian dog ancestry (e.g. from the Near East), or the persistence of dog populations that carry West Eurasian dog ancestry from the Mesolithic onwards that are not represented in the current Mesolithic dataset.”

Third, for f4-ratio or related estimates, give standard errors or confidence intervals, list the exact

right-set populations, and justify the Near Eastern proxy choice. If possible, include more than one Near Eastern wolf proxy and summarize concordance.

Standard errors for f4-ratio comparisons are shown in Extended Figure 6 – the caption has been modified to:

“F4-ratio showing proportion of Near Eastern wolf ancestry ($\alpha \pm$ standard error) in Near Eastern dogs using the ancient Iranian wolf (W_Wezmeh1_2708) as a source.”

All right populations used in both f4-ratio and qpAdm are detailed in both the methods section of the main text and supplementary materials.

The Near Eastern proxy choice has been justified in the main text as:

“We used a newly-sequenced genome (3x) from a ~2,700-year-old wolf from Iran (Wezmeh Cave; Figure 1A) as a proxy for Near Eastern wolf ancestry, since some present-day Near Eastern wolf populations (e.g., Israel and Saudi Arabia) likely acquired dog ancestry through hybridisation more recently (Extended Figure 6A).”

Fourth, in the supplement, please provide the complete list of coyote outgroups with estimated dog ancestry and uncertainty. Add a compact sensitivity check using an alternative canid or non-canid outgroup for a few sentinel tests to demonstrate that small levels of dog introgression in coyotes do not shift inferences.

We have added the following section to the Supplementary Material (line xxx):

Outgroup Selection

We combined four coyotes as outgroups for our analyses. These were selected based on their low level of detectable dog ancestry with respect to Coyote01, i.e. the coyote which share the lowest drift with dogs/wolves as measured by outgroup-f3 statistics of the form $f_3(\text{BlackBackedJackal01}, \text{Coyote}, \text{Dog/Wolf})$.

We then used a F4 ratio of the form:

$$\alpha = \frac{F_4(W_BeleyaGora1_18148, \text{BlackBackedJackal01}; X, \text{Coyote01})}{F_4(W_BeleyaGora1_18148, \text{BlackBackedJackal01}; Y, \text{Coyote01})}$$

where X represents coyotes with suspected dog ancestry, and Y represents the following diverse dog ancestries: Palaeolithic West Eurasian (D_Pinarbasi1_15787), Near Eastern (D_TepeGhelaGap1_5826), and East Eurasian (D_Zhokhov1_9515), to calculate excess dog ancestry in the other three coyotes used as outgroups. This test

shows that all three coyotes possess marginal excesses of dog ancestry: Coyote02 (5.2–6.2%), Coyote05 (2.5–3.3%), and Coyote06 (2.8–3.6%).

To determine the impact of this low level dog ancestry on our analyses, specifically those testing for gene flow from Near Eastern wolves into dogs, we ran D-statistics of the form: $D(\text{Outgroup}, \text{Near Eastern Wolf}, X, D_Pinarbasi1_15787)$, where "Outgroup" is either Coyote01, or all four coyotes (Supplementary Figure 21). We found no significant difference between the D-statistics ($t = -0.26437$, $p\text{-value} = 0.7919$) or Z-scores ($t = 0.51575$, $p\text{-value} = 0.6069$) between the two tests, which suggests the impact of low-level dog ancestry in our coyote outgroup is negligible.

We suspect that the low amount of dog ancestry in these coyotes has a minimal impact on our results because it is unlikely to affect allele polarisation, but rather lead to conflicting genotypes among coyotes which will lead to missing data.

Supplementary Figure 21. Boxplot of D-statistic comparisons of the form: $D(\text{Outgroup}, \text{Near Eastern Wolf}, X, D_{\text{Pinarbasi1_15787}})$, where "Outgroup" is either Coyote01 (grey), or all four coyotes (red). There was no significant difference between the two tests for each individual, indicating a negligible impact of variable dog ancestry in coyotes on allele polarisation.

Fifth, please consider including the imputation probabilities in the supplementary table. Also, in the kinship paragraph, report a single summary statistic in the main text, for example, median READ theta with an interquartile range, and place full pairwise tables in the supplement.

All reference to imputation was removed from the manuscript in the previous revision.

We favoured outgroup-f3 statistics rather than kinship, as they are more robust to variations in sequencing coverage. However, as we present this data in the supplementary material, we agree with the review that some statistics should be referenced in the main text. To that end, we have added the following:

“Furthermore, pairwise distance calculations, outgroup-f3 (Extended Figures 3–5), and kinship ($\theta_{\text{Gough'sCave/Pinarbaşı}}: 0.0998$, $\theta_{\text{median}}: 0.0051$; Supplementary Figures 7-8)…”

Finally, we have opted not to include the full pairwise comparison values of all pairwise kinship comparisons in the supplement.

Last, please typeset the delta (δ) symbol in italics in all isotope notations.

Corrected.

Referee #3 (Remarks to the Author):

In general the authors have done a very robust job in addressing most of my earlier criticisms of the paper. It is now much easier to read and the conclusions are more grounded, and involve less handy-wavy speculation. The new section describes the model of the spread of paleolithic dog ancestry is much easier to understand, and supported by some interesting new analyses involving dog and human ancestry across space.

There are a few minor points/issues that perhaps need some clarification in a finished draft (but do not necessarily require my re-review).

Basal Ancestry in Gough's Cave

The supplementary material mentions the presence of some form of basal ancestry in the Gough's cave sample (line 1149 Supplementary) that appears to complicate its use in downstream analysis. It is curious that the authors do not mention this in the main manuscript at all, and I wonder why, given it seems an interesting potential finding. There seems to be some similarity to the wolf from Gough Cave (Supp fig 18). It seems to somewhat parallel the idea of near eastern wolf ancestry in near eastern dogs. And this somewhat contradicts the conclusion about a barrier to gene flow between dogs and wolves by the upper paleolithic (line 472-475).

We agree that this is an interesting finding. The exact source of this ancestry in the Gough's cave sample (unlike the Near Eastern example), however, remains unclear and thus we decided not to expand on this issue in the main text.

We suspect this ancestry does not derive from wolves since it is basal to all canids in our dataset, which is why we do not believe it contradicts our conclusions. Whilst we have not found any evidence of human-based contamination, there are other contaminants that may lead to a similar signal. Further, the similarities indicated between the dog and wolf from Gough's Cave (e.g. Supplementary Figure 18) may reflect contaminants shared by both samples from archaeological, curatorial or lab-based sources. The presence of this unknown ancestry in the Gough's Cave sample is why we use the Pinarbasi sample as the reference for "Palaeolithic ancestry" rather than both samples.

We have now included a statement in the section "Basal ancestry in Gough's Cave dog" in the Supplementary Material:

"Given the presence of this unidentified ancestry component in D_GoughsCave1_14269, we solely use D_Pinarbasi1_15787 as the representative genome of Palaeolithic West Eurasian dog ancestry in all nuclear-based genomic analyses."

Dog-Human Ancestry Comparison

Lines 236-251 (also lines 712-721, ext. Figure 3, Supp. lines 924-965): "We assessed the extent

to which dogs and humans shared their demographic history by comparing the genetic similarity of dogs from Pınarbaşı and Gough's Cave to that of humans at each site."

This analysis is detailed under Dog-Human Ancestry Comparison in the supplemental material. This describes an interesting isolation-by-distance pattern inferred independently within each species (human and dog), which is new to the manuscript and certainly adds some useful and valuable insights. The authors show that genetic distance between ancient humans is spatially autocorrelated, and the genetic distance between ancient dogs is spatially autocorrelated. The following paragraph describes the partial Mantel test, used to compare pairwise outgroup-f3 matrices in humans and in dogs. A p-value is given in the main text on lines 242-243 ("p < 0.0002"); attaching significance to the resulting Mantel r and used to support the conclusion that "human and dog ancestries are closely linked" (line 244). This is stated as reflecting "shared demographic history between humans and dogs" (supp. line 951). Fundamentally I do not think this analysis addressed shared demographic history and that it really doesn't answer a question beyond both species being spatially autocorrelated. For example, if we took other species that don't actually share a demographic history, we may also find a similar pattern simply due to the general common feature of spatial autocorrelation.

We have now clarified in the text that we perform a partial Mantel test correcting for space and time distance. This implies that the observed significant positive correlation can be attributed to shared evolutionary history, after accounting for the underlying space-time structure, due to the effects of isolation-by-distance and evolutionary change through time. We have also noticed a typo in the manuscript: the p-value is equal to 0.0001 and not 0.0002

We have now modified the text in the section titled 'High degree of similarity between Palaeolithic dogs across Western Eurasia' as follows:

'After correcting for time, and spatial autocorrelation (see Methods), we found a strong positive correlation (Mantel $r \approx 0.40$, $p < 0.0001$) between human-human and dog-dog outgroup-f3 values, suggesting shared evolutionary histories between humans and dogs that cannot be explained by shared spatial or temporal structure alone.'

We have also added the following to the Methods section:

'We standardised the data to have zero mean and variance one, and evaluated the correlation between the two corresponding matrices using a partial Mantel test whilst controlling for space and time. Partial Mantel test allows assessing the association between two distance matrices while statistically controlling for the underlying spatial and temporal structure. Thus, any significant positive correlation would be consistent with shared demographic histories between humans and dogs, beyond that expected by isolation-by-distance (Wright, 1946) and evolutionary change through time.'

Additionally, there are potential statistical issues with this approach. A number of papers in the literature have demonstrated that the Mantel test (and partial Mantel test) suffer from an excess

of Type I error in the case where both variables being tested are structured or autocorrelated (Guillot & Rousset, 2013; Quilodrán et al., 2023). Quilodrán et al. 2023 explicitly simulate a similar scenario, demonstrating the resulting Type 1 error rates. In addition, on page 11 they suggest some helpful guidelines for alternatives and conservative interpretation. I would urge caution in interpretation of the results of this test and how it's described in the manuscript.

To evaluate the impact of spatial structure on our distance-based analyses, we estimated the scale parameter k of an exponential spatial covariance model following Quilodrán et al. (2023). Because our empirical distances are measured in real units rather than on a unit square, the fitted k was substantially larger than 1, indicating that spatial autocorrelation decays only slowly across the study area and is therefore stronger than the highest autocorrelation levels examined in their simulations. Quilodrán et al. show that when both matrices in a Mantel or MRM framework exhibit moderate to strong autocorrelation, Type I error rates become inflated and the usual $\alpha = 0.05$ threshold is no longer appropriate. In such cases, they recommend adopting a conservative significance threshold proportional to the inflation factor observed in their simulations. Because our estimated autocorrelation exceeds their maximum simulated range, we applied their most conservative correction (α approximately 0.0045). Using this corrected α -level, our observed association ($p = 0.0001$) remains significant, indicating that the relationship cannot be explained by shared spatial or temporal structure alone. This approach follows the suggestions provided by Quilodrán et al. (2023) for distance-based hypotheses involving strongly structured matrices.

We added the following text into the methods section to clarify this:

“Following the suggestions in Quilodran et al. (2023) concerning inflated Type I error rates when both similarity matrices exhibit strong spatial autocorrelation, we consider a conservative significance threshold 0.0045. Using this corrected α -level, our observed association ($p = 0.0001$) remains significant, indicating that the relationship cannot be explained by shared spatial or temporal structure alone.”

The genomic legacy of Western Eurasian Palaeolithic dogs

This section focuses on a straw-man hypothesis (stated on lines 404-407) apparently drawn from the literature (initially citing a study of ancient humans). Line 437 cites the dog literature they are referencing. Even being charitable in the interpretation of east Eurasian dog ancestry, this is not addressing a hypothesis that seems to exist in the literature.

This confusion stems from the fact we included the wrong citation (Damgaard et al. 2018, rather than Botigué et al. 2017) when referencing the existing argument that eastward migration of Steppe pastoralists introduced East Eurasian dog ancestry into West Eurasian dog populations.

The following is an excerpt from the Botigué et al. 2017 paper, which presents this hypothesis: “Furthermore, we detect an additional ancestry component in the End Neolithic sample, consistent with admixture from a population of dogs located further east that may have migrated concomitant with steppe people associated with Late Neolithic and Early Bronze age cultures,

such as the Yamnaya and Corded Ware culture.”

Putting its existence aside, the relevance of this hypothesis and how it is addressed by the new data in this study is unclear. Lines 414-416 reference the same hypothesis. Why did Eastern Eurasian ancestry reach parts of Europe earlier than we thought? The text states “other parts of Europe”. What other parts of Europe exactly? The dog from Veretye was modeled as deriving ancestry from a similar east Eurasian source in 2020, so this is not exactly new information.

Firstly, while it is true that our 2020 publication first detected Eastern dog ancestry in far Eastern European dogs from Karelia, it was not until Poth et al. 2023 that the impact of Eastern Hunter-Gatherer ancestry in Mesolithic Europeans became clear. As a result, this hypothesis was not available at the time and was therefore not proposed or reported in earlier papers.

Another reason we did not put forward this hypothesis previously is that an earlier arrival of this ancestry in Europe could not be ruled out: earlier studies lacked genomes from both the Mesolithic and the Palaeolithic. Here, we show that East Eurasian dog ancestry is absent in the Palaeolithic but present in the Mesolithic, indicating that it arrived much earlier than previously thought—likely linked to the movement of Eastern Hunter-Gatherers rather than Steppe pastoralists. We consider this an important finding, made possible only by synthesizing our new dog data with the human genomic data published in 2023.

The argument on lines 433-437 reference the results of the partial Mantel test (which has issues that were described above). Even ignoring those issues, I do not think the test is showing a correlation between particular ancestries at any time or place. What is the relevance of this new data to this question? It isn't stated. I would suggest this section be either removed or reduced somewhat. The paper as it is does not really need it.

Given the above, we have opted to retain this section of the manuscript.

Modern dog model fit

Line 44: “Our qpAdm analysis also indicates that the Eastern Eurasian dog ancestry persisted in European dog populations from the Mesolithic to the present. Approximately 30% of this ancestry component is present in dogs during the Neolithic (e.g., Pločnik, Serbia), 33% during the Bronze Age (e.g., Belverde Di Cetona, Italy), 22% during the Iron Age (e.g., Parknabinnia, Ireland), 18% during the Medieval period (e.g., Vilnius Castle, Lithuania) and ~20% in modern dogs (Figure 3B). Whilst this does not preclude additional influxes of Eastern Eurasian dog ancestry into Europe post- Mesolithic, 52,56 these findings show that the fundamental ancestry components in European dogs (i.e. both Eastern and Western Eurasian ancestry) which characterises many popular modern breeds including the Cavalier King Charles Spaniel, German Shorthaired Pointer, Jack Russell and Labrador (Supplementary Figure 17) was established by at-least 10,900 years ago..”

The authors mention the fit to modern samples here, but actually Supplementary Figure 17 seems to show a very poor fit for almost every modern dog tested, with very low qpAdm p-values. So this statement does not seem particularly justified based simply on this analysis.

We have changed the final sentence of that paragraph to relate specifically to the name-checked breeds, rather than the diversity of European dogs as a whole (which as we alluded to, may have received additional contributions from East Eurasia):

“Whilst this does not preclude additional influxes of Eastern Eurasian dog ancestry into Europe post-Mesolithic, 58,62 these findings show that the fundamental ancestry components (i.e. both Eastern and Western Eurasian ancestry) in European dogs were established by at least 10,900 years ago, and persisted into modern breeds (Supplementary Figure 17).”

Dog Exchange

Line 459: “The presence of dogs in close association with humans possessing both Epigravettian and Magdalenian ancestry, however, implies exchange of dogs between culturally distinct human populations during the Late Upper Palaeolithic, sometimes in the absence of widespread gene flow or population turnover, as seen at Gough's Cave.”

I wonder if the wording here should be softened a bit. While “exchange” can of course mean many different things depending on the context, as it’s used here, it kind of implies dogs were actively being exchanged, almost like they were being traded. This may not be the author’s intention but that’s how it reads. However, there is no reason to distinguish this kind of active movement between human cultures, and ones where dogs just came with one culture and then ran around like village dogs and became part of the gene pool in the region.

In response to reviewer one, we have added the following sentence at the end of the paragraph which explains the mechanisms of dog dispersal:

“However, an alternative scenario which involves dogs being exchanged through networks that are not associated with the Epigravettian expansion cannot presently be ruled out due to the paucity of Palaeolithic dog remains.”

We do not believe that dogs could have rapidly dispersed on their own across Western Eurasia, and then been incorporated into human cultures thousands of kilometers apart (for instance, cultural attitudes towards wolves in non-dog societies would have restricted this). The close association described in the paper more generally also indicates close human-dog interaction both in-life and in-death.

Minor edits:

Line 371: incorrectly states the results of the supervised ADMIXTURE analysis are shown in supplementary figure 12. Consequently the reference to figure supplementary figure 14 on line 386 is incorrect.

Corrected to Supplementary Figure 14.

Line 336: “Palaeolithic dogs from Europe and Anatolia share more drift with the Western Eurasian lineage (Extended Figures 3–5)”

I do not think Extended Figure 3 is relevant here.

Corrected to solely Extended Figure 3

Line 516: incorrectly references supp. fig 17 for additional admixture bayes graphs.

Corrected to Supplementary Figure 16.

Extended Figure 3: Might be useful somewhere in the main text or figure legend to explain that the geographic distance and time are standardized with mean 0, otherwise the units as presented in the figure are a little confusing.

The following has been added to the caption of Extended Figure 3:

“Values for each comparison were normalised based on median absolute deviation (MAD)”.

Referee #4 (Remarks to the Author):

I co-reviewed this manuscript with one of the reviewers who provided the listed reports.